# Pitfalls and Caveats in Applying Chromogenic Immunostaining to Histopathological Diagnosis

**DOI:** 10.3390/cells10061501

**Published:** 2021-06-15

**Authors:** Yutaka Tsutsumi

**Affiliations:** 1Diagnostic Pathology Clinic, Pathos Tsutsumi, 1551-1 Sankichi-ato, Yawase-cho, Inazawa 492-8342, Aichi, Japan; pathos223@kind.ocn.ne.jp; Tel.: +81-587-96-7088; Fax: +81-587-96-7098; 2Specially Appointed Professor, School of Medical Technology, Yokkaichi Nursing and Medical Care University, 1200 Kayou-cho, Yokkaichi 512-8045, Mie, Japan

**Keywords:** chromogenic immunostaining, epitope retrieval, fixation, histopathological diagnosis, sensitivity, specificity, trouble-shooting

## Abstract

Chromogenic immunohistochemistry (immunostaining using an enzyme-labeled probe) is an essential histochemical technique for analyzing pathogenesis and making a histopathological diagnosis in routine pathology services. In neoplastic lesions, immunohistochemistry allows the study of specific clinical and biological features such as histogenesis, behavioral characteristics, therapeutic targets, and prognostic biomarkers. The needs for appropriate and reproducible methods of immunostaining are prompted by technical development and refinement, commercial availability of a variety of antibodies, advanced applicability of immunohistochemical markers, accelerated analysis of clinicopathological correlations, progress in molecular targeted therapy, and the expectation of advanced histopathological diagnosis. However, immunostaining does have various pitfalls and caveats. Pathologists should learn from previous mistakes and failures and from results indicating false positivity and false negativity. The present review article describes various devices, technical hints, and trouble-shooting guides to keep in mind when performing immunostaining.

## 1. Introduction

Techniques of immunohistochemistry (IHC) or immunostaining have already been established [1,2,3,4,5,6,7,8]. Particularly for diagnostic IHC, chromogenic IHC (immunostaining using enzyme-labeled probes) is commonly utilized. Measures and tools to prevent technical artifacts and appropriate trouble-shooting tips are needed. The present review article provides an overview of technical aspects, knowhow, pitfalls, and trouble-shooting guides for routinely performing chromogenic immunostaining, especially for experienced personnel with immunohistochemical expertise. It should be emphasized that all data presented herein were obtained by the author and/or the author’s colleagues.

## 2. The Need for Chromogenic Immunostaining in Diagnostic Pathology

IHC is critical both to informing the biology of lesions and to formulating the histopathological diagnosis. The patient’s prognosis and response to therapeutic approaches may thus be predicted.

In daily diagnostic pathology services, chromogenic immunostaining using formalin-fixed, paraffin-embedded (FFPE) sections is indispensable to help in the histopathological diagnosis fundamentally based on hematoxylin and eosin (H&E) staining. Differential diagnoses are determined first, and appropriate immunohistochemical markers for the diagnoses are then selected. The expression of lymphocyte surface markers, hormones and tumor markers contributes to the functional classification of tumors. The degree of malignancy can be analyzed by immunostaining for p53 and Ki-67 (MIB-1), a cell proliferation marker, in a variety of malignancies including carcinomas of the oral cavity, colon, and breast [9,10,11].

IHC markers can be classified into four types, as with other biomarkers: diagnostic, prognostic, predictive, and therapeutic. A diagnostic marker defines the nature (histogenesis/origin) of the lesion. A prognostic marker suggests clinical/biological characteristics that provide information on the likely course of the disease and inform a probable outcome of the patient. A predictive marker predicts the response of the lesion to targeted therapy, and a therapeutic marker represents a structure that can be used as a target of therapy.

We now have a variety of cell-specific markers available to clarify the nature of neoplasms and non-neoplastic lesions. Cytokeratins are useful to show the epithelial nature of the cells (Figure 1) [12]. For determining molecular targeted therapy of breast cancer, for example, such markers as estrogen receptor (ER), progesterone receptor (PgR), human epidermal growth factor receptor-2 (HER2), and Ki-67 are consistently immunostained [13].

Appropriate applications of chromogenic immunostaining to diagnostic pathology are dependent upon the following three points: (a) how to stain, (b) how to select IHC markers, and (c) how to evaluate the immunostaining findings. The present review article focuses on chromogenic immunostaining for diagnostic pathology that routinely uses FFPE sections.

## 3. Selection of Methodology for Chromogenic Immunostaining

A variety of primary antibodies and immunostaining kits are commercially available, and automated instruments have become popular. Now, chromogenic immunostaining is adequately sensitive and technically reliable. As a secondary reagent with high sensitivity, horseradish peroxidase (HRP)-labeled polymers such as Envision Flex (Agilent Technologies/Dako, Santa Clara, CA, USA), Simple Stain Max (Nichirei, Tokyo, Japan), and Novolink (Leica Biosystems/Novocastra, Nussloch, Germany) are widely used [14]. When necessary, catalyzed signal amplification (CSA)-II using fluorescein isothiocyanate (FITC)-labeled tyramide (Agilent/Dako) can be applied in which the anti-FITC antibody mediates amplification of the chromogenic signal (Figure 2) [15]. We have now abandoned biotin-labeled techniques such as the avidin biotinylated peroxidase complex (ABC) method and labeled streptavidin biotinylated peroxidase (LSAB) method for the following reasons: (a) endogenous biotin activity in mitochondria is retrieved by pretreatment with heat-induced epitope retrieval (HIER) [16] (for a clear explanation, see Section 7.8.2), (b) three steps are required for immunostaining, and (c) the sensitivity of detection is inadequate when compared with the polymer technique. The original version of CSA using biotinylated tyramide should thus be avoided [15].

Antigenic detection with ultra-high sensitivity techniques such as CSA-II is not necessarily needed because of the augmentation of background staining and the difficulty in maintaining the highly diluted antibodies in refrigerators or freezers. The method used should be selected based on ease of handling, stability, and reproducibility. The antibodies must be diluted with 0.01 M phosphate-buffered saline (PBS), pH 7.4, containing 1% bovine serum albumin (BSA) to avoid inactivation of antibody reactivity. Repeated freezing and thawing damages the diluted antibodies.

When there is a 100 µL aliquot of an antibody at a working dilution of 1:1000, a recommended management suggestion for the antibody is as follows. Half of the volume is kept in an undiluted form, and the remaining 50 µL is diluted at 1:10 with BSA-PBS to allow 10 plastic tubes containing the remaining 50 µL of 1:10-diluted antibody to be stored in a freezer after appropriate labeling. It is critically important to avoid the repeated freezing and thawing of diluted antibodies to ensure reliable and reproducible chromogenic immunostaining. To eliminate the risk associated with repeated freezing and thawing cycles, the addition of 25–50% glycerol in the diluent as a cryopreservative is a convenient and practical method of choice [17].

FFPE sections are the common target of chromogenic immunostaining in diagnostic pathology, so the antibodies selected must be applicable to the FFPE sections. Diaminobenzidine (DAB) solution, containing 20 mg DAB and 3 mg hydrogen peroxide in 100 mL of 50 mM Tris-HCl buffer, pH 7.6, is routinely utilized for the final chromogenic reaction in brown. The nuclei are briefly counterstained with Mayer’s hematoxylin in deep blue (usually by dipping for 10 s).

## 4. Application of Double Staining

Double staining is a method of simultaneously localizing two different antigenic substances in a single section. Immunofluorescent IHC can be applied to FFPE sections to give beautiful double labeling [18,19]. Chromogenic IHC (immunostaining with enzyme-labeled probes) is also applicable, as described below.

Two different colors are used for the differential localization. For example, Antigen A is visualized in brown with a mouse monoclonal antibody and the HRP-labeled anti-mouse IgG polymer reagent, whereas Antigen B is localized in blue with a rabbit polyclonal antibody and the alkaline phosphatase (ALP)-labeled anti-rabbit IgG polymer reagent [20]. Representative examples are displayed in Figure 3.

It should be noted that because ALP activity is hampered by the presence of inorganic phosphate iron [21], Tris-buffered saline (TBS) should be used for rinsing instead of PBS. Endogenous ALP activity is totally quenched during the preparation of FFPE sections [22]. To inhibit endogenous ALP activity in cytology specimens or fresh-frozen sections, 5 mM levamisole hydrochloride should be added to the reaction mixture for the final chromogenic reaction with azo dyes [23]. The sensitivity and beauty of chromogenic immunostaining are inferior to immunofluorescence techniques, but the cellular localization of the antigen is much clearer and more distinct than with immunofluorescent IHC. Under a fluorescent microscope, it may be difficult for pathologists to identify what types of cells are positive, particularly when focal positivity is obtained.

Double immunostaining with enzyme-labeled probes should be applied, particularly when the intracellular localization of the antigens is separated (e.g., the nucleus in one and the plasma membrane in the other localization). When the intracellular localization of antigens overlaps (i.e., both antigens are localized on the plasma membrane), immunofluorescent IHC should be used. Double immunostaining with enzyme-labeled probes is superior to immunofluorescent IHC, in that the stained sections can be preserved permanently.

Two different neuropeptides in single nerve fibers can be visualized with the following meticulous technique of chromogenic immunostaining [24]. This approach provides an alternative method for double immunofluorescent IHC. Consecutive sectioning is not suitable in this situation because the same nerve fibers cannot be included in two separate sections at 4 μm thickness. For example, antibodies raised in rabbit are used here. The first neuropeptide is immunostained with 4-chloro, 1-naphthol for HRP coloration or with azo dyes for ALP coloration in blue. After photographing, the sections are soaked in glycine-hydrochloric acid buffer, pH 2.2, to dissociate the primary antibody. Then, the dyes can be removed by dipping sections in 100% ethanol. The second neuropeptide is visualized with the DAB coloration in brown, and the same area should be photographed again to compare the localization of the two neuropeptides.

Here, the author wants to emphasize the usefulness of double staining for chromogenic immunostain and routine special stains, such as periodic acid-Schiff (PAS), alcian blue, Congo red, and Berlin blue (Figure 4). In the case of PAS reaction, periodic acid oxidation functions as a quencher of endogenous peroxidase activity [25]. Therefore, the section is first oxidized with 0.5% periodic acid solution for 10 min, and then immunostaining with DAB coloration should follow. Finally, the Schiff reagent is reacted to complete the double staining. It is noteworthy that periodate oxidation destroys sugar molecules, so carbohydrate antigens such as CA19-9, CA15-3, CA125, blood group substances, and some lymphocyte surface antigens including CD4 become undetectable. In this situation, periodic acid oxidation should be done after the completion of immunostaining with DAB coloration.

It should also be noted that periodate oxidation has an effect on strengthening nuclear staining by hematoxylin or methyl green. Namely, nuclear staining can be achieved in a shorter period after periodic acid oxidation [26].

## 5. Effect of Formalin Fixation

### 5.1. Antigenic Alterations Caused by Formalin Fixation

The target sections for immunostaining are routinely fixed in 10% or 20% formalin (3.5% or 7% formaldehyde aqueous solution). The formalin solution naturally contains formic acid as a result of auto-oxidation of formaldehyde, so the pH value in formalin solution is low. Formalin fixation provokes a cross-linkage of protein molecules and fragmentation of DNA and RNA, but it causes relatively minor damage to sugar moieties. The damage to the antigenicity or the destruction of genomic information on DNA or RNA can be mitigated when buffered or neutralized formalin is used. Noteworthy is that genomic DNA or total RNA extracted from acidic FFPE sections can be amplified by polymerase chain reaction (PCR) for DNA analysis and reverse transcription-PCR (RT-PCR) for RNA analysis, when the primer pairs are designed to yield a target genomic sequence as short as 100 base-pair lengths [27,28].

The antigenicity of certain antigens may be weakened or even lost when the specimen is fixed in acidic formalin and embedded in paraffin. Representative examples are shown in Figure 5. However, we can now choose commercially available antibodies applicable to FFPE sections. When combined with high-sensitivity detection kits and epitope retrieval techniques described below, a considerable number of antigens can be reproducibly visualized in FFPE sections.

### 5.2. Selection of Fixatives

Formaldehyde provokes a cross-linkage between tissue protein components such that the epitope retrieval step is often needed. In brief, the epitope retrieval procedure can break and loosen the protein cross-linkage to expose the masked antigenicity, as described in Section 7.8. In contrast, dehydration-induced protein aggregation is the main mechanism of fixation in such organic solvents as ethanol and acetone. In general, the antigenicity and DNA/RNA structure are well preserved in organic solvent-fixed paraffin sections. Sato et al. invented cold acetone-based tissue processing nicknamed the AMeX method that can preserve not only antigenicity but also high-molecular-weight DNA sequences in paraffin blocks [29]. Disadvantages of this method include significant tissue shrinkage that may cause difficulty in preparing sections, and the shrunken nuclei tend to be stained with hematoxylin hyperchromatically.

It is noteworthy that antigenic substances packaged in small granules in the cytoplasm are lost after fixation in ethanol and acetone [1]. This is particular true if the granules are ultrastructurally electron-dense. Presumably, the osmiophilic (lipid-rich) granule content is easily extracted by the organic solvent. Figure 6 represents false negativity of platelet factor 4 and insulin caused by fixation in acetone and ethanol. In fact, the α-granules of the platelets and β-granules of the insulin cells are rich in lipid components [30,31]. In contrast, somatostatin can be visualized in ethanol-fixed paraffin sections. It is known that somatostatin δ-granules show low electron density in ultrastructural appearance [32].

### 5.3. Use of Archival Pathology Material

The author would like to emphasize a beneficial aspect of formalin fixation by providing IHC examples of archival pathology specimens.

Some antigens and RNA signals remain detectable in archival pathology specimens fixed in acidic formalin for a considerably long period of time. In other words, fixation in acidic formalin can maintain the antigenic and genomic signals at an evaluable level of histochemical detection. Surprising examples are described below.

As shown in Figure 7, we detected HBs antigen of hepatitis B virus (HBV) and the genomic RNA sequence of hepatitis C virus (HCV) in liver cirrhosis specimens soaked in the fixative for up to 110 years, by applying immunostaining using a monoclonal antibody against HBs antigen and nested RT-PCR against the core region of HCV, respectively [33].

The author visited the Gordon Museum of Pathology at Guy’s Hospital in London, UK, where he had a chance to examine unstained glass slides of autopsied Hodgkin’s lymphoma, which was sampled by Dr. Thomas Hodgkin himself 170 years ago. The specimens were fixed in ethanol for 80 years and then in formalin for 90 years. The unstained sections were then kept at room temperature for months. By applying the cell transfer technique described in Section 10.2, immunostaining for CD15 and in situ hybridization for Epstein-Barr virus-encoded small nuclear RNA (EBER) were performed. CD15 was immunoreactive on the plasma membrane of Reed-Sternberg (RS) cells and Hodgkin’s cells, and EBER signals were detected in the nuclei of RS cells and Hodgkin’s cells (Figure 8) [34].

Empirically, the antigenicities of amyloid A protein in secondary amyloidosis, *Bacillus* Calmette Guérin (BCG) antigens in tuberculous and leprosy mycobacteria, and bacterial antigens such as streptococcal and pneumococcal antigens are steadily and reproducibly immunolocalized in specimens fixed in formalin for more than 70 years (Figure 9). HCV antigens are also immunohistochemically detectable with monoclonal antibodies in archival FFPE specimens (Figure 10).

## 6. Artifacts Caused by Formalin Fixation

### 6.1. Penetration of Plasma Proteins into the Cytoplasm of Certain Cells: A Diffusion Artifact

During formalin fixation, plasma proteins rich in plasma and tissue fluid penetrate into the cytoplasm of certain cells in FFPE sections, while the nuclei remains completely negative. These findings represent a nonspecific diffusion artifact, which the author proposed in 1984 [35]. This phenomenon is often apparent in giant cells. Plasma proteins such as albumin, fibrinogen, IgG, IgA, IgM, and the kappa/lambda chains of immunoglobulins permeate the cytoplasm of unfixed cells through a damaged plasma membrane, and the cells are then fixed by formaldehyde. The immunoreactivity is specific, but the positivity represents a fixation artifact. Typical examples include RS cells and Hodgkin’s cells in Hodgkin’s lymphoma (Figure 11) [36]. The cytoplasm of both of these cell types is polyclonally positive for both kappa and lambda chains. Such diffusion artifacts are observed in a variety of cells, including non-Hodgkin’s lymphoma cells, hepatocytes and hepatocellular carcinoma cells, normal and neoplastic thyroid epithelial cells, and epidermal keratinocytes (Figure 12) [37]. As cells immunoreactive for IgG and albumin are surrounded by completely negative cells, it seems that the positive signals indicate something, but in fact, there is no meaning at all. Figure 13 illustrates albumin immunostaining in autopsied normal pancreas. The cytoplasm of ductular cells, acinar cells or islet cells/peripheral nerve appears positive in different part of the tissue. This is why albumin should be selected as an indifferent control when IgG and α1-antitrypsin are immunostained in FFPE sections.

### 6.2. False-Positive and False-Negative Results Due to Uneven Formalin Fixation

When the specimen is immersed in formalin solution, uneven fixation may lead to both false positivity and false negativity [1]. Not only cells improperly fixed by formalin but also cells overfixed by formalin show false negativity. As vimentin in lymphocytes, macrophages, fibroblasts, and vascular endothelial cells often turns out to be negative in poorly fixed tissues, vimentin can serve as an internal control for judging whether the fixation is proper [38]. Lymphocyte surface markers such as CD20 may be positive in the peripheral part of B-cell lymphoma tissue, whereas lymphoma cells in the poorly fixed (vimentin-negative) central part are false negative (Figure 14). Figure 15 illustrates glutathione peroxidase immunoreactivity seen at the outermost part of paraformaldehyde-fixed frozen rat liver tissue, probably because of the difficulty in fixing highly soluble cytoplasmic proteins such as glutathione peroxidase. Occasionally, a reversed pattern of antigen localization is also observed. The peripheral area is overfixed to show false negativity for leukocyte common antigen (LCA, CD45), whereas antigenicity is preserved in the central part. Vimentin is negative in the CD45-positive central zone. In this case, vimentin does not function as an internal control. Figure 16 illustrates such paradoxical immunostaining findings.

If phosphorylated epitopes are targeted in immunostaining, the timely exposure to formalin is critical to obtaining consistent and reproducible IHC results. For example, phospho-heat shock protein-27 and phospho-S6 ribosomal protein, which are involved in post-translational modification and stress response pathways, increased in expression or phosphorylation levels. The phosphorylated epitopes are quite labile, and loss of antigenicity is reported within 1–2 hours of the delay in fixation [39].

To avoid such fixation artifacts, perfusion fixation can be tried in surgical and autopsy samples, i.e., transarterial perfusion fixation for the brain [40] and intrabronchial infusion fixation for the lung [41]. However, in the routine practice of diagnostic pathology, immersion fixation in formalin must be used. The size of the specimen is the key factor: for better and homogeneous fixation, the specimen should be cut as small as possible.

## 7. Tips and Pitfalls in Immunostaining

### 7.1. How to Achieve Low Background Staining

The signal-to-noise ratio should be high. One must aim for chromogenic immunostaining with distinct positive signals and minimal background staining. The recent development of epitope retrieval techniques (refer to Section 7.8.2) and the availability of a wide variety of monoclonal antibodies has significantly contributed to the art of immunostaining.

When immunostaining with high background noise is encountered, the following three possibilities should be checked first.

(a)Whether the concentration of the primary or secondary antibodies is too high.(b)Whether the titer of the antibody is insufficient.(c)Whether rinsing in PBS is insufficient.

As a prerequisite for beautiful immunostaining with a high signal-to-noise ratio, the appropriate antibody dilution should be predetermined (Figure 17). Generally speaking, monoclonal antibodies provide low background staining when compared with polyclonal antibodies. However, monoclonal antibodies of the IgM type may be aggregated during repeated freezing and thawing, which results in high background staining.

To lower the background noise, the following tips should be tried.

(a)The primary antibody should further be diluted 10 times for incubation overnight instead of a short incubation time of 30–60 min.(b)The PBS rinse should be prolonged to overnight.(c)A high concentration (1 M) of sodium chloride should be added to PBS for rinsing.d)Non-ionic detergents such as Tween 20 or Triton X-100 (0.05–0.1%) should be added to PBS for rinsing.e)Skim milk (5–10%) or BSA (1–5%) should be added to the diluted primary antibody.

To obtain satisfactory and beautiful immunostaining, the additional variables described in the following sections should also be considered, i.e., (i) the protein blocking procedure using normal animal serum, (ii) the choice of epitope retrieval procedure that has an effect on lowering background staining [42], and (iii) the choice of methodology for chromogenic visualization.

If an ALP-labeled probe is used for immunostaining, TBS should be selected for the rinsing solution instead of PBS because the inorganic phosphate iron inhibits ALP activity [21].

### 7.2. Removal of Endogenous Confounding Reactivities

#### 7.2.1. Endogenous Peroxidase Activity

Peroxidase activity in FFPE sections can be quenched by the following methods. The most common approach to inactivating endogenous peroxidase activity is dipping in 0.3% hydrogen peroxide-methanol solution or 3% hydrogen peroxide aqueous solution for 15–30 min. Oxidation in 0.5% periodic acid for 10 min is also effective [25] but is never suitable for localizing carbohydrate antigens. The hydrogen peroxide-mediated method is also harmful to the carbohydrate antigens when the treatment is too long, up to 2 h. Addition of 10 mM sodium azide to the DAB solution also assists in inhibiting endogenous peroxidase activity [43]. Eosinophils and neutrophils have strong peroxidase activity, and hemoglobin in red cells also shows pseudoperoxidase activity. Peroxidase activity in macrophages, platelets, and epithelial cells (salivary gland, mammary gland, thyroid, and renal tubules) is completely inactivated during FFPE preparations.

When a lymphocyte surface marker study is performed with ethanol-fixed fresh-frozen sections, the endogenous peroxidase quenching should be performed after the incubation of primary antibodies [44]. The epitopes of lymphocyte surface markers frequently belong to sugar moieties, and the sugar molecules in fresh-frozen sections are easily destroyed by the hydrogen peroxide-mediated oxidation step, as illustrated in Figure 18.

#### 7.2.2. Endogenous Biotin Activity

Biotin (vitamin H), a coenzyme of the decarboxylation reaction, is mainly distributed in mitochondria. Cells rich in mitochondria, such as proximal renal tubules, hepatocytes, and striated muscle cells, show strong endogenous biotin activity, particularly in fresh frozen sections or cytology preparations. When biotin-based detection systems such as the ABC or LSAB method are used, quenching of endogenous biotin activity should be performed by preincubating with avidin solution [45]. Binding between biotin and avidin is quite strong, and once bound, they never dissociate.

The endogenous biotin activity is inactivated by formalin fixation. Exceptions include “opaque nuclei” seen in endometrial gland cells, endometrioid carcinoma cells and embryonal-type lung adenocarcinoma cells [46]. The opaque nuclei reveal strong endogenous biotin activity in FFPE sections. It should be noted that the endogenous biotin activity in mitochondria-rich cells in FFPE sections is retrieved by the HIER step [16]. This is especially evident when 10 mM citrate buffer, pH 7.0, or 1 mM ethylenediamine tetraacetic acid (EDTA), pH 8.0, is used as a dipping aqueous solution (Figure 19) [47]. This is why biotin-based detection methods should be avoided, and instead, the HRP-labeled polymer technique has become the mainstream method.

#### 7.2.3. Endogenous Protein A and Protein G Activity

*Staphylococcus aureus* and *Streptococcus pyogenes* have protein A and protein G on their respective cell walls. These proteins strongly bind the Fc portion of IgG molecules, functioning as IgG-Fc receptors. Their IgG-binding activity is lost by formalin fixation, but HIER applied to FFPE sections retrieves the IgG Fc-binding activity of the cocci. This may introduce a pitfall in immunohistochemical identification of the coccal pathogen seen in FFPE sections (Figure 20) [48]. For epitope retrieval of the coccal antigens, proteinase digestion should be used instead of HIER. When the heating pretreatment is applied, preincubation with diluted normal human or animal serum is indispensable.

IgG-Fc receptors on mononuclear blood cells can bind to IgG molecules, and nonspecific binding of IgG-type antibodies to inflammatory/immune cells in tissues may occur in frozen sections of lymphoid tissue and cytological preparations. However, such interaction does not cause a problem in FFPE specimens because of the inactivation of Fc receptors during the preparation of FFPE sections [49]. Fc-binding activity of the immune cells is not retrieved by HIER. Nonspecific staining of mast cell secretory granules in chromogenic immunostaining is attributed to ionic interaction between the F(ab’)2 fragments of IgG and the heparin constituent of the mast cell granules [50].

#### 7.2.4. Endogenous Pigments

Brown or black-colored endogenous pigments may hamper the observation of the DAB chromogenic reaction of the target. Melanin shows metachromasia, so Giemsa or methyl green should be chosen for counterstaining instead of hematoxylin. The greenish-colored metachromatic melanin is clearly distinguishable from the brown DAB deposit. Hemosiderin pigments can be distinguished from the DAB product by counterstaining with Berlin blue. Figure 21 illustrates representative examples.

Black-colored formalin pigments are often dispersed in and around the lesion with hemorrhage, and formalin pigments are particularly troublesome in autopsy cases. The formation of formalin pigments is greatly reduced by the use of buffered formalin instead of acidic (unbuffered) formalin. To remove the formalin pigments, unstained slides should be treated with alcoholic solutions containing picric acid, sodium hydroxide, or ammonium hydroxide [51]. In the liver, aggregated bilirubin pigments may be confused with the specific DAB products (Figure 22).

As another trouble-shooting tip, ALP-labeled secondary reagent should be selected, instead of HRP-labeled probe [23]. The ALP reaction products with azo dyes can be visualized in red or blue.

### 7.3. Staining Artifacts: Effects of Drying, Contamination of Cytokeratin-Positive Scales, and Insufficient Deparaffinization

When sections happen to be dried during antibody incubation, DAB coloring reaction is seen in the area of drying, mainly in the periphery of the sections, because the antibody molecules are dry-fixed onto the glass slide in the area of dehydration. Sections are occasionally contaminated with cytokeratin-positive scales of epidermal origin, which may cause a false-positive judgment. Fine adjustment of the microscopic focus reveals that the positivity is seen on the section but not in the section. If deparaffinization is incomplete or when air bubbles are formed on the glass slide, round-shaped negative zones are observed because of the lack of antibody penetration. Representative features are depicted in Figure 23.

### 7.4. Antigenic Deterioration Due to Prolonged Conservation of Unstained Sections

It is convenient for us to conserve unstained glass slides in diagnostic pathology divisions as positive control sections. Importantly, for the detection of certain antigens, storage at room temperature for a long period of time is not suitable [52]. Particularly susceptible are nuclear antigens such as Ki-67 (MIB-1), p53, ER and PgR (Figure 24) [53]. Positive control sections should thus be cut from the control paraffin block just before immunostaining. Other methods of convenience are to store unstained glass slides in a refrigerator at 4 °C for a short period of time (up to 4 months) [54] or in a freezer at −20 °C or lower [53]. Economou et al. described a practical method: freshly cut slides after one day of drying should be dipped into paraffin solution as a seal to reduce oxidation [55].

### 7.5. Effects of Section-Stretching Temperature on a Hot Plate and of Drying Period after Cutting Sections

Some antigens in FFPE sections are quite susceptible to high section-stretching temperature on a hot plate (Figure 25). Immunoreactivities of glutathione-S-transferase (GST)-π, an isozyme of cytoplasmic peroxide-reducing enzyme, and orotate phosphoribosyltransferase, a cytoplasmic enzyme phosphorylating the anticancer drug 5-fluorouracil are markedly reduced after stretching on a hot plate at 70 °C for 2–3 s [56]. The temperature at 70 °C is above the melting temperature of paraffin, which is around 56–58 °C. CD8 antigen is also susceptible to high-temperature stretching. The antigenicities of HER2, epidermal growth factor receptor (EGFR), Ki-67, cytokeratins, and CD20 are mildly weakened [57]. The temperature and period for stretching should be set at 40 °C for 20 s or at 50 °C for 10 s for safety purposes [56].

Regarding the drying period after cutting sections, HER2 immunoreactivity is significantly weakened after leaving unstained sections in an incubator at 40 °C for 3 days (Figure 26) [57]. A milder negative effect on HER2 immunostaining is observed after incubation at 40 °C for 1 day. When sections are kept in an incubator at 60 °C for 3 days, immunoreactivities of cytokeratins and pepsinogen 2 appear to be weakened.

### 7.6. Effect of Decalcification

Formalin-fixed bone or calcified tissue should be decalcified before preparing FFPE tissue. Four kinds of solution are commonly used for decalcification, including 10% formic acid in formalin, Plank-Rychlo solution (containing 8.5% hydrochloric acid, 5% formic acid, and 7% aluminum chloride), 5% trichloroacetic acid, and 10% EDTA disodium solution, pH 7. EDTA-mediated decalcification, which requires a longer time for completion, has little effect on the expression of lymphocyte surface markers and Ki-67, whereas significant damage to antigenicity is observed after decalcification by formic acid solution and particularly Plank-Rychlo solution and trichloroacetic acid. Therefore, Plank-Rychlo solution and trichloroacetic acid should be avoided for IHC analysis of lymphocyte surface markers [58]. In contrast, Mukai et al. reported that tissues routinely decalcified with EDTA, formic acid, nitric acid, or Plank-Rychlo solution could be used for immunostaining without significant loss of immunoreactivity [59].

When necessary, the surface decalcification procedure is performed on paraffin blocks to prepare higher-quality FFPE sections without chattering. The surface of paraffin blocks is exposed to the decalcification solution for 30 min to 24 h. Ki-67 immunoreactivity is especially susceptible to the surface decalcification procedure (Figure 27). However, surface decalcification with formic acid for less than 1 h appears to show only a negligible effect on marker expression [60].

### 7.7. Pitfalls and Caveats in High-Sensitivity Immunostaining

HRP-labeled polymer techniques such as Envision Flex, Simple Stain Max, and Novolink have enabled pathologists to visualize various tissue antigens clearly in FFPE sections [14]. CSA using FITC-labeled tyramide (CSA-II, available from Agilent Co./Dako) is an ultrasensitive detection system for visualization of hidden antigens that have never been detectable before in FFPE sections [15]. However, the higher the sensitivity of detection, the higher the background staining [61]. Thus, intracellular localization of antigens can become blurred: i.e., the membrane antigen may diffuse into the cytoplasm (see Figure 29).

PharmDx^TM^ is an immunostaining kit available from Agilent/Dako for the detection of EGFR in colonic adenocarcinoma. The basic problem is that positive signals are commonly weak not only in cancer cells but also in the normal colonic mucosa. This kit uses EnVision as the secondary reagent. However, the substitutive use of the CSA-II system available from the same company as the secondary reagent significantly enhances the positive signals (Figure 28) [62].

When the concentration of the primary antibody is too high, false-negative results may paradoxically occur when the CSA-II system is used [63]. We evaluated the localization of CD4, interleukin-6 (IL-6), and interferon-gamma (IFN-γ) in heat-retrieved FFPE sections of pharyngeal tonsil with the CSA-II system. When the antibodies were diluted at 1:5000, clear positivity was observed in immunocytes in the FFPE sections, whereas false negative findings were seen with anti-IFN-γ polyclonal antibody at a 1:500 dilution. In the case of CD4 (monoclonal) and IL-6 (polyclonal), the antibodies at 1:50 dilution also gave false negative results (Figure 29). The background staining was increased, principally due to artificial diffusion of the membrane antigen into the cytoplasm. False negativity was partially recovered by 10-times dilution of the HRP-labeled secondary reagent and the FITC-labeled tyramide.

Automated instruments are usually programmed to detect antigens with ultrahigh sensitivity. For example, HER2 immunostaining for breast and gastric cancer using a Ventana’s Benchmark ULTRA (Roche Diagnostics, Rotkreuz, Switzerland) results in very strong positivity in lesions overexpressing HER2, and normal mammary ducts and normal gastric foveolar cells also occasionally tend to be stained moderately. When pathologists are unfamiliar with the results of this system, a false-positive determination may result for cancerous lesions not overexpressing HER2.

### 7.8. Pitfalls and Caveats in Antigen Retrieval Sequences

FFPE sections often require an antigen retrieval step before immunostaining [3,6]. Cross-linkage of cellular proteins formed during formalin fixation may mask antigenic sites (epitopes). Epitope retrieval sequences, pretreatments performed before antibody incubation, can expose antigenic sites to allow antibodies to bind by breaking or loosening the formaldehyde-mediated methylene bridges. They also have an effect on lowering background staining [42]. Two methods are mainly utilized for epitope retrieval: enzymatic digestion and HIER. To prevent detachment of sections during the pretreatments, the use of coated glass slides is required. There are different types of glass slides with enhanced adhesion properties [64]. Classically, glass slides were coated with ovalbumin or gelatin, but they are not suitable when enzymatic digestion is used. Because of the contamination of avidin molecules, ovalbumin coating should be avoided when the biotin-based method is used. The use of synthetic resins for hydrophobic coating, including 0.1% Neoprene (polychloroprene) in toluene (Okenshoji, Tokyo, Japan), 0.1% poly-L-lysine aqueous solution (molecular weight >150 kDa, Sigma-Aldrich, St. Louis, MO, USA), and 2% (3-aminopropyl)trimethoxysilane in acetone (Sigma-Aldrich) is practical. Presently, (3-aminopropyl) trimethoxysilane-coated glass slides are the most widely used.

#### 7.8.1. Proteinase Pretreatment

A variety of proteinases have been used for the digestion of FFPE sections for epitope retrieval. These include proteinase K, protease-1, pronase, actinase, trypsin, pepsin and ficin. The proteinase pretreatment is effective for retrieving the antigenicity of type 4 collagen (Figure 30) and laminin. Cytokeratins and lymphocyte surface markers may also be retrieved with this pretreatment: some will work well with proteinase digestion, but others will not.

The antigenicity of cell-bound immunoglobulins and immune deposits in FFPE sections can be restored by proteolytic enzyme digestion [65]. Glomerular immune deposits consisting of immunoglobulins and complements can be detected reproducibly by thoroughly digesting the sections (Figure 31) [66]. However, prolongation of the digestion period is required. In the case of trypsin digestion, the immune deposits are visible after 2-h treatment (4 times longer than is usual). We detected IgM deposits in glomeruli of IgM nephropathy in the FFPE autopsy kidney [67]. IgG deposition on the plasma membrane was proven in the skin, gut, and bronchus in lethal paraneoplastic pemphigus [68].

Of note is that antigenicity may be lost by the proteinase pretreatment. Representative examples include peptide hormones (substance P and gastrin-releasing peptide), IgD and the J-chain of IgA and IgM (Figure 32). The epitopes detected by certain monoclonal antibodies, such as anti-vimentin V9 and anti-cytokeratin KL-1, are lost by the pretreatment. Cytokeratin immunoreactivity detected with monoclonal antibody KL-1 is lost after trypsin treatment but enhanced by pepsin digestion. The protease pretreatment should also be avoided for localizing CD20 (L26), CD45 (LCA) (2B11+PD7/26), CD45RO (UCHL1), desmin (D33), Ki-67 (MIB-1), neutrophil elastase (NP57), neuron-specific enolase (NSE) (BBS/NC/VI-H14), p53 (DO-7), and proliferating cell nuclear antigen (PC-1) [1].

#### 7.8.2. Heat-Induced Epitope Retrieval (HIER)

Hydrated heating is quite effective for retrieving hidden antigenicities in FFPE sections [3,6]. The HIER process may represent the simple removal of formalin fixation-induced protein cross-linkage sterically interfering with the binding of antibodies to linear epitopes in tissue sections [69]. Boenisch has uniquely shown that HIER recovers electrostatic charges on the hydrophilic surfaces of antigens [70].

Typically, deparaffinized sections are heated at 60–121 °C in 10 mM citrate buffer, pH 6.0 or pH 7.0, 1 mM EDTA solution, pH 8.0, or in Tris-EDTA (10 mM Tris base, 1 mM EDTA, and 0.05% Tween 20), pH 9.0. The devices used for heating include an incubator (at 60 °C, overnight), water bath (at 95 °C), microwave oven (at 100 °C), steamer (at >100 °C), pressure pan (at 121 °C), and autoclave oven (at 121 °C). Pressure pan cooking is strongly recommended because of its stability, reproducibility, and ease of handling (Figure 33).

A variety of antigens (epitopes) can be retrieved by HIER, including intranuclear antigens (p53, Ki-67, ER, PgR, thyroid transcription factor-1 [TTF1], caudal-type homeobox-2 [CDX2], etc.), plasma membrane antigens (lymphocyte surface markers, epithelial membrane antigen [EMA], etc.), cytoskeletal proteins (cytokeratins, vimentin, desmin, actin, etc.), and secretory proteins (parathyroid hormone, a-fetoprotein, fibrinogen, etc.) [1]. Typical examples are displayed in Figure 34 and Figure 35. Peptide hormone immunoreactivity may also be retrieved by HIER. Selection of the solution is important for appropriate HIER of the respective antigens (Figure 36). The effects of HIER are dependent upon the antibodies used, so it must be predetermined whether HIER will work effectively and what kind of conditions are most suitable for its use. Supposedly, calcium iron chelation by citric acid and EDTA is a key factor in HIER. It is noteworthy that nuclear antigens such as ER and p53 show a HIER effect in ethanol-fixed cytology preparations (Figure 37) [71].

The heating treatment changes double-stranded DNA into single-stranded to allow nuclear antigens hidden within the DNA stretches to bind antibodies. Occasionally, false negativity of nuclear antigens such as p53, ER, and PgR can be experienced when monoclonal antibodies are incubated overnight at 4 °C or at room temperature. Nuclear reactivity is evident when the same antibodies at the same dilution are incubated for 60 min (Figure 38). This paradoxical phenomenon can be explained theoretically by the following: During the prolonged incubation period, heat-provoked single-stranded DNA naturally returns to its double-stranded form such that the nuclear antigens are hidden again and become inaccessible to the antibodies.

Hematoxylin is suitable for nuclear counterstaining in heat-treated sections. Methyl green stains the nuclei only faintly in the heat-treated sections. This is explained by the fact that methyl green dye has an affinity for double-stranded DNA. Nuclear affinity to hematoxylin is evidently decreased after heating in 1 mM EDTA solution, pH 8.0, so prolongation of hematoxylin staining time is necessary (Figure 39) [72]. It should also be noted that during heating in 10 mM citrate buffer, pH 7.0, sections tend to become detached.

Ironically, the antigenicity of some antigens is markedly weakened or lost after the heating pretreatment (Figure 40). These include BM-1 (a marker of myeloid precursors), neutrophil elastase (NP57), von Willebrand factor, NSE, and GST (α, µ, and π) [1].

Pathologists must recognize that the nuclear localization of Ki-67 is dependent upon how the sections are cooled down after heating [73]. When sections are rapidly cooled down in tap water after heating, Ki-67 often reveals a false-negative finding. In contrast, clear nuclear positivity is observed in sections cooled down gradually by leaving for more than 30 min (Figure 41).

When a high-molecular-weight polymer (EnVision, Agilent/Dako Co) is used as the secondary reagent, the cytoplasm of the mitotic cells may be stained for Ki-67 whereas the nuclei of the proliferative cells remain unstained (Figure 42). Such a strange phenomenon is not experienced when a secondary polymer reagent of a smaller molecular size, such as EnVision Plus or EnVision Flex, is used [73].

#### 7.8.3. Other Methods for Epitope Retrieval

Immunoreactivity of β-amyloid protein in the brain and other amyloidogenic substances, including prion protein and prealbumin (transthyretin), in various tissues are retrieved by soaking in 100% formic acid solution for 5 min (Figure 43) [74], whereas protease treatment and HIER are ineffective. In contrast, immunoreactivity of β-amyloid protein can be visualized with a mouse monoclonal antibody, D3D2N, after conventional hydrated heating. Bromodeoxyuridine (BrdU) experimentally incorporated into the nuclei of proliferative cells can be detected in FFPE sections after exposure in 2–4 N hydrochloric acid solution for 20–90 min [75]. DNase I from the bovine pancreas can also be used for the same purpose [76]. Actin immunoreactivity of intracytoplasmic eosinophilic inclusion bodies in infantile digital fibromatosis is retrieved by treating sections in 1 N potassium hydroxide in 70% alcohol for 60 min, followed by trypsin digestion [77].

## 8. How Results Are Judged

### 8.1. False Positivity or Equivocal Negativity

When the target cells appear to be only faintly immunostained, how the results are judged becomes critically important, i.e., whether they are weakly positive or equivocally negative. The author once experienced the regrettable misjudgment of a mediastinal small round cell tumor in a middle-aged man. Placental alkaline phosphatase (PALP) visualized with a polyclonal antibody was judged as weakly positive, so the author’s final diagnosis was germinoma (seminoma). However, the diagnosis was later proven to be small cell carcinoma of the lung with false-positive (or equivocally negative) PALP immunoreactivity.

In general, immunoreactivity of secretory proteins in neoplastic cells tends to be weaker than that in normal cells. Examples include chromogranin A, insulin, gastrin, von Willebrand factor and immunoglobulins. The key point in appropriate judgment is to carefully recognize the intracellular localization of the target antigen, as described in the next section.

### 8.2. Intracellular Localization Pattern of Antigens

The specificity of immunostaining can be judged by the intracellular localization pattern of the antigenic substances, as schematically illustrated in Figure 44 [1,35]. Representative immunohistochemical markers are displayed in Figure 45.

Peptide hormones and chromogranin A are localized as intracytoplasmic fine granules, but lysosomal and mitochondrial proteins are visualized as intracytoplasmic coarse granules. Secretory proteins such as alpha-fetoprotein, human chorionic gonadotropin, and immunoglobulins, which are seen as fine or coarse vesicles in the cytoplasm, represent ultrastructural localization in the rough endoplasmic reticulum and Golgi apparatus. Particularly in neoplastic cells, the immunoreactivity of secretory proteins may be accentuated and clustered in the Golgi area. Diffuse cytoplasmic reactivity is seen for cytosolic proteins such as myoglobin, NSE, creatine kinase-MM isozyme and GST. S-100 protein, heat-shock proteins, ubiquitin, and β-catenin are distributed diffusely in both the cytoplasm and nucleus. The nuclei are diffusely positive for Ki-67 (MIB-1), p53, CDX2, ER, and PgR, but nucleolar staining is frequently accentuated for Ki-67. Plasma membrane proteins reveal distinct membrane staining, and the Golgi area is also frequently positive. Carcinoembryonic antigen (CEA) is expressed along the apical plasma membrane in normal gastric and colonic mucosa, whereas in adenocarcinomas of the stomach and colon, the plasma membranes are circumferentially positive [78]. Such intracellular antigen distribution patterns are clearly recognized in FFPE sections when compared with fresh-frozen sections.

When Ki-67 is localized along the plasma membrane instead of the nucleus, it is easy to recognize that its localization is nonspecific (Figure 46). Ki-67 antigen is positive along the plasma membrane in certain tumors, and this finding has been used as a diagnostic marker [79]. PgR may also be localized along the plasma membrane of breast cancer cells, and the lymphocyte surface markers happen to be localized in the nucleus (Figure 47). When an ultra-sensitive method is used for immunostaining together with HIER, plasma membrane proteins may appear to be localized diffusely in the cytoplasm (see Figure 29). Periodic acid treatment after HIER may suppress such non-physiological (artificial) localization.

### 8.3. Specificity of Antibodies

The specificity of the antibody should be confirmed before practical use. Information supplied by the manufacturer is valuable, but one must not believe it with blind faith. It is noteworthy that rabbit antiserum occasionally contains natural antibodies against intermediate filament proteins [80]. For example, in Figure 48, a certain lot of anti-myoglobin rabbit antiserum stains epidermal keratinocytes, vascular endothelial cells and pancreatic islet cells. Monoclonal antibodies supplied as a form of ascitic fluid may show unexpected cross-reactivity due to mouse ascitic components [81].

Small molecules (haptens) such as peptide hormones and thyroxine are immunized with carrier proteins, so antibodies designated against peptide hormones also contain anti-carrier protein antibodies [82]. When bovine thyroglobulin is used as a carrier protein, the antiserum reacts with colloid and epithelial cells of the human thyroid gland. Anti-cholecystokinin (CCK) antiserum immunized with *Ascaris* proteins as the carrier protein reveals cross-reactivity of the *Ascaris* proteins to smooth muscle, cartilage, and some epithelial cells in human tissues, as well as to neuroendocrine tumors (Figure 49) [1]. Because the extract of *Mycobacterium tuberculosis* is commonly used as an adjuvant for immunization [83], many antisera may stain tuberculous bacilli and other mycobacteria in FFPE sections (Figure 50).

### 8.4. Uncommon Expression of Antigenicities Widely Used in Diagnostic Pathology

The poorer the degree of cancer differentiation, the more infrequent the expression of specific markers. The lack of marker expression does not necessarily exclude the possibility of specific differentiation of neoplastic cells. The epithelial cells may express vimentin, whereas a variety of non-epithelial tumor cells are immunoreactive for cytokeratins. Spindle cell carcinoma of the skin co-expresses cytokeratin and vimentin [84] (see Figure 35), and anaplastic large cell lymphoma may express cytokeratins [85]. The usefulness and pitfalls of cytokeratin immunostaining are displayed in Figure 51.

The lack of basal cells is an excellent indicator of prostatic adenocarcinoma. Post-atrophic hyperplasia of the prostate histologically resembles adenocarcinoma, causing diagnostic confusion [86]. The preservation of cytokeratin 5/6-immunoreactive basal cells around the respective acini is of significant diagnostic value (Figure 52).

Neuron-specific enolase (NSE) or γ-enolase is not necessarily specific to neurons and neuroendocrine cells [87]. A variety of non-neuroendocrine cells and their tumors may express NSE. Glial fibrillary acidic protein (GFAP) is known to be expressed in some myoepithelial cells and their tumors of the salivary gland [88]. Antiserum against prostate-specific antigen (PSA) may be cross-reactive to salivary gland duct cells because PSA or human kallikrein-3 shares epitopes common to other kallikreins expressed in the salivary gland ducts [89]. Prostatic acid phosphatase (PAcP) is frequently expressed in neuroendocrine cells and tumors of the rectum [90]. Representative examples are illustrated in Figure 53. Diagnostic pathologists must study and recognize such unexpected expression of “specific” markers in unrelated cells.

### 8.5. Nonspecific Adsorption of Antibodies by Certain Cells

In FFPE sections, tissue mast cells [91], neuroendocrine cells (particularly gastrin cells) [92], parietal cells of the gastric oxyntic mucosa, and HBs antigen-positive ground-glass hepatocytes in HBV carriers [93] may adsorb antibodies nonspecifically. Nonspecific staining of mast cell granules is attributed to ionic interaction between F(ab)’2 segments of IgG and the heparin constituent of the mast cell secretory granules [49]. In frozen sections, non-specific signals may be seen on the plasma membrane of inflammatory or immune cells that result from binding of the Fc portion of IgG through their Fc receptors [94]. However, such Fc receptor-mediated nonspecific staining is rarely encountered in FFPE sections. Representative examples of nonspecific adsorption of antibodies are shown in Figure 54. Stromal collagen fibers also tend to show nonspecific binding with antibodies [95]. The Fc portion of IgG is nonspecifically attracted to the basic groups present in collagen fibers [96].

### 8.6. Positive and Negative Controls

Biological and technical controls can be used for immunostaining. Biological controls are created by including tissues that are known to express or not to express the target protein. For example, for gastrin immunostaining, gastric antral mucosa should be included as a biologically positive control and colonic mucosa as a biologically negative control. Technical controls use consecutive sections of the specimen under diagnosis that are incubated with normal animal serum for polyclonal antibodies or isotype-matched immunoglobulins for monoclonal antibodies.

Negative controls that use normal animal serum are important to check the specificity of the immune reaction. When multiple antibodies are evaluated simultaneously, the antibodies mutually function as indifferent antibodies such that negative control sections may be unnecessary.

Appropriate positive controls are inevitably needed in each run of immunostaining as they function as the evidence to exclude the possibility of inactivation of the antibodies. When positive cells are consistently included in the same sections, for example, cells positive for vimentin, vascular markers, and epithelial markers, they act as internal positive controls [38].

In the diagnostic practice for breast cancer, immunostaining for ER, PgR, HER2, and Ki-67 is consistently requested [13]. When the author looked at borrowed slides sent from an outside hospital for a second opinion consultation for “ER-negative” breast cancer, not only the breast cancer cells but also the non-neoplastic mammary ductal cells were negative for ER. ER was then immunostained again in our laboratory, and clear nuclear positivity of ER was confirmed in both the normal and cancerous cells. Appropriate judgment for such technical pitfalls is critically important to provide valuable information for the treatment of the patient.

## 9. When Unexpected Results Are Obtained

Clinicopathological significance should be recognized when immunohistochemical results are different from H&E-based expectations. However, pathologists must make careful judgment by considering the specificity of the markers they immunostained. Cases of cytokeratin-positive malignant lymphoma, glioma, sarcoma, or melanoma may be encountered. Vimentin is expressed in certain normal and neoplastic epithelial cells (see Figure 35). Pathologists should recognize the expression of EMA (as a MUC-1-related molecule) on normal and neoplastic perineurial cells and plasma cells (Figure 55) [97]. CD138, a marker of normal plasma cells, may not be expressed in neoplastic plasma cells [98].

Caution is needed when the clinical and histopathological diagnoses are discrepant. First of all, we must exclude the possibility of tissue contamination or mistakes in patient identification. The container holding the formalin-fixed specimen and the paraffin block should always be checked first to verify identification. We can immunostain blood group substances such as antigens A, B, and H or Lewis a and b for blood typing. The blood group substances are expressed consistently in vascular endothelial cells and occasionally in epithelial cells, so we can apply IHC blood typing to prove tissue contamination or mistakes in the identification of FFPE sections when the blood group of the specimen differs from that of the patient [99]. Figure 56 displays successful IHC blood typing for contaminated needle biopsy specimens of the prostate, in which one specimen contained cancer but the other did not.

## 10. When Only One Preparation Is Available

It can happen that only one unstained or H&E- or Papanicolaou-stained specimen on a non-coated glass slide is available for analysis. This situation can often occur particularly with cytology specimens. Can one slide be used for immunostaining analysis? Yes, and two methods can be applied. One is the re-staining method, and the other is the cell transfer technique. See Ref [100] for details.

### 10.1. Re-Staining Method

Suppose only one glass slide already stained with H&E or Papanicolaou sequences is available. The specimen can be re-used [101]. The target cells or areas should be photographed beforehand. The first step is the removal of the cover glass in warm xylene. Then the stained dyes are bleached in acid alcohol solution (0.5% hydrochloric acid in 70% ethanol) for more than 1 h, or the specimen is left in tap water overnight to bleach the dyes. Thereafter, the glass slide is ready for immunostaining. A typical example of a cervical cytology smear re-stained for a *Chlamydia trachomatis* antigen is illustrated in Figure 57.

When a silane-coated glass slide has been used, the cell transfer technique described below is not applicable, but chromogenic immunostaining with HIER can be applied to the re-staining method. If not, the sections or cells should be transferred to a silane-coated glass slide before the bleaching step to apply to HIER-assisted immunostaining, as mentioned below. Finally, the same cells or target areas should be re-photographed. If multiple markers need to be immunostained, the antigenic substances should be visualized repeatedly with an ALP-labeled secondary reagent [20]. The azo dyes used for the color reaction for ALP, colored red or blue, can be easily bleached by soaking the specimens in ethanol.

When the immunostained (DAB-colored) preparations are re-stained, negative control sections with negative results can be used for new immunostaining. If there is focal positivity of the DAB reaction, the second immunostaining with the ALP-labeled method can be chosen to have a double-immunostained section.

### 10.2. Cell Transfer Technique

Histological sections or cytology specimens mounted on uncoated glass slides can be transferred to silane-coated glass slides [102]. The cover glasses should be removed by soaking in warm xylene. The sections or cell preparations are covered with mounting resin solution, and after leaving them for 1–2 days in an incubator at 40 °C, the solidified resin forming a thick membrane can be peeled off after soaking in warm water for 1 h. The sections or cells have thus been detached and transferred to the solidified membrane. In warm water, the resin membrane is then placed onto silane-coated glass slides. The glass slides with transferred resin membrane should be fully dried in an incubator. The resin itself can easily be removed by dipping the specimens in xylene. The resin membrane can be cut into several pieces with scissors to obtain multiple silane-coated glass slides for immunostaining (Figure 58).

Itoh et al. invented a modified method for rapid cell transfer [103]. The mounting resin solution should be diluted double with xylene, and the resin is solidified on a hot plate at 70–80 °C. With these modifications, the cell transfer can be achieved in one hour.

It should be noted that the cell transfer technique is neither applicable to dry-fixed Giemsa-stained preparations nor to preparations mounted on silane-coated glass slides, such as cytology specimens of the urine, cerebrospinal fluid, or effusions. For these specimens, the re-staining method should be tried. We found that miracle-coated glass slides for cytology practice termed TACAS (Thinlayer Advanced Cytology Assay System supplied by Medical & Biological Laboratories, Nagoya, Japan) allow both immunostaining with HIER and cell transfer. The TACAS glass slides prevent detachment of sections during immunostaining with HIER and allow cell transfer after immunostaining using the inverted beam capsule method [104]. Both the antigen and genome of severe fever with thrombocytopenia syndrome (SFTS) virus were successfully localized at the ultrastructural level with the pre-embedding technique (Figure 59).

## 11. Immunostaining with Low Specificity and High Sensitivity

The presence of pathogens infecting tissues can be proven by the use of antibodies with low or unknown specificity. Two examples are described here: the use of widely cross-reactive commercial antisera and the use of sera of patients suffering infection. Please refer to the author’s previous article [105] for additional details.

### 11.1. Use of Antisera against Pathogens Showing Wide Cross-Reactivity

The antigenicities of pathogens are distinct from those of human cells. Therefore, pathogens can suitably be immunolocalized in FFPE sections. Bacterial or fungal antigens have been visualized with antisera against BCG showing wide cross-reactivity [106,107]. Commercially available rabbit antisera against BCG, *Bacillus cereus*, *Treponema pallidum*, and *Escherichia coli* can be used for this purpose. Such immunostaining is valuable for the screening of bacterial infection in FFPE sections [105], i.e., the identification of bacteria or bacterial antigens in infectious lesions is of diagnostic value, even if the specificity is unknown.

A representative example includes *Corynebacterium kroppenstedtii* infection in granulomatous mastitis. The lipophilic bacteria are visualized with the antisera described above mainly in fat vacuoles within an inflammatory breast lesion [38]. The flesh-eating bacteria *Vibrio vulnificus*, which provokes gangrene of the extremities, can be localized in the affected skin with antisera against *B. cereus*, BCG, and *T. pallidum* [108]. *Leptospira interrogans* infecting the liver is immunoreactive with *E. coli* antiserum. The causative bacteria in *Haemophilus pertussis*-provoked pneumonia can be clearly shown in FFPE sections with all of the antisera. In colon biopsy specimens of intestinal spirochetosis, the long basophilic spiral-shaped bacteria *Brachyspira aalborgi* adhering to the surface of the colonic epithelial cells can also be strongly immunostained with all of the antisera described above [105]. Representative features are illustrated in Figure 60 and Figure 61.

### 11.2. Use of Patient Sera

The sera of patients with infectious disease have high-titer antibodies against the causative pathogen, particularly in those who are in the recovery stage or have chronic persistent infection [105,109]. When the host reactions such as abscess or granuloma are observed microscopically, the serum, diluted at 1:500 to 1:1000, is applicable as a highly sensitive probe to detecting causative pathogens in the infected lesion embedded in paraffin [105,109]. A brief phone call to a laboratorian to save a small aliquot of an infected patient’s serum is all that is needed. The indirect immunoperoxidase method using HRP-labeled anti-human IgG or immunoglobulins is suitable to obtain low-background immunostaining. The methods with high sensitivity of detection such as LSAB or the polymer method result in high background staining due to diffuse distribution of endogenous IgG in tissues. Usually, HIER is unnecessary, but it may occasionally assist in increasing the sensitivity of detection [105]. This low-cost sequence is particularly useful in cases of protozoan and helminthic infections, including cryptosporidiosis, isosporiasis, toxoplasmosis, amebiasis, schistosomiasis, ascariasis, and gnathostomiasis [110].

When the clinical diagnosis has been confirmed, we can observe the causative pathogen in FFPE sections of the infected lesion. Even if the clinical diagnosis is totally unsettled, candidate causative pathogens can be visualized within the lesion. In some cases, the specificity of the patient’s serum is totally unknown. The size, shape, and localization pattern of labelling may suggest certain causative microbes when considered together with clinical features. Once the specificity of the patient’s serum is known, the serum can be used thereafter as a specific probe for immunostaining. To avoid biohazards, the serum of carriers of hepatitis B and C or acquired immunodeficiency syndrome virus must not be used.

As a representative example, images from a liver biopsy of a patient with visceral leishmaniasis (kala azar) are displayed in Figure 62. The patient was a middle-aged Japanese businessman who stayed long in Australia, Thailand, and India for business purposes. He complained of fever and malaise that was accompanied by anemia, thrombocytopenia, and liver dysfunction. The patient’s general condition was poor. A FFPE liver biopsy preparation showed multifocal small non-caseous granulomas. No positive findings were obtained by immunostaining with a routine panel of anti-microbial antibodies on hand. Chromogenic immunostaining using a 1:500 dilution of the patient’s own serum revealed red cell-sized and rounded positive signals in the cytoplasm of epithelioid cells in the granuloma or in Kupffer cells. The size, shape, and localization pattern strongly suggested the possibility of visceral leishmaniasis caused by *Leishmania donovani*, which is endemic in India. Visceral leishmaniasis, which has never been endemic in Japan, was serologically confirmed thereafter, and antimony-based therapy saved the patient’s life [105,109].

Another example is a case of *Balamuthia* encephalitis with a skin nodule. Here, the 1:500 dilution of serum from another patient who suffered lethal chronic meningoencephalitis caused by *Balamuthia mandrillaris* was used [105]. In biopsy samples of both the skin and brain, brown-colored amebic bodies (trophozoites and some cysts) are clearly identified, as shown in Figure 63.

The same strategy is applicable to autoimmune disorders. Serum from a patient with autoimmune (type A) gastritis can be used to detect proton pump (H^+^, K^+^-ATPase) molecules on intracytoplasmic secretory canaliculi of parietal cells in normal oxyntic gastric mucosa in FFPE sections (Figure 64) [111]. A 1:20 dilution was used in this situation. Autoantibodies against amphiphysin, a 128-kDa presynaptic protein, are seen in a patient with breast cancer who manifested progressive rigidity and painful spasms (an autoimmune encephalitis called Stiff-Man syndrome) [112]. Patient serum at the 1:20 dilution stained the plasma membrane of her own breast cancer cells. The serum of another patient with autoimmune limbic encephalitis of the non-paraneoplastic type [113] stained astroglial cells and glial fibers in FFPE basal ganglia of an autopsied normal brain. Immunostaining using a patient’s diluted serum may contribute to clarifying the pathogenesis of autoimmune encephalitis, as illustrated in Figure 65.

## 12. Concluding Remarks

Nowadays, it is clearly expected that chromogenic immunostaining will produce beautiful and specific images. Biomedical engineers (i.e., medical technicians) should know how to select appropriate markers and how to judge the results. Pathologists and investigators must understand the pitfalls and caveats of immunostaining procedures. To apply immunostaining to diagnostic pathology services, it is critically important for pathologists to know how to stain, how to select markers, and how to judge the results. The importance of close teamwork and cooperation between the biomedical engineers and pathologists cannot be understated.

## Figures and Tables

**Figure 1 cells-10-01501-f001:**
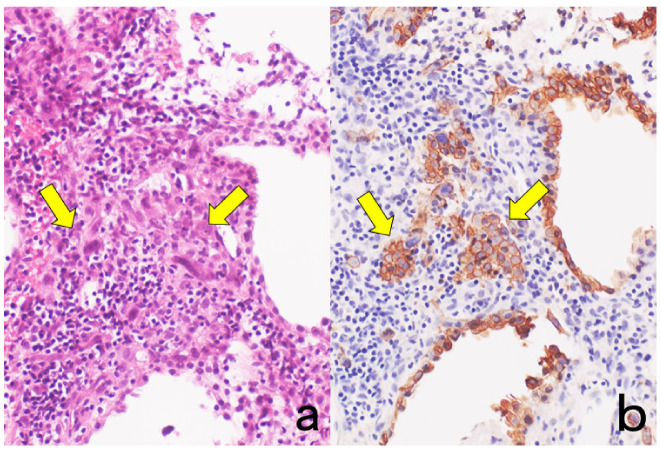
Cytokeratin for identifying intermediate trophoblasts in the placenta. (**a**): H&E, (**b**): cytokeratin immunostaining. Intermediate trophoblasts (arrows) are observed in the stroma of the placental tissue sampled by curettage. Cytokeratin immunoreactivity with a monoclonal antibody CAM5.2 clearly illustrates their distribution.

**Figure 2 cells-10-01501-f002:**
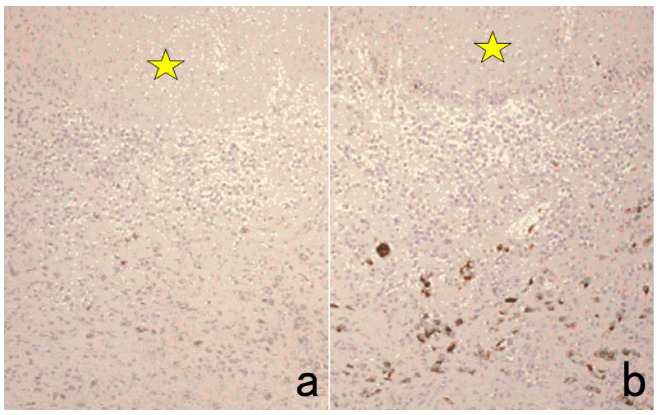
p53 immunostaining without HIER in esophageal squamous cell carcinoma. (**a**): Polymer method (EnVision) and (**b**): catalyzed signal amplification (CSA)-II method. Submucosally invading cancer cells express p53 in the nuclei. The immunoreactivity is significantly enhanced by the CSA-II method using FITC tyramide as an amplifier. Stars indicate non-cancerous esophageal squamous mucosa.

**Figure 3 cells-10-01501-f003:**
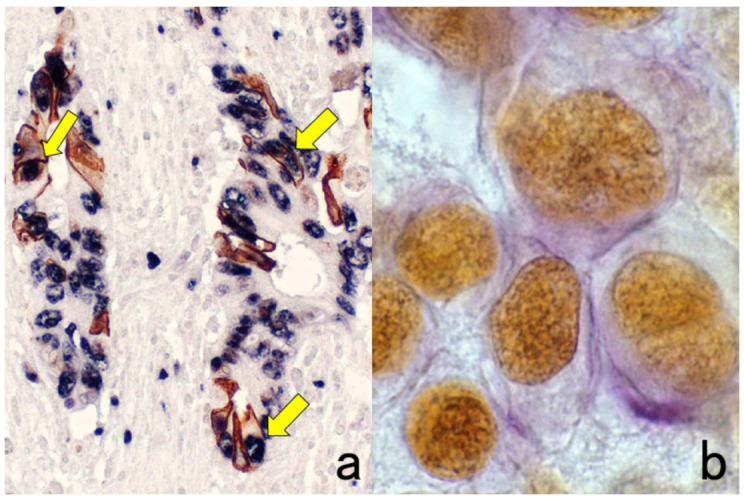
Chromogenic double immunostaining. (**a**): Ki-67 (nuclei of proliferative cells are blue) and cleaved cytokeratin 18 (cytoplasm of apoptotic columnar cells is brown) in FFPE colonic adenocarcinoma after neoadjuvant chemotherapy. In colon cancer, some proliferative cells are apoptotic, as the arrows indicate. Cleaved cytokeratin 18 functions as an excellent marker for apoptotic columnar cells. (**b**): HER2-positive plasma membranes in blue and p53-positive nuclei in brown in an ethanol-fixed aspiration cytology specimen of breast cancer. In this high-grade breast cancer, the atypical ductal cells overexpress both HER2 and p53.

**Figure 4 cells-10-01501-f004:**
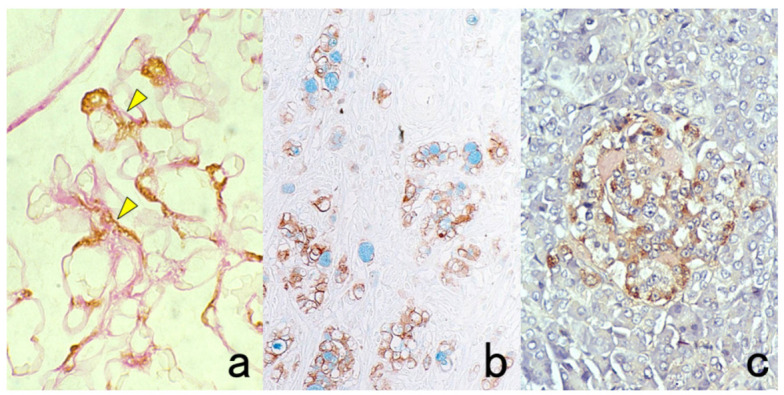
Double labeling of immuno- and conventional staining. (**a**): IgA and PAS in renal biopsy of IgA nephropathy. The PAS-reactive basement membrane surrounds mesangial deposition of IgA visualized after prolonged trypsin digestion (arrowheads). (**b**): Carcinoembryonic antigen (CEA) and alcian blue in gastric signet ring cell carcinoma. CEA expression of mucin-containing cancer cells appears weaker. (**c**): Insulin and Congo red in a pancreatic islet in type 2 diabetes mellitus. Orange-colored amyloid deposition is seen among the insulin-positive β-cells.

**Figure 5 cells-10-01501-f005:**
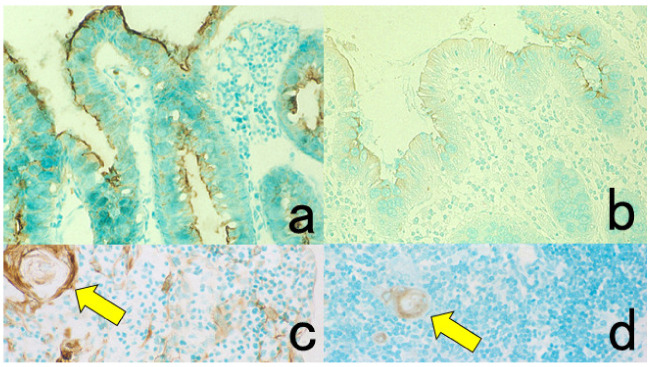
Antigenic deterioration in FFPE sections. Top panels (**a**,**b**): CEA in normal gastric mucosa. Bottom panels (**c**,**d**): Cytokeratin in normal infantile thymus. Left (**a**,**c**): paraformaldehyde-fixed frozen sections. Right (**b**,**d**): FFPE sections. Antigenicities of CEA and cytokeratin detected by polyclonal antibodies are evidently weakened after the FFPE process. Arrows indicate Hassall’s corpuscles in the thymic medulla. The nuclei are counterstained with methyl green.

**Figure 6 cells-10-01501-f006:**
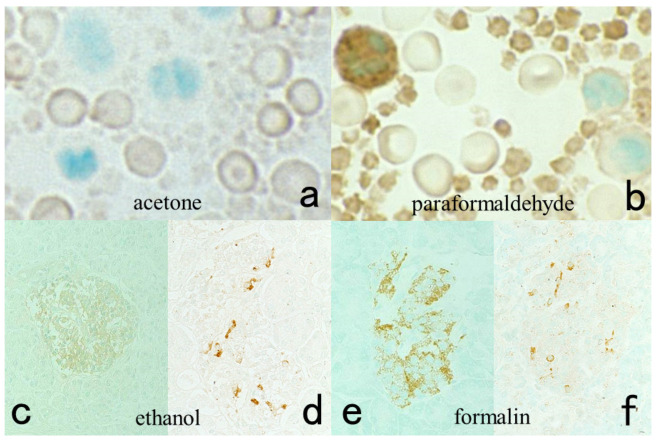
Antigenic loss by fixation in organic solvent. Top panels (**a**,**b**): Immunostaining for platelet factor 4 in a smeared buffy coat of the peripheral blood. Bottom panels (**c**–**f**): Immunostaining for insulin (**c**,**e**) and somatostatin (**d**,**f**) in paraffin-embedded pancreas. Left (**a**,**c**,**d**): Fixation in acetone or ethanol. Right (**b**,**e**,**f**): Fixation in 4% paraformaldehyde or in formalin. Platelet factor 4 and insulin show false negative findings when fixed in the organic solvent. Somatostatin is resistant to ethanol fixation to be clearly immunolocalized. The nuclei are counterstained with methyl green.

**Figure 7 cells-10-01501-f007:**
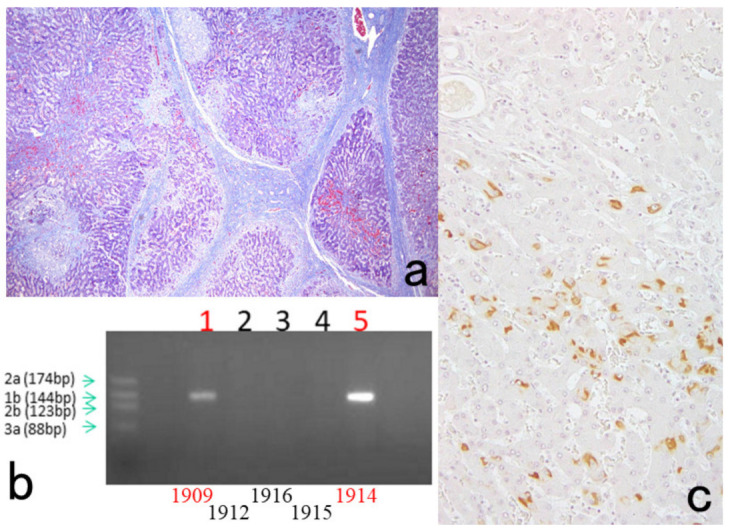
Demonstration of HBs antigen and HCV genome in archival specimens of liver cirrhosis fixed for up to 110 years. (**a**): Azan stain, (**b**): Nested reverse transcription-PCR for detecting the C-region of the HCV genome, (**c**); Immunostaining for HBs antigen. HCV-RNA subtype 1b was detected with nested RT-PCR (lane 1), and the same cirrhotic specimen in 1909 was immunoreactive for HBs antigen. Double infection of HBV and HCV was proven. Another specimen in 1914 (lane 5) was also double-positive for HCV, 1b-type, and HBs antigen.

**Figure 8 cells-10-01501-f008:**
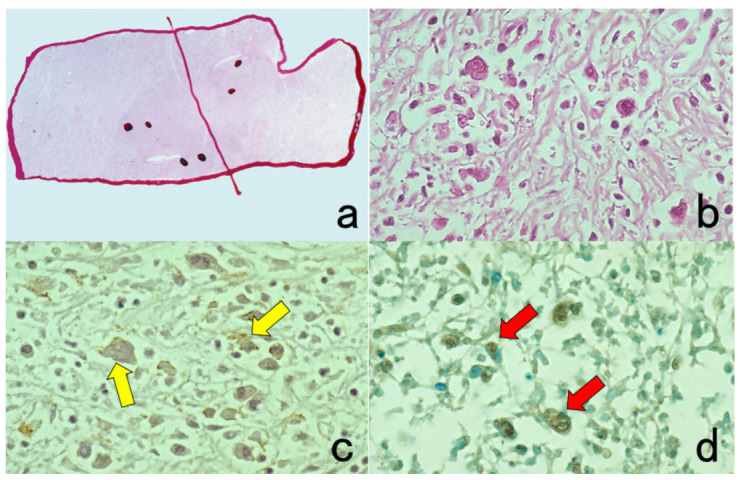
Demonstration of CD15 and EBER in Hodgkin’s lymphoma in a male patient in his 50s autopsied by Dr. Thomas Hodgkin himself 170 years ago. Top panels: H&E. Only one unstained slide of the liver available was stained with H&E. After photographing, the section was transferred to two silane-coated glass slides (**a**). Infiltration of Reed-Sternberg cells or Hodgkin’s cells is microscopically recognized (**b**). Bottom panels: Immunostaining for CD15 (**c**) and in situ hybridization for EBER (**d**) gave positive signals on the plasma membrane and in the nucleus, respectively, as indicated by yellow and red arrows.

**Figure 9 cells-10-01501-f009:**
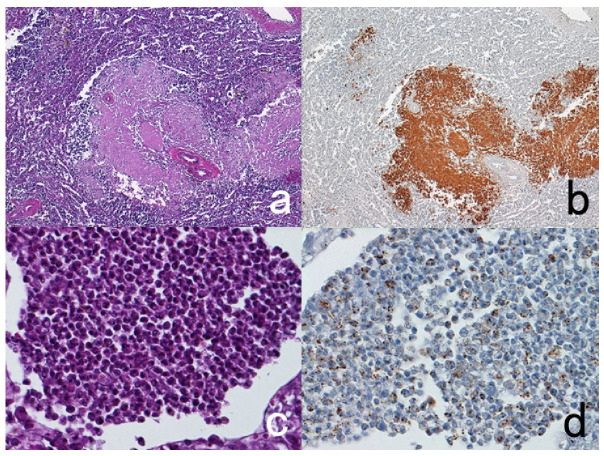
Demonstration of amyloid A in secondary amyloidosis and pneumococcal antigens in lobar pneumonia in archival material. Top panels (**a**,**b**): Splenic white pulp with amyloid A deposition. Bottom panels (**c**,**d**): pneumococcal antigens in lobar pneumonia showing diffuse neutrophilic infiltration. Left (**a**,**c**): H&E and Right (**b**,**d**): chromogenic immunostaining. The specimens were fixed in formalin for 70 years. Deposition of amyloid A protein is evident in the splenic white pulp (no pretreatment needed). Pneumococci phagocytized by neutrophils are visualized immunohistochemically after HIER.

**Figure 10 cells-10-01501-f010:**
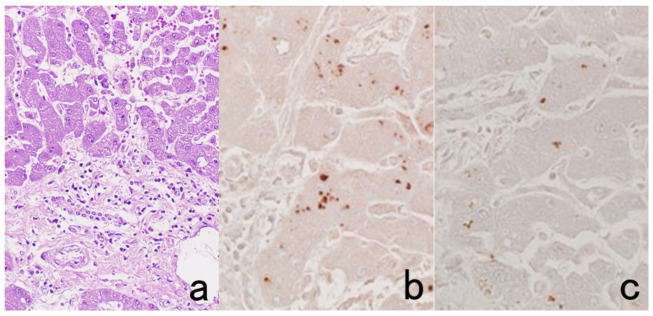
Demonstration of HCV antigens in an archival liver specimen of chronic hepatitis after HIER. (**a**): H&E, (**b**): monoclonal antibody against the NS3 region of HCV and (**c**): monoclonal antibody against the E1 region. Dot-like immunoreactivity of HCV antigens is seen in the cytoplasm of hepatocytes. The specimen was kept in formalin for 40 years.

**Figure 11 cells-10-01501-f011:**
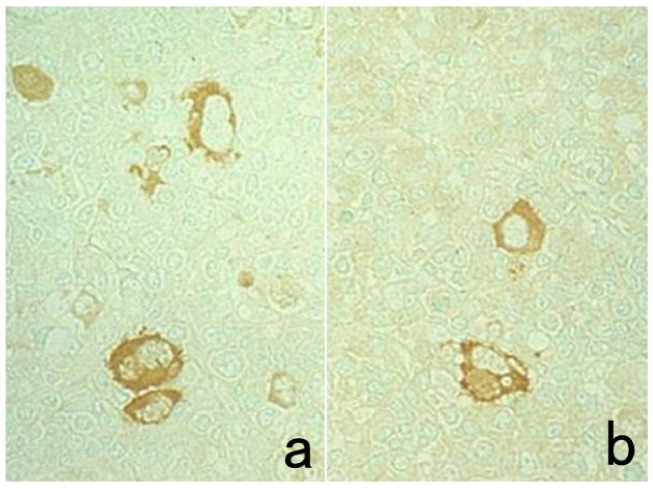
Diffusion artifact (I). Hodgkin’s lymphoma. The cytoplasm of Reed-Sternberg cells or Hodgkin’s cells is polyclonally positive for both kappa (**a**) and lambda (**b**) chains. The nuclei are negative. This represents an example of diffusion artifact of plasma proteins in FFPE preparations. The nuclei are counterstained with methyl green.

**Figure 12 cells-10-01501-f012:**
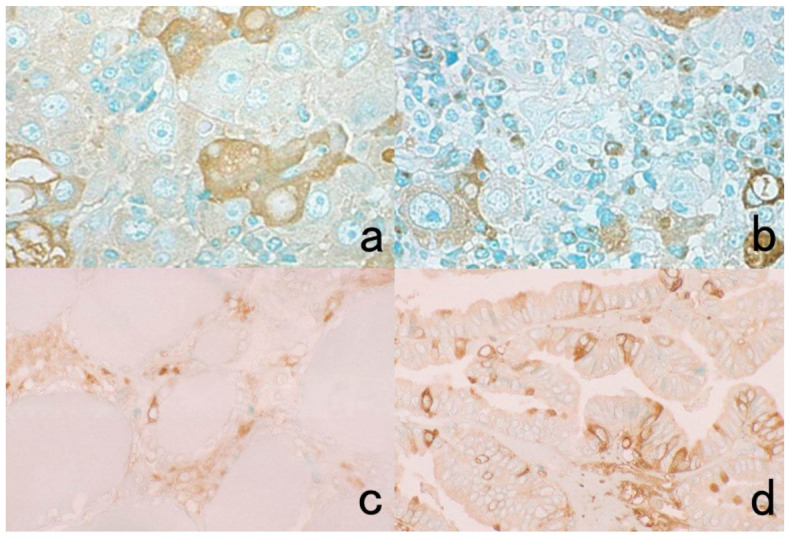
Diffusion artifact (II). Top panels: Albumin (**a**) and IgG (**b**) immunoreactivities in hepatocellular carcinoma metastatic to the hepatic hilar lymph node. Bottom panels: IgA immunostaining in normal (**c**) and neoplastic (**d**) thyroid. Hepatocellular carcinoma cells, normal thyroid follicular cells, and papillary thyroid carcinoma cells reveal diffuse cytoplasmic staining for the plasma proteins (albumin, IgG, and IgA) in some cells. The nuclei remain negative. The Golgi area of plasma cells is immunoreactive for IgG in panel (**b**). The nuclei are counterstained with methyl green.

**Figure 13 cells-10-01501-f013:**
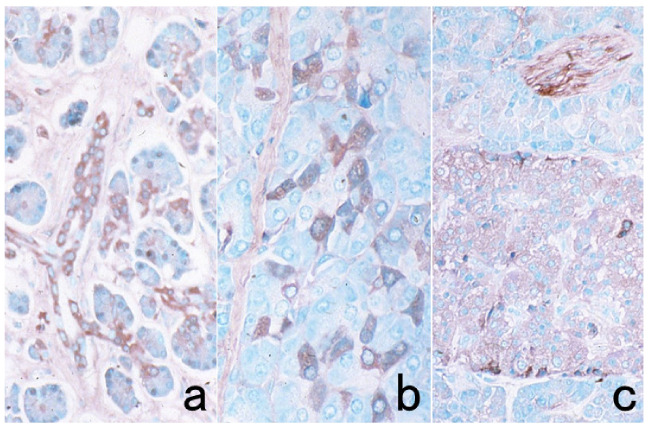
Diffusion artifact (III). Albumin immunoreactivity in FFPE autopsied pancreas. Ductular cells (**a**), acinar cells (**b**), and islet cells and peripheral nerve (**c**) show diffuse cytoplasmic staining. The signals are randomly observed in different cells in different parts of the pancreas. The nuclei, counterstained with methyl green, remain negative.

**Figure 14 cells-10-01501-f014:**
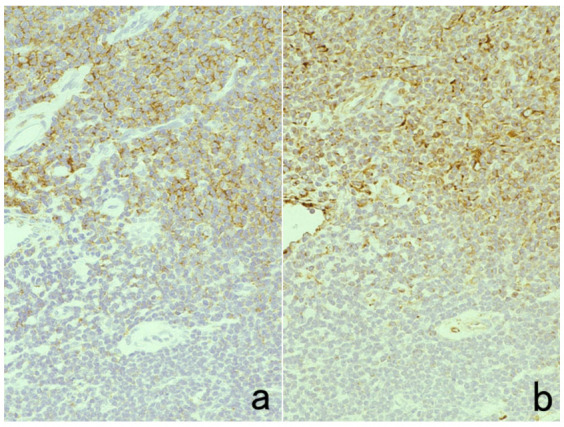
Vimentin internal control. Diffuse large B-cell lymphoma. (**a**): Vimentin and (**b**): CD20. The poorly fixed central part (representing the lower half of the panels) is false negative for both vimentin and CD20. The expression of lymphocyte surface markers should be judged based on the positivity in the appropriately fixed peripheral area (the upper half of the panels).

**Figure 15 cells-10-01501-f015:**
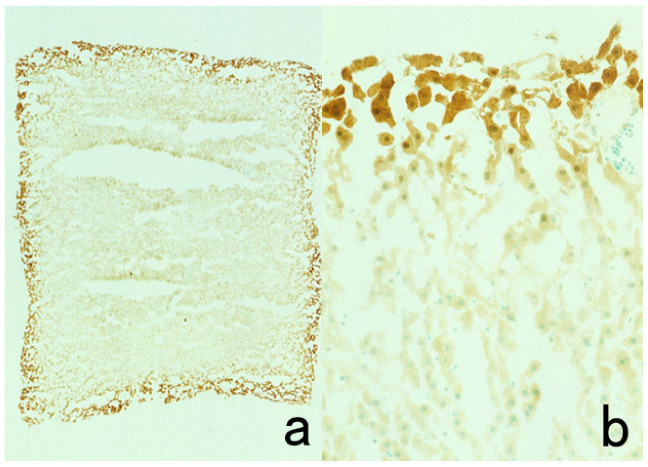
Glutathione peroxidase immunostaining using a frozen section prepared from 4% paraformaldehyde-fixed rat liver tissue (**a**): low power, (**b**): high power. Positive immunoreactivity in the cytoplasm and nucleus of the hepatocytes is seen just at the outermost part of the section.

**Figure 16 cells-10-01501-f016:**
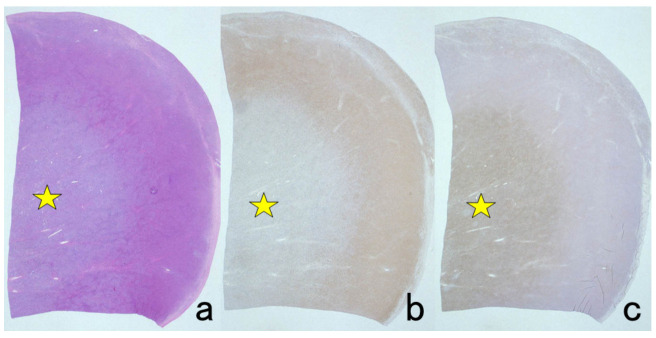
Paradoxical localization of CD45 and vimentin in FFPE malignant lymphoma tissue. (**a**): H&E, (**b**): vimentin and (**c**): CD45 (leukocyte common antigen: LCA). Vimentin is positive in the peripheral zone but negative in the poorly fixed central area (stars). A reversed pattern is recognizable for CD45 localization. The over-fixed outer zone is negative for CD45.

**Figure 17 cells-10-01501-f017:**
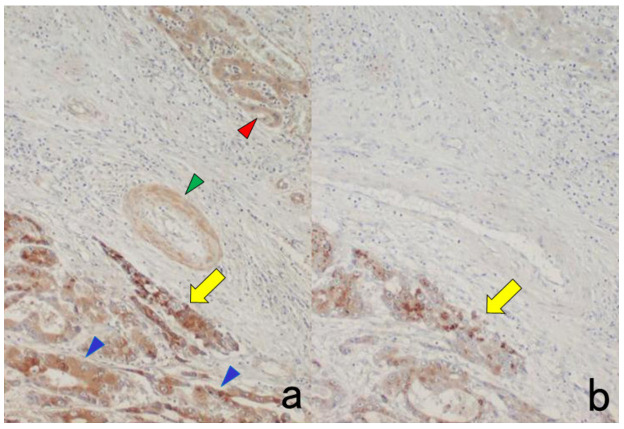
The need for appropriate dilution of the primary rabbit antiserum. Alpha-fetoprotein (AFP) immunostaining (using the polymer method) for hepatocellular carcinoma. (**a**): 1:400 dilution, (**b**): 1:3200 dilution. With 1:400 diluted antiserum, non-neoplastic hepatocytes (blue arrowheads), bile ducts (red arrowhead) and arterial wall (green arrowhead) reveal nonspecific staining. At an appropriate dilution (1:3200), positive signals of AFP are seen only in the hepatoma cells (yellow arrows).

**Figure 18 cells-10-01501-f018:**
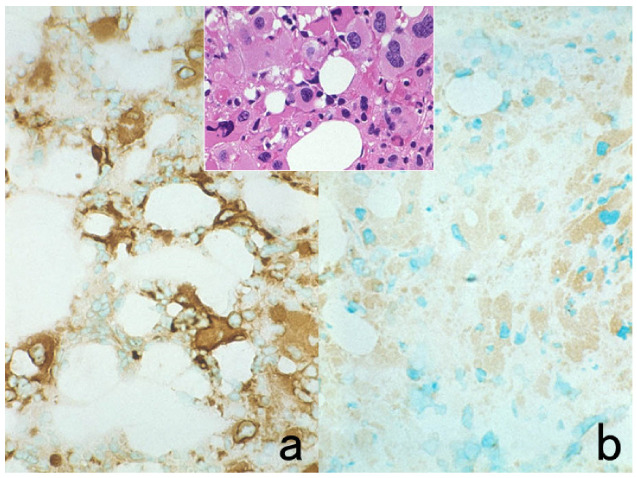
Antigenic loss of surface markers by endogenous peroxidase blockage in fresh-frozen sections. Aspirated bone marrow clot of essential thrombocythemia. Immunostaining for glycoprotein IIb/IIIa (CD41/CD61), a megakaryocytic marker. (**a**): Without endogenous peroxidase blockage and (**b**): with endogenous peroxidase blockage, inset: H&E. Endogenous peroxidase blockage in 0.3% hydrogen peroxide in methanol significantly hampers the immunoreactivity. The nuclei are counterstained with methyl green.

**Figure 19 cells-10-01501-f019:**
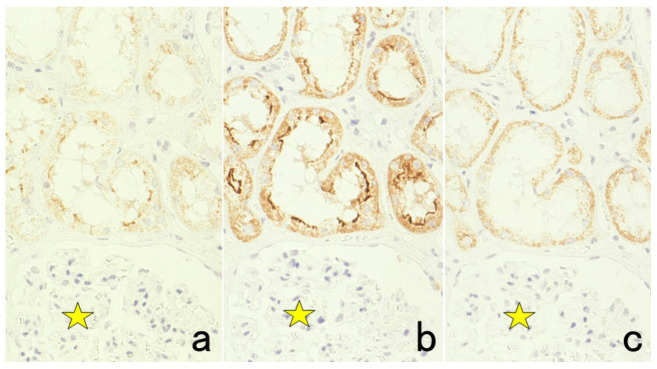
Retrieval of endogenous biotin activity by heating pretreatment (LSAB immunostaining for villin in the normal kidney with pressure pan heating in 10 mM citrate buffer, pH 6 (**a**), 10 mM citrate buffer, pH 7 (**b**) and 10 mM citrate buffer, pH 7, without primary antibody (**c**). The brush border of proximal renal tubules is immunoreactive for villin. Hydrated heating in citrate buffer, pH 7, enhances retrieval of endogenous biotin activity, as typically represented in panel (**c**). Coarse granular (mitochondrial) positivity is observed in the cytoplasm. Stars indicate a glomerulus.

**Figure 20 cells-10-01501-f020:**
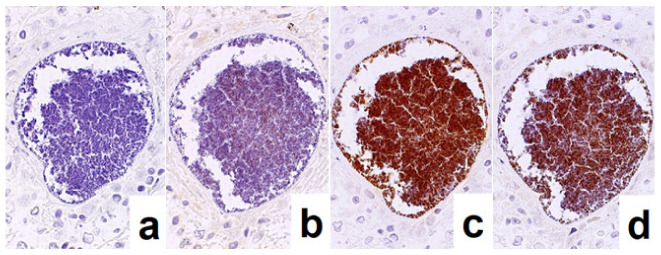
Retrieval of protein A activity on methicillin-resistant *Staphylococcus aureus* (MRSA) by heating pretreatment in 1 mM EDTA, pH 8. MRSA infection in the lung. (**a**): Without primary antibody (negative control), (**b**): incubation with rabbit antibody, F(ab’)2 fragments, against mouse immunoglobulins, (**c**): incubation with rabbit antibody, whole IgG, against mouse immunoglobulins and (**d**): incubation with mouse monoclonal antibody against *Pseudomonas aeruginosa* as an indifferent antibody. It is evident that the MRSA colony in the lung binds to rabbit and mouse IgG, as shown in panels **c** and **d**, respectively. The F(ab’)2 fragments of IgG lack this binding (**b**), so the nonspecific binding is caused through the Fc portion of IgG.

**Figure 21 cells-10-01501-f021:**
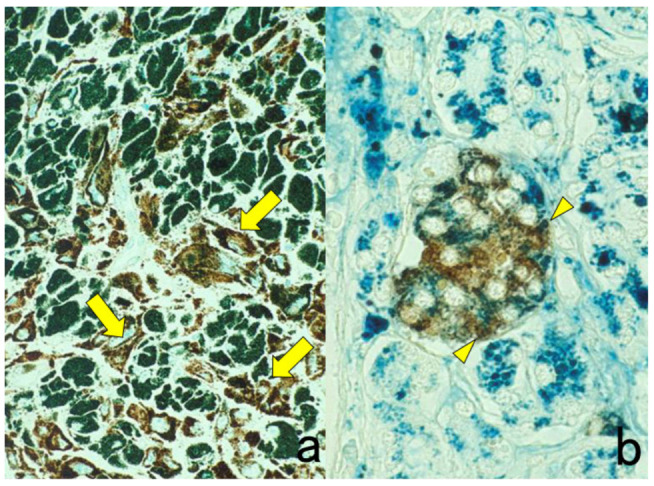
Metachromasia of melanin by methyl green counterstaining in heavily pigmented nasal malignant melanoma (**a**) immunostaining for HMB45 and Berlin blue counterstaining for hemosiderin in pancreatic hemochromatosis (**b**) immunostaining for insulin. Brown-colored pigments such as melanin and hemosiderin hamper the judgment of DAB coloration. Counterstaining with methyl green and Berlin blue is quite effective for distinguishing the DAB color from the endogenous pigments. The rich distribution of HMB45-negative metachromatic melanophages is noted in panel (**a**). Arrows indicate HMB45-positive melanoma cells. In panel (**b**), hemosiderin deposition is observed in insulin-positive β-cells of the pancreatic islet (arrowheads).

**Figure 22 cells-10-01501-f022:**
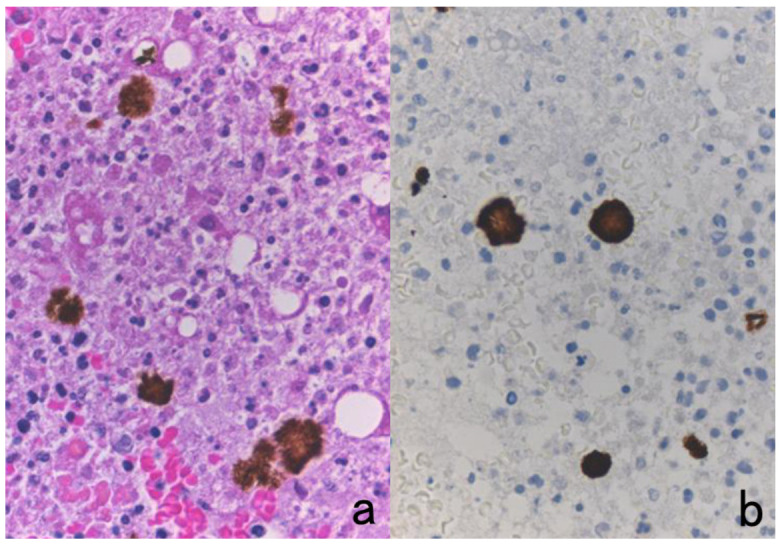
Concentrated bile pigments in a cell block preparation of liver abscess clinically suspected of being amebic infection. (**a**): H&E and (**b**): immunostaining using a monoclonal antibody EHK153 against *Entamoeba histolytica*. Brown-colored structures of concentrated bile pigments somewhat resemble strongly immunoreactive amebic trophozoites. The entamoeba antigen must be judged as negative in this case. Careful comparison with H&E histology is needed to avoid falsely judging these as positive immunohistochemical signals.

**Figure 23 cells-10-01501-f023:**
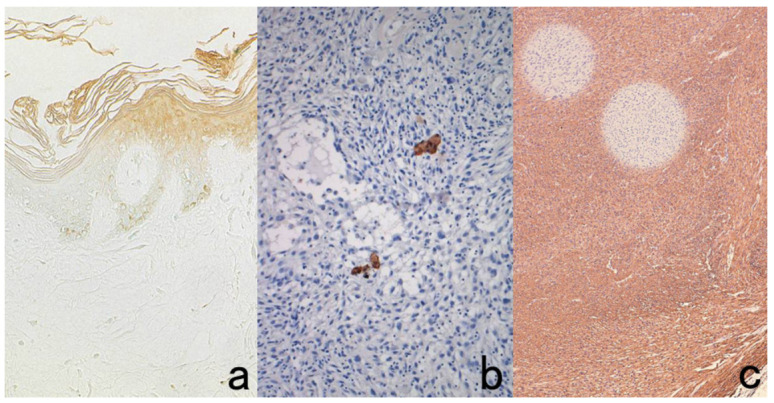
Technical artifacts. Effects of drying. (**a**): surfactant apoprotein immunostaining for the skins, contamination of cytokeratin-positive scales. (**b**): cytokeratin immunostaining for spindle cell sarcoma and insufficient deparaffinization (**c**): smooth muscle actin immunostaining for leiomyosarcoma. Drying of sections during antibody incubation causes nonspecific binding in the dried area (**a**). Contaminated cytokeratin-positive scales on the section should be distinguished from true positivity (**b**). False-negative rounded areas result from insufficient deparaffinization. (**c**).

**Figure 24 cells-10-01501-f024:**
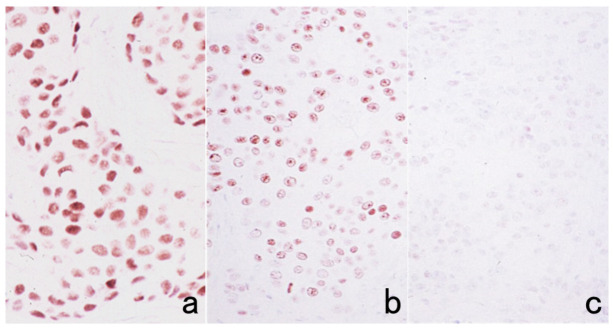
Effect of prolonged storage of unstained sections (HIER-assisted ER immunostaining for breast cancer). Immunostaining just after cutting (**a**) and immunostaining after prolonged storage for 6 months in a −20 °C freezer (**b**) or at room temperature. (**c**). The prolonged storage at room temperature severely deteriorates the immunoreactivity of nuclear antigens such as ER. Storage in a freezer is effective for avoiding such false negativity.

**Figure 25 cells-10-01501-f025:**
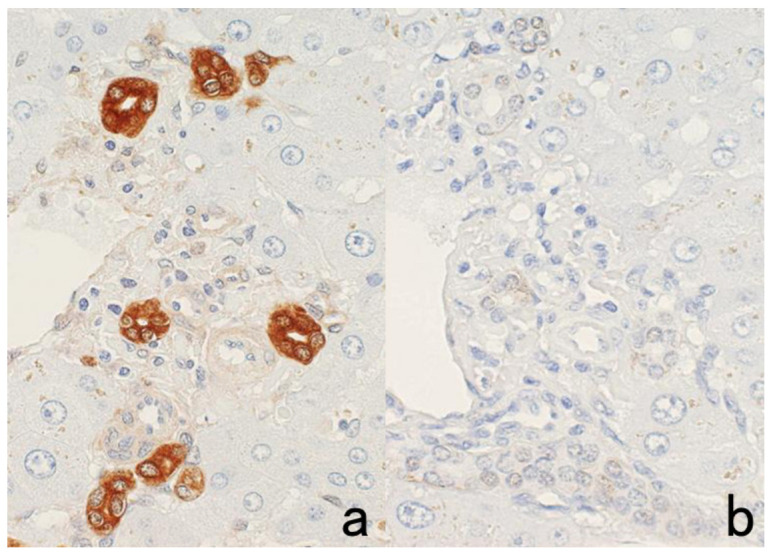
Antigenic loss by stretching sections on a hot plate. Immunostaining for glutathione-S-transferase (GST)-π in the normal liver after stretching on a hot plate at 50 °C for 10 s (**a**) and at 70 °C for 3 s (**b**). The cytoplasmic immunoreactivity of GST-π in the bile duct cells is completely lost by high-temperature stretching at 70 °C for even a very short period.

**Figure 26 cells-10-01501-f026:**
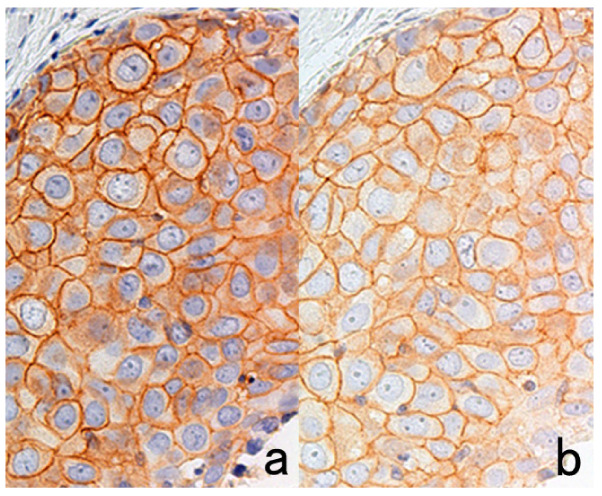
Weakened HER2 immunoreactivity in breast cancer cells by prolonged drying of unstained sections. Drying in an incubator at 40 °C for 1 h (**a**) and for 3 days (**b**). The intensity of HER2 immunostaining is considerably weakened after prolonged drying in an incubator over a weekend.

**Figure 27 cells-10-01501-f027:**
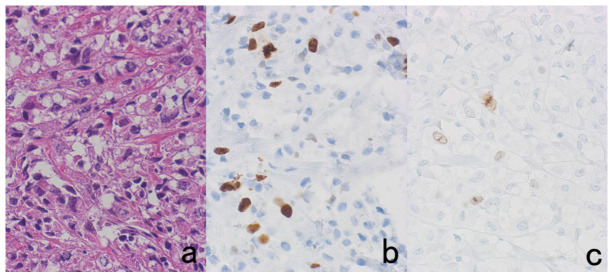
Loss of Ki-67 immunoreactivity in breast cancer (with foci of dystrophic calcification) by surface decalcification with 5% trichloroacetic acid for 1 h. (**a**): H&E, (**b**): Immunostaining without surface decalcification and (**c**): Immunostaining after surface decalcification with trichloroacetic acid. Surface decalcification significantly deteriorates Ki-67 immunoreactivity.

**Figure 28 cells-10-01501-f028:**
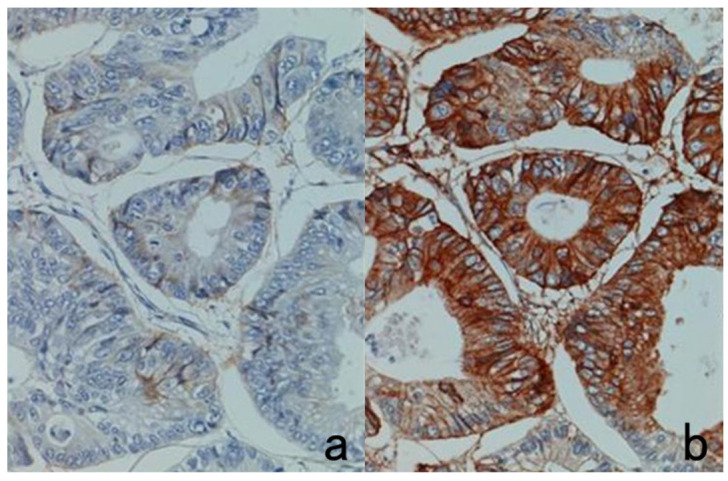
PharmDx^TM^ for EGFR immunostaining in colonic adenocarcinoma. The kit supplied from Agilent/Dako provides faint immunoreactivity (**a**), which may affect appropriate judgment. However, when replacing EnVision (as the secondary reagent) with CSA-II (available from the same company), very strong EGFR immunoreactivity can be observed (**b**).

**Figure 29 cells-10-01501-f029:**
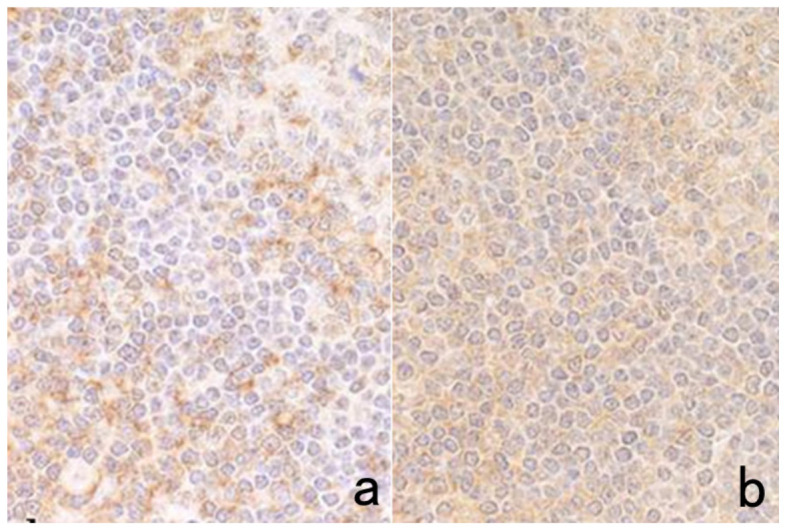
Paradoxical weakening of the antigenicity in CSA-II-mediated chromogenic immunostaining when the primary antibody concentration is high. CD4 immunostaining of the pharyngeal tonsil after HIER using CD4 monoclonal antibody diluted at 1:500 (**a**) and 1:50 (**b**). Equivocal CD4 staining on the plasma membrane with a high background occurs when the primary antibody concentration is 10 times higher. The background staining was caused by artificial diffusion of the membrane antigen into the cytoplasm.

**Figure 30 cells-10-01501-f030:**
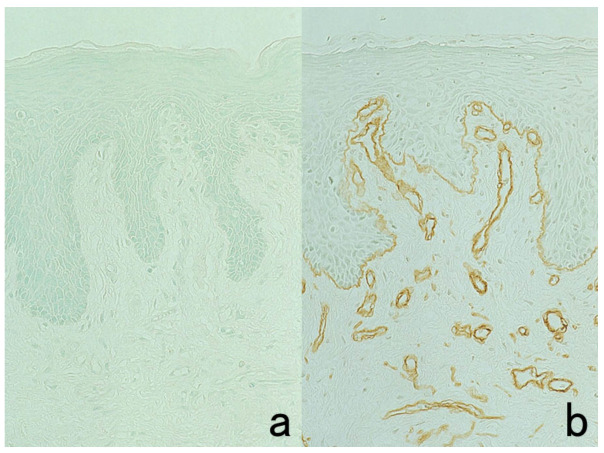
Effect of trypsin digestion in immunostaining for type 4 collagen in the normal skin. (**a**): Immunostaining without treatment and (**b**): after 0.1% trypsin digestion at 37 °C for 30 min. Type 4 collagen immunoreactivity in the basement membrane is clearly visualized after trypsin digestion. The nuclei are counterstained with methyl green.

**Figure 31 cells-10-01501-f031:**
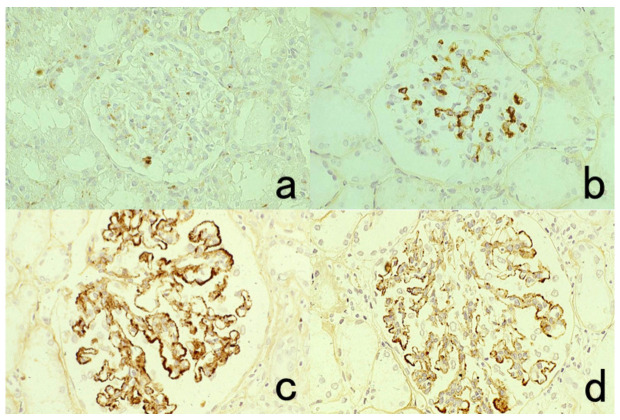
Detection of glomerular deposits of immunoglobulins and complement in FFPE renal biopsy sections with the aid of prolonged 0.1% trypsin digestion for 120 min. (**a**,**b**): IgA nephropathy (IgA staining, (**a**): without digestion, (**b**): after digestion). Mesangial deposition of IgA is evident in (**b**). (**c**,**d**): Membranous nephropathy after 0.1% trypsin digestion for 120 min ((**c**): IgG, (**d**): C3). Granular deposition of IgG and C3 is observed along the capillary loop, a diagnostic feature of membranous nephropathy.

**Figure 32 cells-10-01501-f032:**
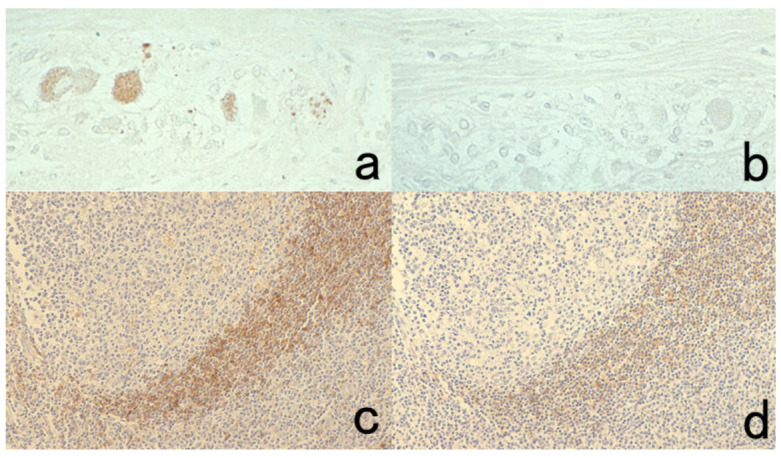
Deterioration of antigenicity by protease pretreatment. Top panels (**a**,**b**): substance P in the myenteric plexus of the small bowel. Bottom panels (**c**,**d**): IgD in the mantle zone of the pharyngeal tonsil. Left (**a**,**c**): without treatment and Right (**b**,**d**): after trypsin digestion for 15 min. Immunoreactivities of substance P and IgD are lost or markedly decreased by the enzymatic pretreatment.

**Figure 33 cells-10-01501-f033:**
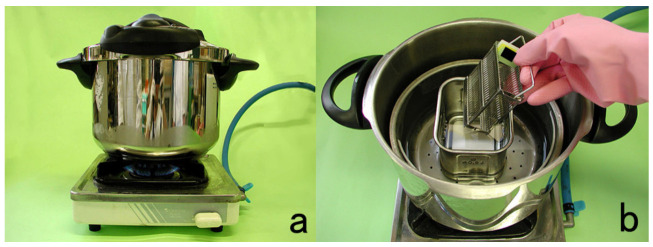
Recommendation of a pot typically used in French cuisine for use as a pressure pan (**a**,**b**). The pressure pan made by T-Fal (Rumilly, Haute-Savoie, France) is easy to handle and use for HIER in a diagnostic pathology laboratory.

**Figure 34 cells-10-01501-f034:**
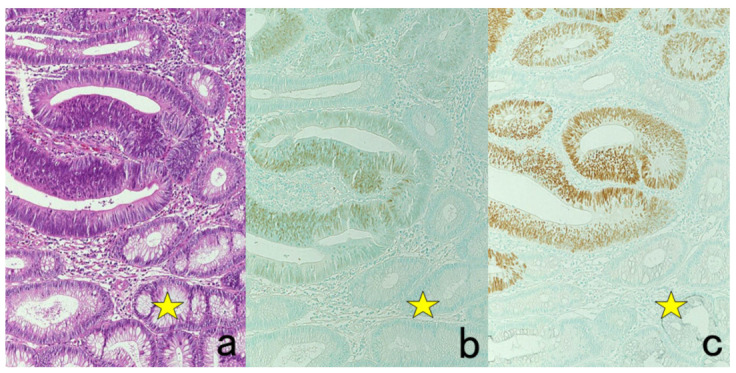
Application of HIER (I). p53 expression in cancer in adenoma of the colon. (**a**): H&E, (**b**): immunostaining without HIER and (**c**): immunostaining after HIER with autoclaving at 121 °C for 10 min in 10 mM citrate buffer, pH 6.0. The nuclei of adenocarcinoma cells express p53, whereas the surrounding adenoma cells (stars) remain negative. The nuclei are counterstained with methyl green.

**Figure 35 cells-10-01501-f035:**
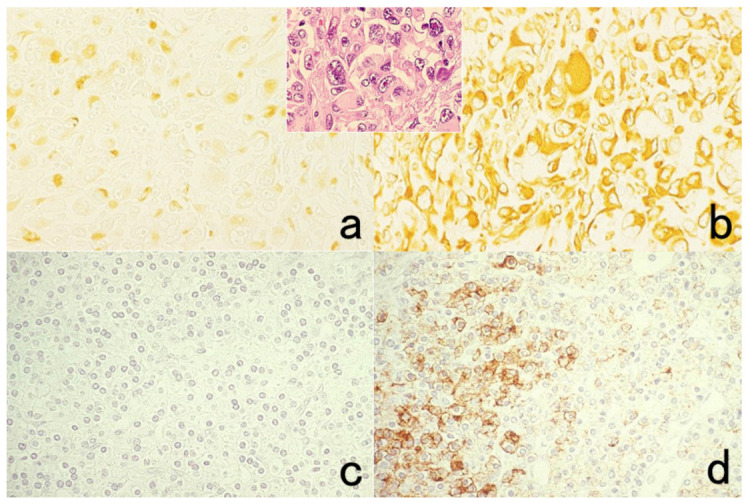
Application of HIER (II). Top panels (**a**,**b**): Vimentin in lung metastasis of anaplastic thyroid carcinoma (inset: H&E). Bottom panels (**c**,**d**): Parathyroid hormone in normal parathyroid gland. Left (**a**,**c**): Without HIER treatment and Right (**b**,**d**): after HIER with autoclaving at 121 °C for 10 min in 10 mM citrate buffer, pH 6. Immunoreactivities of vimentin and parathyroid hormone are significantly retrieved by the hydrated heating pretreatment.

**Figure 36 cells-10-01501-f036:**
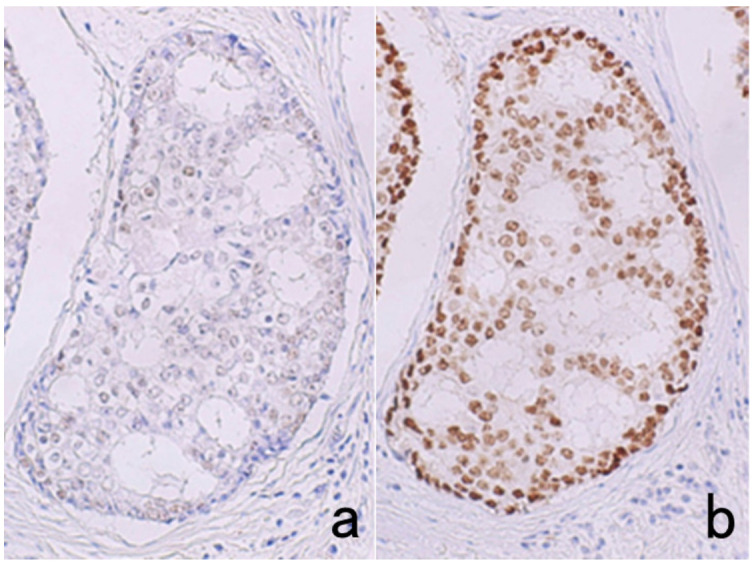
Selection of the aqueous solution for appropriate HIER. ER immunostaining for breast cancer after pressure pan heating at 121 °C for 10 min. (**a**): 10 mM citrate buffer, pH 6.0, (**b**): 1 mM EDTA, pH 8.0. ER in cribriform ductal carcinoma in situ is effectively retrieved by HIER in 1 mM EDTA solution, pH 8.0.

**Figure 37 cells-10-01501-f037:**
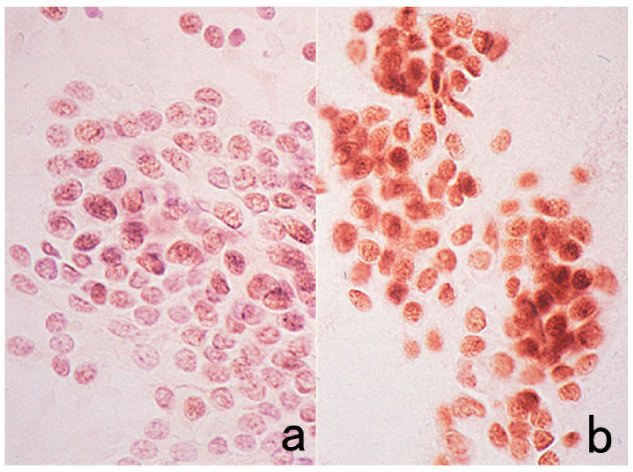
ER immunostaining for an aspiration cytology preparation after ethanol fixation. (**a**): Without pretreatment and (**b**): HIER with pressure pan heating at 121 °C for 10 min in 10 mM citrate buffer, pH 7.0. HIER is effective for ER immunolocalization even in cytology specimens.

**Figure 38 cells-10-01501-f038:**
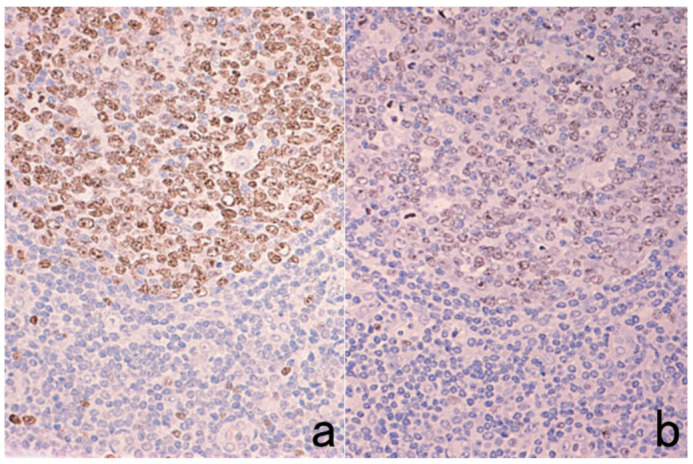
Weakening of Ki-67 immunoreactivity by overnight antibody incubation after HIER with pressure pan heating at 121 °C for 10 min in 10 mM citrate buffer, pH 6.0. (**a**): One-hour antibody incubation and (**b**): overnight incubation. Ki-67 labeling in the germinal center lymphocytes of the pharyngeal tonsil, clearly seen in the 1-h antibody incubation, is significantly weakened after overnight incubation.

**Figure 39 cells-10-01501-f039:**
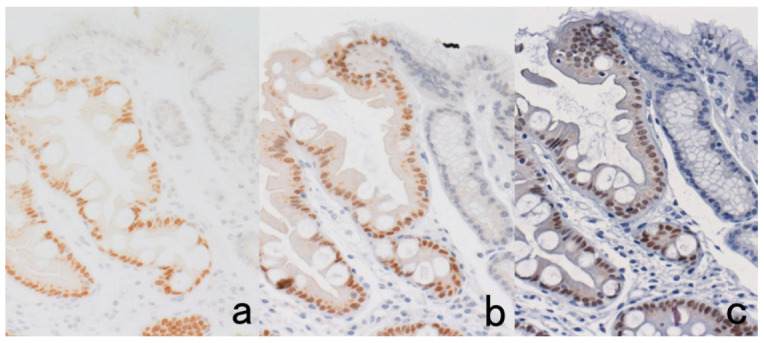
Poor nuclear stainability by hematoxylin after EDTA-mediated HIER. CDX2 (a nuclear marker for intestinal-type epithelial cells) immunoreactivity in intestinal metaplasia of the stomach. Mayer’s hematoxylin staining for 10 s (**a**), 1 min (**b**), and 3 min (**c**). Prolonged nuclear staining is necessary after HIER in 1 mM EDTA solution, pH 8.0.

**Figure 40 cells-10-01501-f040:**
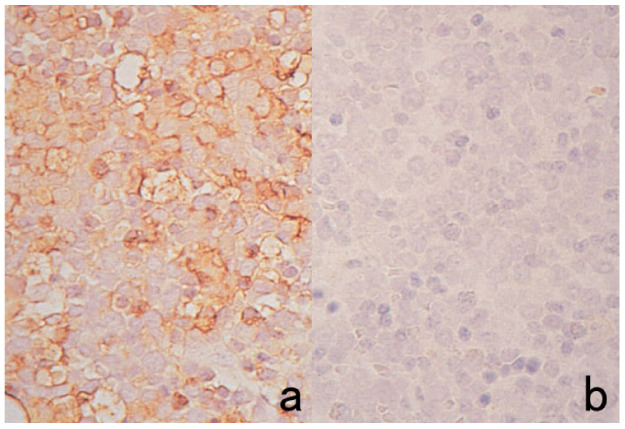
Antigenic loss by HIER in 10 mM citrate buffer, pH 6.0. BM-1 (a myeloid marker) in FFPE bone marrow aspiration. (**a**): Without HIER and (**b**): after HIER with pressure pan heating at 121 °C for 10 min. BM-1 immunoreactivity is quite susceptible to the heating pretreatment.

**Figure 41 cells-10-01501-f041:**
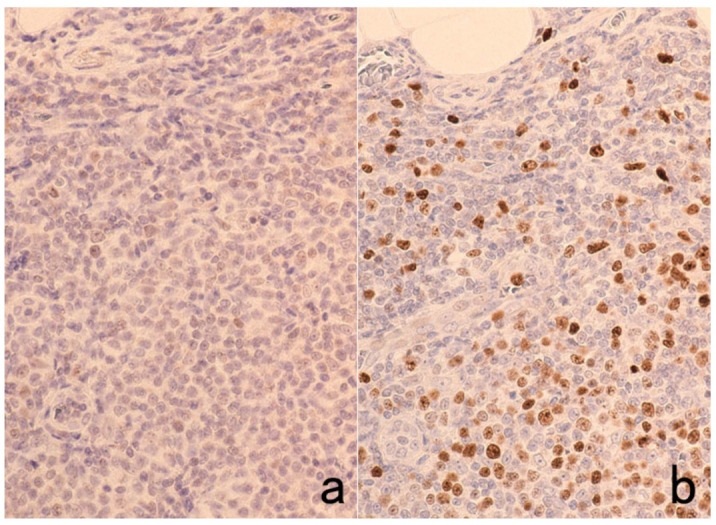
How to cool down after HIER. (**a**): Rapid cooling by soaking in tap water and (**b**): gradual cooling by leaving the slide container in the heating solution after it reaches room temperature. Ki-67 antigenicity in reactive lymphadenopathy becomes undetectable after rapid cooling. Sections after gradual cooling reveal distinct nuclear labeling.

**Figure 42 cells-10-01501-f042:**
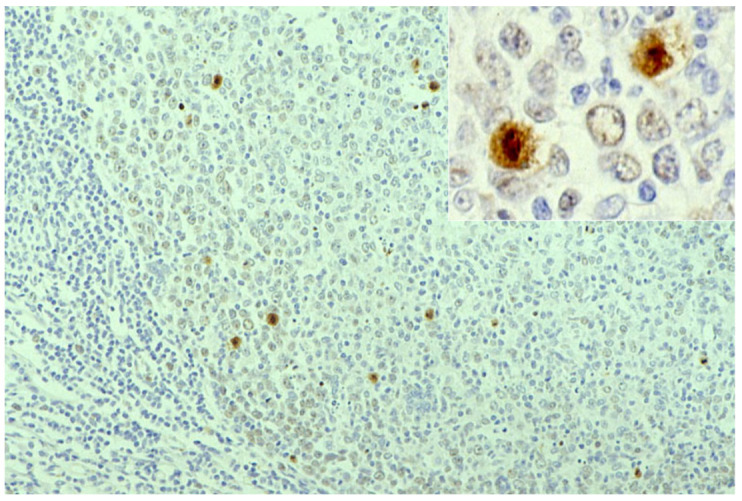
Labeling of mitotic cells by Ki-67 immunostaining in the pharyngeal tonsil (inset: high-power view). A high-molecular-weight polymer reagent in the EnVision system provokes a false-negative result for Ki-67 nuclear labeling of proliferative cells. Only mitotic cells are labeled. It seems that the huge molecules of the secondary reagent cannot reach the non-mitotic proliferative cells but do reach the mitotic cells lacking the nuclear membrane.

**Figure 43 cells-10-01501-f043:**
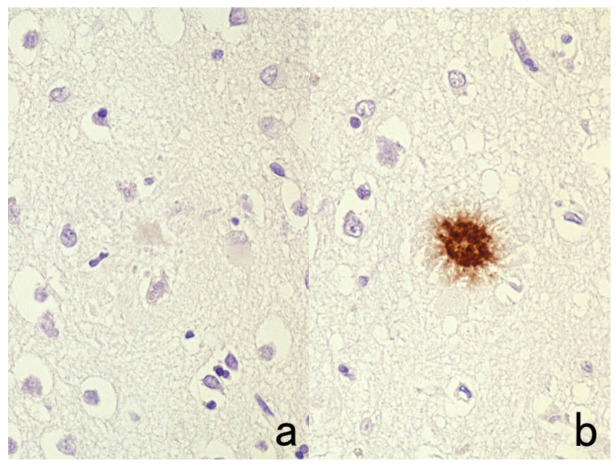
Retrieval of β-amyloid immunoreactivity in the brain by formic acid pretreatment. (**a**): Without treatment and (**b**): with 100% formic acid treatment for 5 min. Brief soaking in formic acid retrieves β-amyloid immunoreactivity of senile plaques in Alzheimer’s disease. A polyclonal antibody available from Agilent Co., Santa Clara, CA, USA, was used.

**Figure 44 cells-10-01501-f044:**
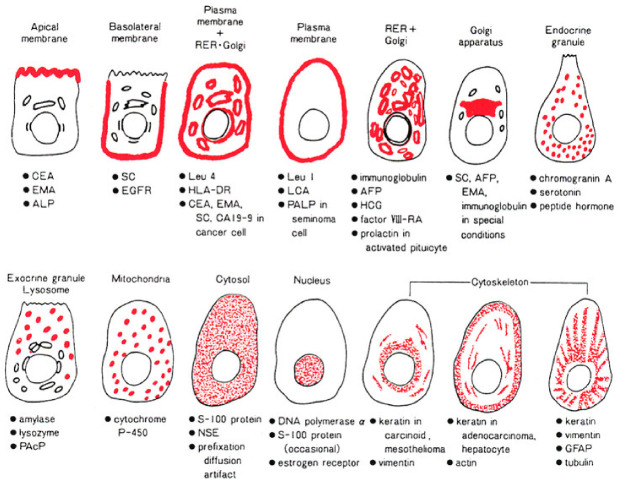
Schematic illustration of intracellular localization of antigens. The specificity of immunostaining can be judged by the intracellular localization pattern of the antigenic substances as either membranous, diffuse cytoplasmic, granular/vesicular cytoplasmic or nuclear.

**Figure 45 cells-10-01501-f045:**
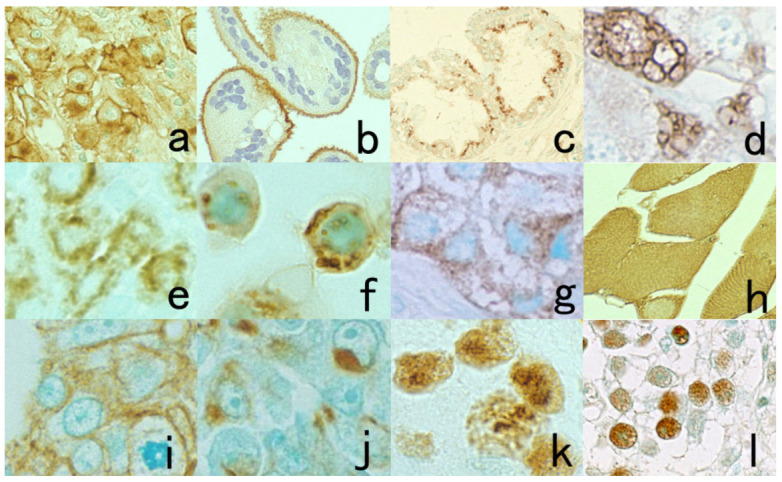
Representative immunohistochemical markers showing various intracellular localization patterns. (**a**): CD30 in anaplastic large cell (Ki-1) lymphoma (plasma membrane plus Golgi area), (**b**): placental ALP on the villi of normal placenta (apical plasma membrane), (**c**): CA125 in ovarian serous adenocarcinoma (mainly seen in the Golgi area), (**d**): IgG in plasmacytoma (coarse vesicles), (**e**): intrinsic factor in parietal cells in the oxyntic gastric mucosa (fine vesicles), (**f**): nonspecific cross-reacting antigen in lysosomes of lung adenocarcinoma cells (cell line PC-9) (coarse granular), (**g**): insulin in pancreatic insulinoma (fine granular), (**h**): creatine kinase-MM isozyme in normal striated muscle (diffuse cytoplasmic), (**i**): cytokeratin in well-differentiated gastric adenocarcinoma (along the plasma membrane), (**j**): cytokeratin in poorly differentiated gastric adenocarcinoma (concentrated in the Golgi area), (**k**): Ki-67 in breast cancer (nucleolar accentuation) and (**l**): proliferating cell nuclear antigen in testicular seminoma (diffuse nuclear).

**Figure 46 cells-10-01501-f046:**
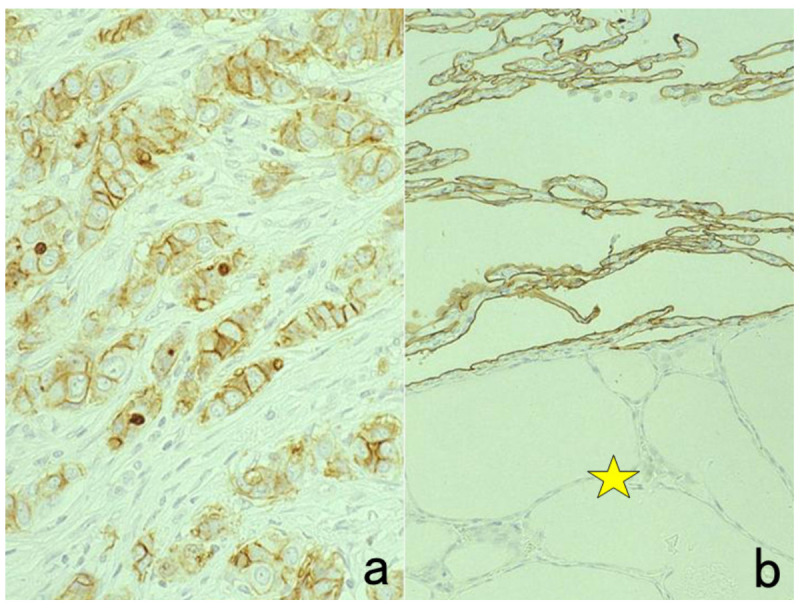
Plasma membrane localization of Ki-67. (**a**): Breast cancer and (**b**): lung with ectopic thyroid tissue. Ki-67 may show strange, cross-reactive localization on the plasma membrane. Membranous positivity in invasive ductal carcinoma cells and alveolar type I pneumocytes is unique but nonspecific. The star indicates ectopic thyroid tissue in the lung.

**Figure 47 cells-10-01501-f047:**
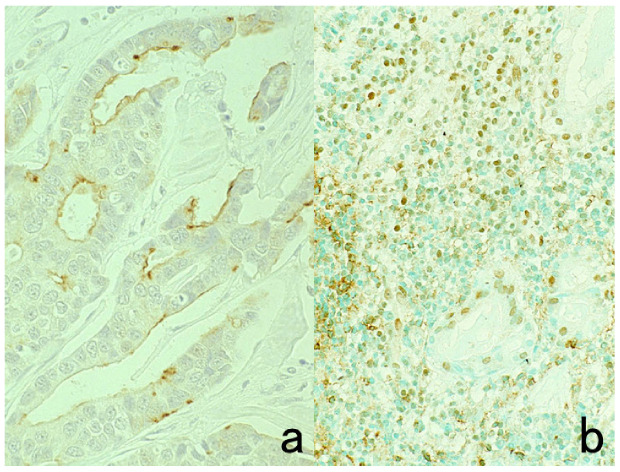
Examples of nonspecific immunostaining. (**a**): Membranous positivity of PgR in breast cancer and (**b**): nuclear positivity of CD45RO (UCHL1) in nasal NK-cell lymphoma. The intracellular localization pattern is unusual, so the results must be regarded as nonspecific. The nuclei in panel (**b**) are counterstained with methyl green.

**Figure 48 cells-10-01501-f048:**
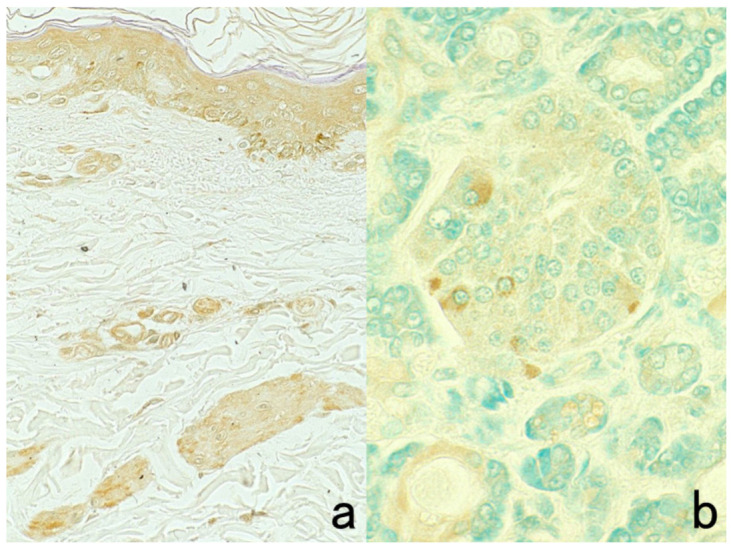
Myoglobin immunostaining in the skin (**a**) and pancreas (**b**) using a commercial rabbit antiserum. Rabbit antiserum available from Agilent/Dako may contain natural antibodies against intermediate filament proteins. In the skin, epidermal keratinocytes, vascular endothelial cells, and smooth muscle cells of erector pili are weakly stained. In the pancreas, some islet cells are faintly immunostained by the incidentally contaminating antibodies. The nuclei in panel (**b**) are counterstained with methyl green.

**Figure 49 cells-10-01501-f049:**
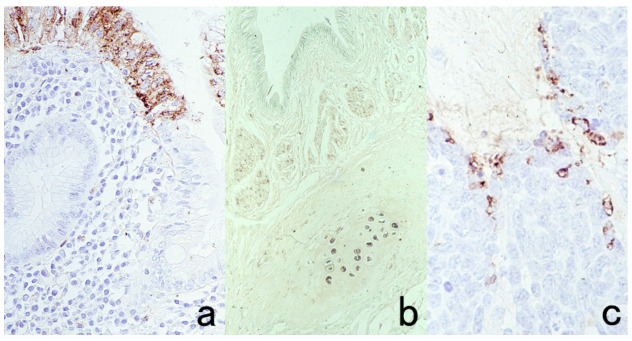
Cross-reactive antibodies against the carrier protein. Cholecystokinin (CCK) is a polypeptide with 33 amino acid residues. The CCK antiserum was raised in a rabbit by using the *Ascaris* extract as carrier proteins. *Ascaris* protein-like (cross-reactive) immunoreactivities are observed in the gastric foveolar cells (**a**), smooth muscle cells and cartilage in the bronchial wall (**b**), and small cell lung carcinoma cells (**c**). One must not judge as CCK-producing small cell carcinoma of the lung. Note the vesicular cytoplasmic staining in panel (**c**) rather than fine granular staining.

**Figure 50 cells-10-01501-f050:**
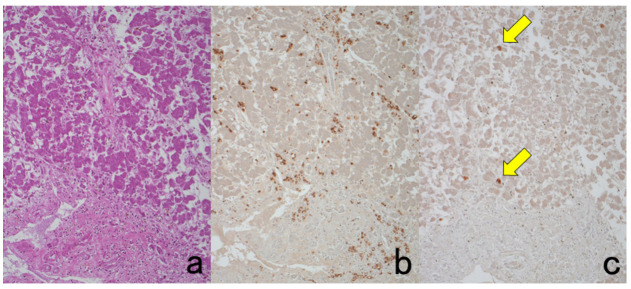
Cross-reactive immunostaining of *Mycobacterium leprae* in the liver with a rabbit anti-HBs antiserum, immunized with complete adjuvant containing the extract of *M. tuberculosis*. (**a**): H&E, (**b**): anti-HBs rabbit antiserum, (**c**): anti-HBs monoclonal antibody. Leproma is seen mainly in the portal area. The antiserum visualizes *M. leprae* in the cytoplasm of macrophages richly distributed both in the portal triad (leproma cells) and in the sinusoid (Kupffer cells). The monoclonal antibody detects HBs antigen-positive hepatocytes (arrows) without background cross-reactivity.

**Figure 51 cells-10-01501-f051:**
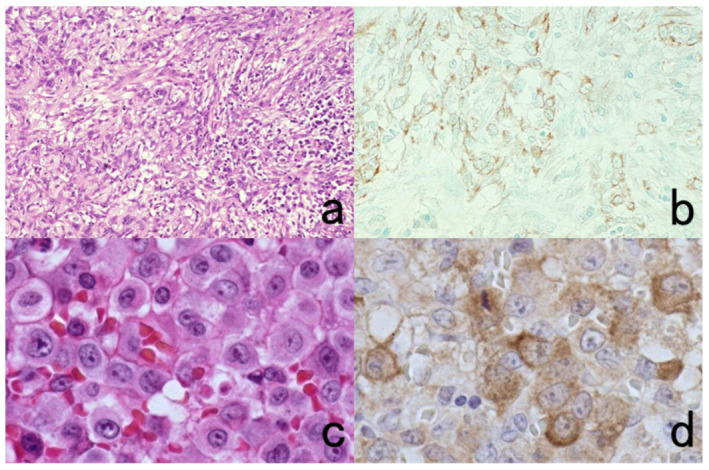
Diagnostic utility and pitfalls of cytokeratin immunostaining. Top panels (**a**,**b**): Spindle cell carcinoma of the skin and Bottom panels (**c**,**d**): anaplastic large cell (Ki-1) lymphoma**. Left panels** (**a**,**c**): H&E and Right panels (**b**,**d**): immunostaining with wide-spectrum anti-cytokeratin antiserum. Cytokeratin expression confirms the epithelial nature of the spindle cells in the skin. Because of the epithelioid morphology and cytokeratin expression in Ki-1 lymphoma, diagnostic confusion with metastatic undifferentiated carcinoma may occur. The nuclei in panel (**b**) are counterstained with methyl green.

**Figure 52 cells-10-01501-f052:**
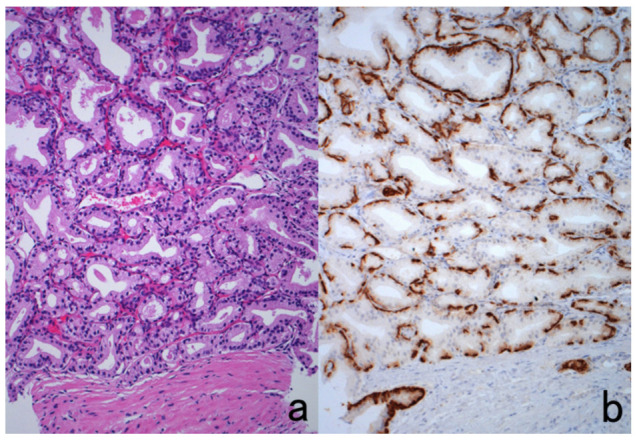
Post-atrophic hyperplasia of the prostate. (**a**): H&E and (**b**): immunostaining for cytokeratin 5/6. H&E histology resembles well-differentiated prostatic adenocarcinoma. Immunostaining for cytokeratin 5/6, a basal cell marker, clearly shows the consistent association of basal cells around the acini, definitely indicating a benign lesion.

**Figure 53 cells-10-01501-f053:**
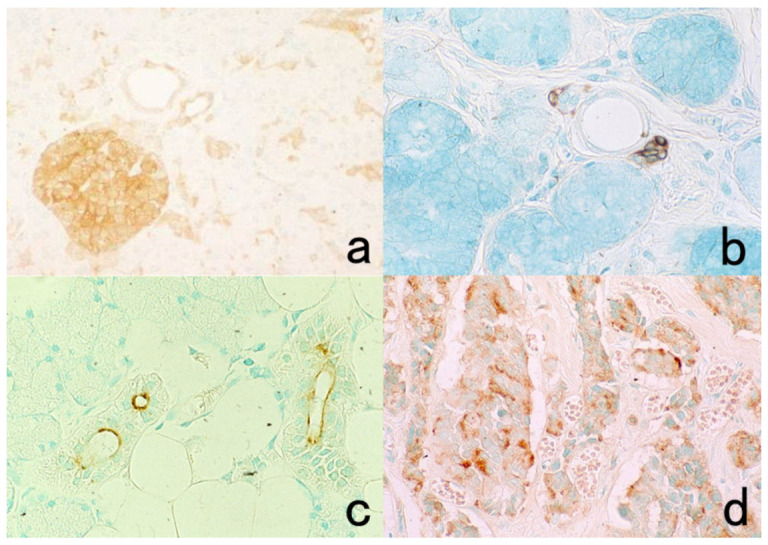
Expression of specific markers in unrelated cells. (**a**): Neuron-specific enolase (NSE) in the normal pancreas, (**b**): glial fibrillary acidic protein (GFAP) in the normal salivary gland, (**c**): prostate-specific antigen (PSA) in the normal salivary gland and (**d**): prostatic acid phosphatase (PAcP) in rectal carcinoid tumor. NSE is expressed not only in islets but also pancreatic ductal cells (**a**). In the salivary gland, GFAP immunostains some of the myoepithelial cells (**b**), and PSA antiserum decorates the apical cytoplasm of the ductal cells (**c**). Rectal carcinoid tumor cells are frequently immunoreactive for PAcP (**d**). The nuclei are counterstained with methyl green.

**Figure 54 cells-10-01501-f054:**
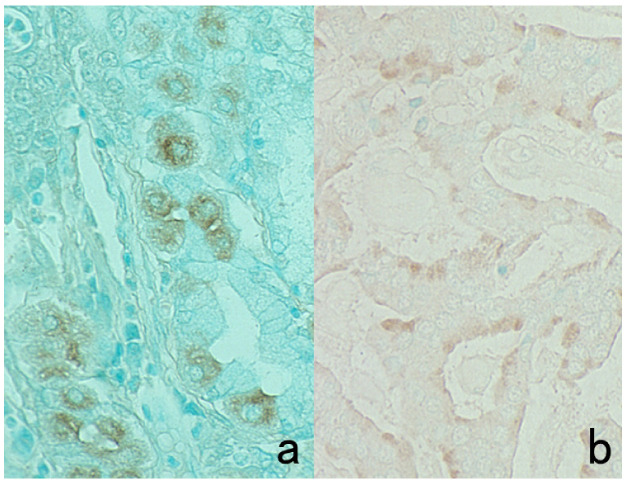
Nonspecific binding of IgG in certain cells in FFPE sections. (**a**): Glicentin (enteroglucagon equivalent) immunoreactivity in gastric parietal cells and (**b**): motilin immunoreactivity in rectal carcinoid tumor. Certain cells may adsorb the antibody molecules nonspecifically, causing false-positive immunostaining. The nuclei are counterstained with methyl green.

**Figure 55 cells-10-01501-f055:**
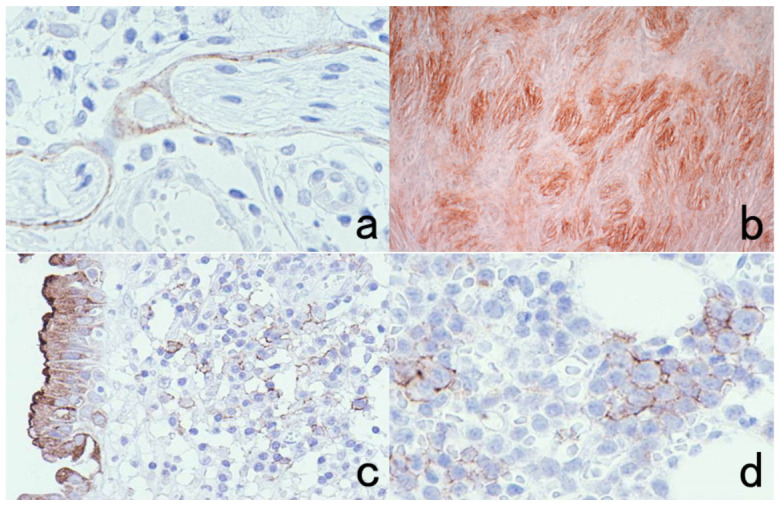
Epithelial membrane antigen (EMA) expression in normal (**a**,**c**) and neoplastic (**b**,**d**) perineurial cells (top panels: (**a**,**b**)) and plasma cells (bottom panels: (**c**,**d**)). It should be noted that EMA is positive in certain non-epithelial cells and their tumors (perineurioma and multiple myeloma). Perineurial cells surrounding the peripheral nerve sheath represent peripheral meningothelial cells. Chronic cervicitis containing numerous plasma cells is covered with EMA-positive columnar epithelial cells.

**Figure 56 cells-10-01501-f056:**
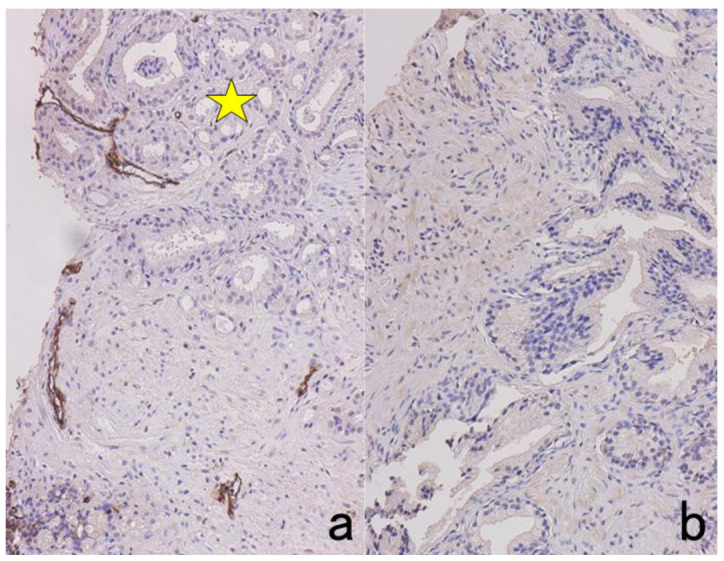
Immunohistochemical blood typing in needle biopsy specimens of the prostate. Specimens from two patients were mixed up in one tissue container. The blood group was type A in one patient and type O in the other. Immunostaining for type A substance is positive in endothelial cells in panel (**a**) but negative in panel (**b**). It is evident that the specimen of the patient with type A blood group contains well-differentiated adenocarcinoma (star).

**Figure 57 cells-10-01501-f057:**
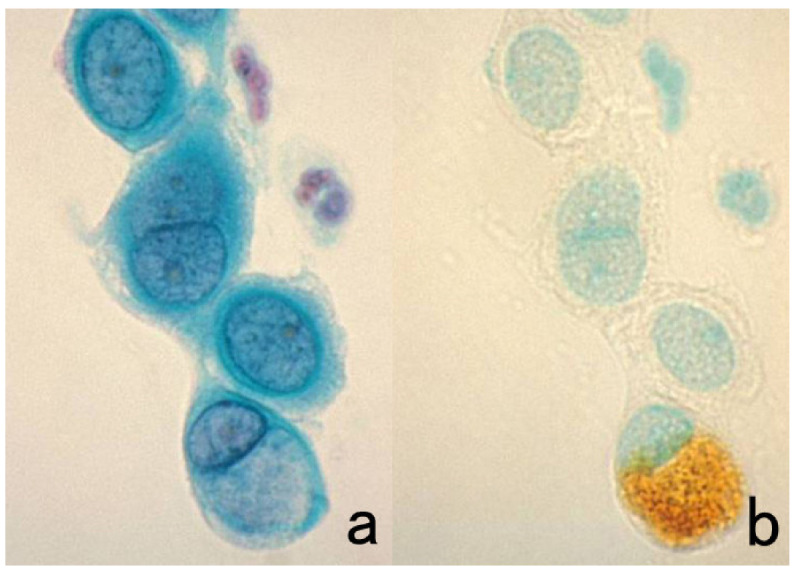
Re-staining of a cytology specimen with *Chlamydia trachomatis* infection. (**a**): Papanicolaou and (**b**): immunostaining with a mouse monoclonal antibody B104.1 after removal of the cover glass and decoloration. A cytoplasmic nevular inclusion body in a cervical columnar cell is clearly immunoreactive for the *C. trachomatis* antigen, confirming the cytodiagnosis of chlamydiasis. The nuclei in panel (**b**) are counterstained with methyl green.

**Figure 58 cells-10-01501-f058:**
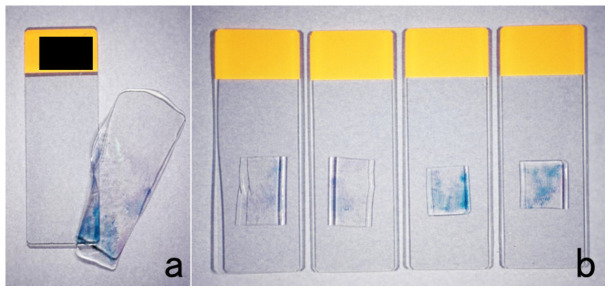
The cell transfer technique. After removal of the cover glass, the cell preparation is covered with mounting resin solution that solidifies to become a thick membrane sheet. Cells are transferred to the solidified resin (**a**). After cutting the resin membrane into several pieces, they are placed onto silane-coated glass slides, which should be fully dried in an incubator (**b**). The resin component can easily be removed by dipping the specimens in xylene. Now, the specimens are ready for immunostaining using HIER.

**Figure 59 cells-10-01501-f059:**
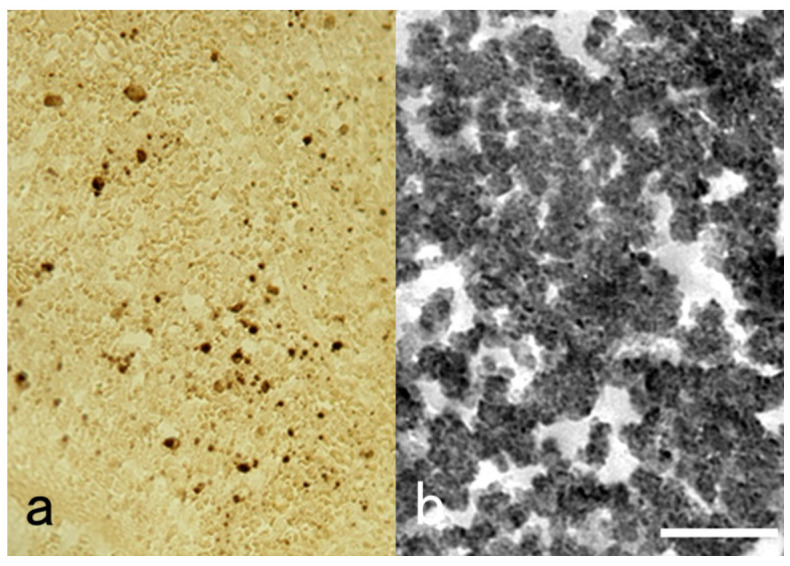
Pre-embedding immunoelectron microscopy for SFTS virus using a TACAS glass slide. (**a**): Light microscopic chromogenic immunostaining and (**b**): immunoreactive round-shaped viral particles, around 100 nm in size, at the ultrastructural level. The FFPE splenic red pulp from an autopsy case was immunostained with a monoclonal antibody to SFTS virus after HIER. For pre-embedding immunoelectron microscopy, the immunostained section was peeled off for transfer to an Epon-embedded block by the inverted beam capsule method. The miraculous TACAS glass slide prevents detachment of sections during immunostaining and allows cell transfer after immunostaining. Bar indicates 500 nm.

**Figure 60 cells-10-01501-f060:**
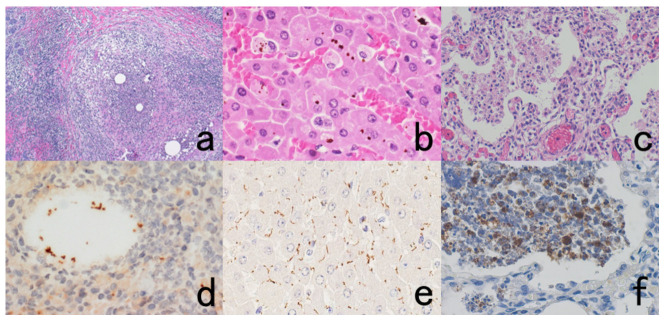
Visualization of bacteria in infectious lesions using cross-reactive antisera. Left panels (**a**,**d**): *Corynebacterium kroppenstedtii*-induced granulomatous mastitis, Central panels (**b**,**e**): leptospirosis (*Leptospira interrogans* infection or Weil’s disease) in the liver and Right panels (**c**,**f**): *Haemophilus pertussis*-induced pneumonia. Top (**a**–**c**): H&E, Bottom (**d**–**f**): immunostaining using commercially available antiserum against *Treponema pallidum* (**d**), *Escherichia coli* (**e**), and *Bacillus cereus* (**f**). The causative bacteria are visible through the cross-reactivity of the antisera. Bacteria are not recognizable in H&E preparations.

**Figure 61 cells-10-01501-f061:**
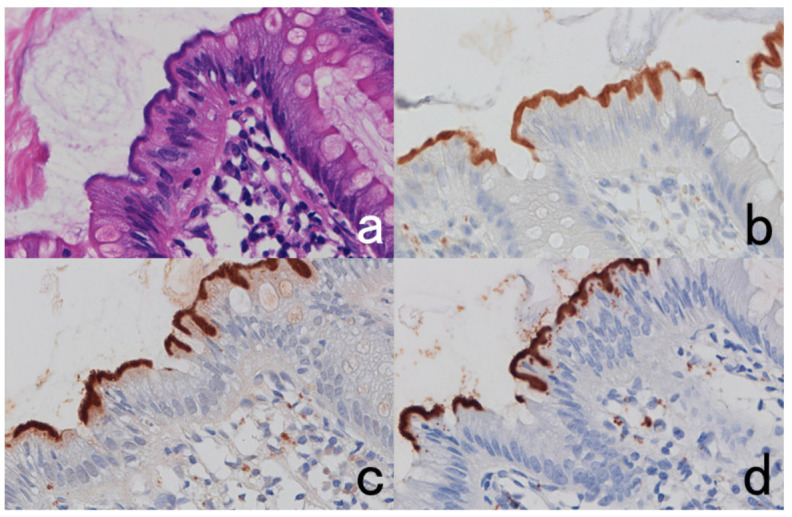
Intestinal spirochetosis caused by colonization of *Brachyspira aalborgi*. (**a**): H&E, (**b**–**d**): immunostaining using *Treponema pallidum* antiserum (**b**), BCG antiserum, (**c**) and *E. coli* antiserum (**d**). Long and basophilic spiral bacteria closely attach onto the apical surface of the colonic columnar cells (**a**). They are clearly immunostained with antisera to *T. pallidum*, BCG, and *E. coli* (**b**–**d**).

**Figure 62 cells-10-01501-f062:**
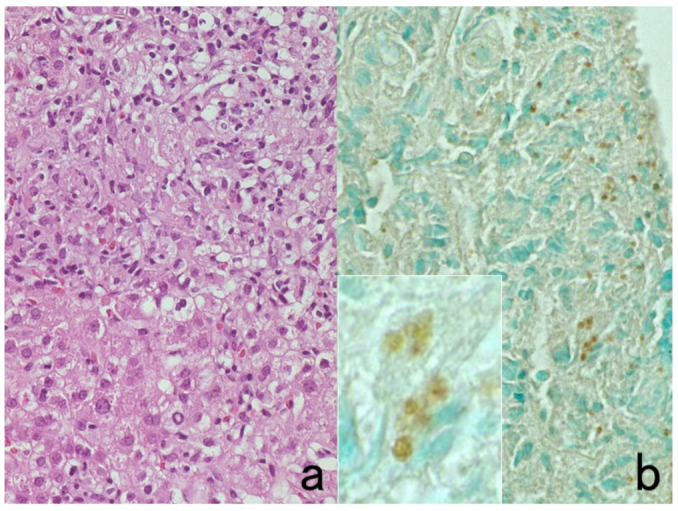
Immunostaining with the patient’s serum (I). Liver biopsy of visceral leishmaniasis (kala azar). (**a**): H&E and (**b**): 1:500 dilution of the patient’s own serum (inset: high-power view). Non-caseous epithelioid granulomas are dispersed in the liver parenchyma. Antibodies in the patients’ own serum identify red cell-sized positive signals in the cytoplasm of epithelioid cells and Kupffer cells. Although the specificity of the patient’s serum is totally unknown in this case, it significantly contributed to suggest the causative pathogens to be protozoa. The nuclei in panel (**b**) are counterstained with methyl green.

**Figure 63 cells-10-01501-f063:**
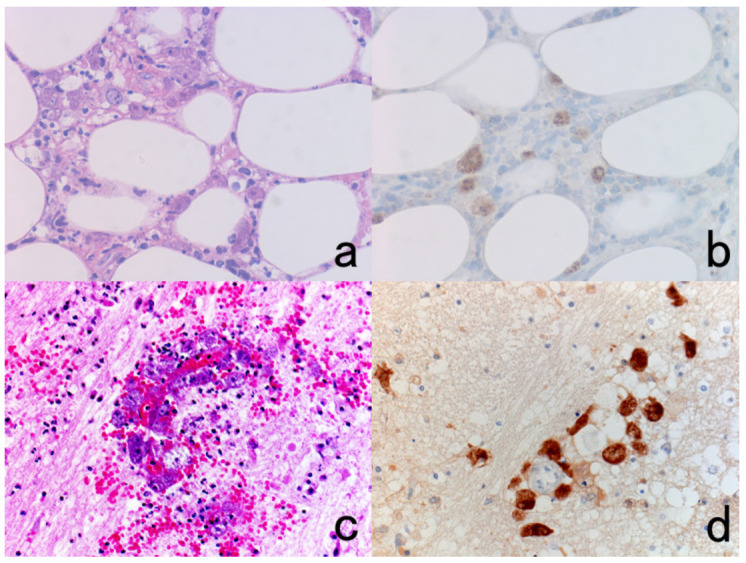
Immunostaining with another patient’s serum (II). Left (**a**,**c**): H&E and Right (**b**,**d**): immunostaining with 1:500 dilution of serum from another case of balamuthiasis. *Balamuthia mandrillaris* infection in the skin (top panels: (**a**,**b**)) and brain (bottom panels: (**c**,**d**)) of the same patient. Trophozoites are visible in H&E-stained cutaneous panniculitis and chronic encephalitis, and they are strongly immunoreactive with the serum from another patient who suffered Balamuthia encephalitis.

**Figure 64 cells-10-01501-f064:**
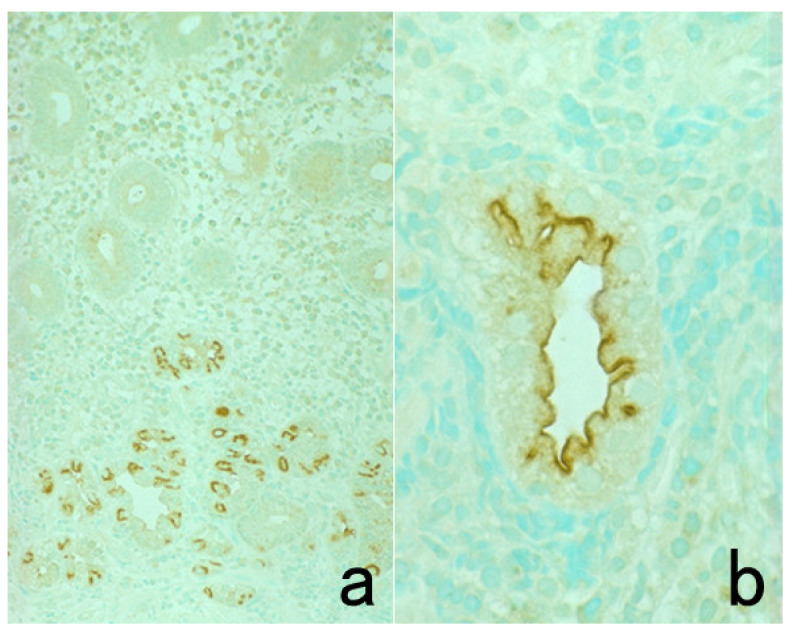
Images of gastric parietal cells in FFPE sections with the serum of a patient with type A (autoimmune) gastritis ((**a**): low power, (**b**): high power). The patient’s 1:20 diluted serum, containing autoantibodies against proton pump (H^+^, K^+^-ATPase) molecules, decorates C-shaped intracytoplasmic structures (secretory canaliculi) in the acid-secreting parietal cells. The nuclei were counterstained with methyl green.

**Figure 65 cells-10-01501-f065:**
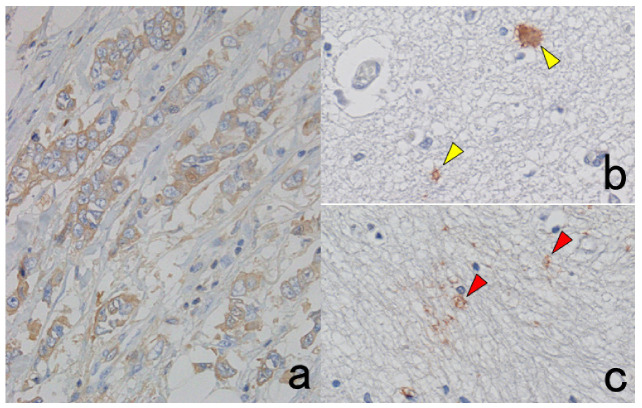
Immunostaining with the serum of two patients with autoimmune encephalitis. (**a**): Paraneoplastic (breast cancer-associated) autoimmune encephalitis (Stiff-Man syndrome) with autoantibodies to amphiphysin, a 128-kDa presynaptic protein. (**b**,**c**): Autoimmune limbic encephalitis. The patient’s 1:20 diluted serum immunostains her own FFPE breast cancer cells (**a**). The 1:20 diluted serum of limbic encephalitis decorates astroglia (yellow arrowheads in panel (**b**)) and their glial processes (red arrowheads in panel (**c**)) in normal basal ganglia embedded in paraffin from another autopsy case.

## Data Availability

The datasets generated during and/or analyzed during the current study are available from the corresponding author on reasonable request.

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
