# Peer review of "Pitfalls and Caveats in Applying Chromogenic Immunostaining to Histopathological Diagnosis"

_cells, 2021, doi:10.3390/cells10061501_

Round 1
Reviewer 1 Report
In the review manuscript entitled “Pitfalls and caveats in applying immunostaining to histopathological diagnosis” the author provides a broad overview of the application of immunohistochemistry (IHC) to the diagnostic field. The review stretches from theoretical notions to very practical tips about the application of specific techniques and interpretation of the findings including multiple examples drawn from personal experience.
As explained by the numerous points of criticisms appended below, the manuscript in its current form does not meet the scientific standards of the journal. Suggestions on how to improve organization, clarity, and presentation of the content have been included in the reviewer’s comments.
In addition, the author must seek the help of a scientific professional who is fully proficient in the English language and review the use of grammar, syntax, and proper terminology before the manuscript can be reconsidered for publication.
GENERAL POINTS OF CRITICISM AND COMMENTS
While the notions included in this manuscript are scientifically relevant, their value is limited to an audience with advanced knowledge of IHC. In this context, only a readership with IHC expertise can fully appreciate the content of this manuscript.
The work is based on the personal experience/preferences of the author rather than providing an objective overview of the topic. This is underlined by the systematic lack of references in the text and the use of personal examples instead. Note that this review of over 50 pages includes an exceptionally low number of references (only 72). In addition, as highlighted in some of the reviewer’s comments, there is an overall tendency to make assumptions based on the reagents and protocols the author is accustomed to without considering the full spectrum of options that are actually available in the field.
As mentioned, the review touches on so many aspects associated with diagnostic IHC, and sometimes it does so without delineating a clear logic structure in the outline or narrative flow. Therefore, there is an overall impression that some features of the manuscript are randomly included and not properly organized. For example, it is not clear why the manuscript has a subsection on the use of archival pathology material. Furthermore, in the section concerning the choice of fixatives, a brief digression on the effect of fixation on DNA and RNA is also reported. This doesn’t appear pertinent to the topic and it distracts the reader from the main subject. Lastly, while it seems that most of the manuscript focuses on the application of standard diagnostic techniques on FFPE section, this is never clearly stated and there are a few instances where cytology preparations, frozen sections, and fixatives different from formalin are considered. This creates confusion in the reader and undermines the fruition and translatability of the concepts herein contained.
The Author should do an effort to simplify the outline of the manuscript providing a more systematic and concise guide into the art of IHC. In this context, it would be useful for the author to include a clear statement on what he/she intends to cover in this review. This would allow narrowing the focus on specific aspects avoiding confusing digressions on aspects that are not directly connected with the scope of the work.
SPECIFIC POINTS OF CRITICISM AND COMMENTS
Abstract:
The term “morphofunctional correlations” is unclear. IHC allows the study of specific biological and clinical features such as the origin/histogenesis, behavioral characteristics, therapeutic targets, and prognostic/predictive biomarkers of a neoplastic lesion. Please, consider replacing “morphofunctional” with “clinicopathological”.
- The needs for immunostaining
Line 30_IHC is not only critical to formulating a diagnosis, but also to inform the biology of a lesion, patient prognosis, and predict response to certain therapeutic approaches. Please, consider these additional aspects in your examples.
Line 33_See above, the term “morphofunctional diagnosis” is unclear.
In general, consider clarifying and using the following definitions in a consistent way throughout the manuscript:
IHC markers (as other biomarkers) can be classified into four types: diagnostic, prognostic, predictive, and therapeutic. A diagnostic marker defines the nature (histogenesis/origin) of a lesion. A predictive marker predicts the response of the patient to targeted therapy. A prognostic marker is a clinical/biological characteristic that provides information on the likely course of the disease informing the probable outcome of the patient. A therapeutic marker is generally a structure that could be used as the target for a therapy (the definition partially overlaps with the one of the predictive markers).
- Choice of methodology for immunostaining
Line 50_Please, consider replacing “autostaining apparati” with “automated instruments” or “automated platforms”.
Lines 51-53_Please, consider including the source/vendor of the reagents listed (here and elsewhere in the manuscript).
Lines 59_“b) staining steps are a little bit complicated”. Please, consider using scientific jargon and explain the added complexity of the standard biotin-based detection methods.
Line 67_”because of the agitation of background staining”. This sentence is unclear. Please, rephrase it. Also, specify what those “ultra-high sensitivity techniques” are.
Line 68_consider replacing “stably keeping” with “maintaining”.
Lines 70-79_There are several options to dilute antibodies and preserve their IHC performances that are not considered in this paragraph. One of the most convenient method is the addition of glycerol in the diluent as cryopreservative thus eliminating the risks associated with repeated freezing and thawing cycles. This method is widely used by leaders in manufacturing and distributing antibodies for IHC such as Abcam and Cell Signaling Technology.
Line 84_Mayer’s hematoxylin is technically BLUE and not PURPLE.
- Application of double staining
It is not clear why the focus is limited to double staining using chromogenic IHC. A comprehensive dissertation on that topic would include a description of other (arguably much more common) approaches for multiplexing using fluorescence. In that context, the author should provide a side-by-side comparison of the pros and cons of multiplexing using chromogenic and fluorescence techniques.
Line 96_Please, consider replacing “immunostaining” with “chromogenic immunohistochemistry”. The author uses the term “immunostaining” in comparison with “immunofluorescence”. However, “immunostaining” is a very broad designation that include both chromogenic IHC and immunofluorescence.
Line 109_The author should explain what he/she means for “serial sections” and how this approach compares to multiplexing.
- Effect of formalin fixation
5.1. Antigenic alterations by formalin fixation
The author should include the specific molecular alterations caused by formalin on the different components of the tissue specimens (DNA, RNA, protein, and other molecules) in a more structured and systematic manner including the potential preanalytical/analytical disadvantages and possible solutions to restore the native molecular properties (e.g. epitote retrieval)
Lines 137-144_The advantages of using neutral buffered formalin instead of acidic formalin have been widely documented. In that context, the author should refrain from the use of misleading and controversial statements that are based on personal preferences and not supported by scientific evidence.
Figure 5. Examining the pictures, it appears that FFPE tissue sections yield a much better preservation of the antigens compared to frozen sections. This seems to be in contrast with what is reported in the text. Please, clarify.
5.2. Choice of fixatives
A precise definition of epitope retrieval is critical at this point for the reader to understand the concept explained in the following paragraphs.
Line 173_The author should include references to confirm that platelet factor 4 and insulin are stored in lipid-rich granules and verify that their claims about the loss of antigenicity and the osmiophilic properties are adequately supported by the literature.
Line 176_Reference needed.
5.3. Use of archival pathology material
Line 187_See comment above regarding the use of acidic formalin.
- Artifacts by formalin fixation
6.1. Penetration of plasma proteins into the cytoplasm of certain cells: A diffusion artifact
Assuming that the diffusion artifact is not anecdotal evidence, the author should include relevant references demonstrating its development.
6.2. False-positive and false-negative results due to uneven formalin fixation
This subchapter would benefit from the discussion on how homogenous fixation resulting from a timely exposure to formalin is critical to obtain consistent and reproducible IHC results when targeting phosphorylated epitopes.
The author should be also suggesting the implementation of relevant procedures for sample preparation to avoid these artifacts.
- Tips and pitfalls in immunostaining
7.1. To get low background staining
Consider rephrasing as “how to get low background staining”
Lines 309-310
- a) The concentration of the primary or secondary antibodies is too high.
- b) The antibody titer is too low.
Point b seems to contradict point a. Please, clarify.
While antibody concentration and inadequate rinsing represent important factors when troubleshooting background staining, other (arguably more important) variables such as protein blocking step, duration of the antigen retrieval, the addition of detergents to the washing buffers are not taken into consideration by the author. A thorough elaboration of these additional features is critical to provide the reader with a comprehensive overview of the matter.
In addition, it should be clarified that PBS is not the only washing buffer used in IHC.
7.2.3. Endogenous protein A and protein G activity
It would be worth adding that, in a similar fashion, an unspecific signal may result from the binding of the Fc fraction of the IgG by the FcR displayed by a variety of inflammatory/immune cells (PMID: 16006601). In addition, nonspecific staining of mast cell granules following immunoperoxidase-based IHC is a well-documented finding that has been attributed to ionic interaction between the F(ab′)2 segments of antibodies and the heparin constituent of the mast cell secretory granules (PMID: 8226102).
7.2.4. Endogenous pigments
Please, note that the formation of formalin pigment is greatly reduced by the use of buffered formalin instead of acidic formalin.
7.3. Staining artifacts: effects of dewatering, contamination of cytokeratin-positive squams (please, consider using scales or debris instead of squams) and insufficient deparaffinization
Consider using the term “dehydration” instead of “dewatering”.
7.4. Antigenic deterioration due to prolonged conservation of unstained sections
Line 453-453_“Particularly susceptible are nuclear antigens, such as Ki-67 (MIB-1), p53, ER and PgR (Figure 24)” Reference needed.
7.5. Effects of section-stretching temperature on a hot plate and drying period after cutting sections
The author should specify which type of section (i.e., frozen or paraffin) he is referring to as it is not clear from the text.
Lines 472-473_Assuming that the author is considering paraffin sections, it is hard to believe that 40C for 20 seconds or at 50C for 10 seconds would be enough for a satisfactory stretching as standard paraffin for histology has a melting point of 56 to 58 C.
7.8. Pitfalls and caveats in antigen retrieval sequences
Line 562_There are different types of glass slides with enhanced adhesion properties for the sections, an overview of the most common ones would be interesting.
7.8.1. Proteinase pretreatment
Lines 567-569_”The proteinase pretreatment is effective for retrieving the antigenicity of type 4 collagen (Figure 30) and laminin. Cytokeratins and lymphocyte surface markers may be retrieved with this pretreatment” this statement is misleading, it all depends on the epitope specifically recognized by the antibody in question and the effect of the proteinase on that molecular target. Since there are thousands of antibodies for those targets, it must be clear to the readers that some will work well with proteinase pretreatment while others will not.
7.8.2. Heat-induced epitope retrieval (HIER)
Lines 371-373_It must be clear to the readers that these effects are not target-related but personal observations for the specific antibodies that the author has been using.
Fig 42, lines 694-698_Even if the large polymeric tail of the original Dako Envision technology is suboptimal for comprehensive antigen detection, the reviewer has used this reagent successfully in the past for the detection of non-mitotic proliferative cells with diverse Ki-67 clones including SolA15, MIB1 and SP6. In addition, it is hard to imagine a situation where only mitotic cells are stained because of the dissolution of the nuclear membrane. Please note that the author is considering FFPE sections where nuclei have been “opened up” by the sectioning itself and the nuclear membrane doesn’t pose any barrier to the penetration of the reagents.
7.8.3. Other methods for epitope retrieval
Lines 700-702_”Immunoreactivity of b-amyloid protein in the brain is retrieved by soaking in 100% formic acid solution for 5 minutes (Figure 43), while protease treatment and HIER are ineffective”. Again this is not universal. It must be stressed that this is true only for specific antibodies, while others might work with more standard antigen retrieval methods (see for example Cell Signaling Technology β-Amyloid (D3D2N) Mouse mAb #15126).
- How to judge the immunostained results
8.1. False positivity or equivocal negativity
Here the author provides a single example, but it would be much more useful if he can also draw a flow chart on how to recognized false positivity/equivocal negativity for diagnostic purposes.
8.3. Specificity of antibodies
Lines 778-779_The author should be more specific about the identity of the antibody. There are several polyclonal antibodies against myoglobin, it is difficult to believe that all of them show the same non-specific signature.
Lines 791_ Reference/s needed.
8.4. Specificity of the specific markers
The meaning of this title is unclear. The author is describing examples of uncommon expression patterns of well-established antibodies used in diagnostic pathology. Please, rephrase the title so that it can better represent what is reported in this subsection.
Line 816_Pitfalls of cytokeratin immunostaining are not described in Fig. 51.
No pitfalls are presented here.
Line 845 (Fig. 53 caption)_Please, replace “indifferent” with “different”.
8.5. Nonspecific adsorption of antibodies by certain cells
As mentioned above, it would be worth adding that, in a similar fashion, an unspecific signal may result from the binding of the Fc fraction of the IgG by the FcR displayed by a variety of inflammatory/immune cells (PMID: 16006601). In addition, nonspecific staining of mast cells granules following immunoperoxidase-based IHC is a well-documented finding that has been attributed to ionic interaction between the F(ab′)2 segments of antibodies and the heparin constituent of the mast cell secretory granules (PMID: 8226102).
8.6. Positive and negative controls
This subsection would benefit from the definition of biological and technical controls. Biological controls are accomplished by including tissues that are known to express (or not to express) the target protein (e.g., for insulin IHC one should include pancreas as biologically positive control and tonsil as biologically negative control). Technical controls are serial sections of the biopsy under diagnosis that, depending on the primary antibody used, are incubated with normal animal serum (for polyclonal antibodies) or isotype-matched immunoglobulins (for monoclonal antibodies).
11.1. Use of antisera against pathogens showing wide cross-reactivity
- In addition to citing his/her own article, the author should add additional references demonstrating the suitability of this technique not only for bacterial organisms but also for fungi and protozoa. Please, see here a couple of examples: PMID: 11301538, PMID: 9448206, PMID: 9448206, PMID: 9448206.
Author Response
Point-by-point Responses to Reviewer 1’s comments
Cells-1219134
In the review manuscript entitled “Pitfalls and caveats in applying immunostaining to histopathological diagnosis” the author provides a broad overview of the application of immunohistochemistry (IHC) to the diagnostic field. The review stretches from theoretical notions to very practical tips about the application of specific techniques and interpretation of the findings including multiple examples drawn from personal experience.
As explained by the numerous points of criticisms appended below, the manuscript in its current form does not meet the scientific standards of the journal. Suggestions on how to improve organization, clarity, and presentation of the content have been included in the reviewer’s comments.
Thank you very much for your detailed advice and mindful suggestions. I revised and improved the manuscript as much as I could. My point-by-point responses are described below.
In addition, the author must seek the help of a scientific professional who is fully proficient in the English language and review the use of grammar, syntax, and proper terminology before the manuscript can be reconsidered for publication.
According to the valuable advice and suggestions given from three reviewers, I have checked the English grammar of my manuscript, as much as possible. I believe that the revised manuscript has been brushed up significantly.
GENERAL POINTS OF CRITICISM AND COMMENTS
While the notions included in this manuscript are scientifically relevant, their value is limited to an audience with advanced knowledge of IHC. In this context, only a readership with IHC expertise can fully appreciate the content of this manuscript.
Yes. That is just the aim of this review article. I added the following phrase in the Introduction: “The present review article overviews technical aspects, knowhows, pitfalls and trouble-shooting guides in routinely performing chromogenic immunostaining, especially for experienced persons with immunohistochemical expertise.”
The work is based on the personal experience/preferences of the author rather than providing an objective overview of the topic. This is underlined by the systematic lack of references in the text and the use of personal examples instead. Note that this review of over 50 pages includes an exceptionally low number of references (only 72). In addition, as highlighted in some of the reviewer’s comments, there is an overall tendency to make assumptions based on the reagents and protocols the author is accustomed to without considering the full spectrum of options that are actually available in the field.
This review was written according to my over 40 years’ experience of diagnostic IHC. I’m proud of being a pioneer of diagnostic IHC using FFPE sections in Japan. It is my deep regret that my past review articles on diagnostic IHC were solely written in Japanese, seen as early as in 1984. I added 37 selected references to be a total of 109 references cited.
The data showing pitfalls and caveats in diagnostic IHC were fundamentally obtained when unexpected or curious observations were experienced. Therefore, it looks that “there is an overall tendency to make assumptions based on the reagents and protocols the author is accustomed to without considering the full spectrum of options that are actually available in the field”. The diagnostic IHC is not meticulously planned as the well-designed experiment. I just simply described the facts/phenomenon which may happen during the practice of diagnostic IHC. The knowledge will help pathology technicians and pathologists understand what happens now, I believe. That is the reason why I summarized the review principally based upon our experience.
As mentioned, the review touches on so many aspects associated with diagnostic IHC, and sometimes it does so without delineating a clear logic structure in the outline or narrative flow. Therefore, there is an overall impression that some features of the manuscript are randomly included and not properly organized. For example, it is not clear why the manuscript has a subsection on the use of archival pathology material.
I would like to emphasize the usefulness of the analysis using archival specimens. Namely, it should be emphasized that formalin fixation can preserve the antigenicity and genomic sequences for a long period of time over 100 years. Fixation in acidic formalin deteriorates them to a considerable degree, but one must not abandon the analysis with IHC and PCR. Formalin can keep them at least above the level of detection. The author sincerely wants the readers to know the “positive” and “beneficial” effect of formalin fixation.
I added two short phrases in 5.3 (Use of archival pathology material):
The author would like to emphasize a beneficial aspect of formalin fixation by demonstrating IHC examples of archival pathology specimens. (lines 233-234)
Surprising examples are described below. (lines 230-231)
Furthermore, in the section concerning the choice of fixatives, a brief digression on the effect of fixation on DNA and RNA is also reported. This doesn’t appear pertinent to the topic and it distracts the reader from the main subject.
We diagnostic pathologists are requested simultaneously to analyze the antigenic expression and genomic change of the specimens. Therefore, the knowledge of DNA/RNA preservation is also very much important. Nowadays, the preservation of antigenicity and genomic information must be inseparable. Hence, a brief comment was included in the text. Not only proteins but also DNA/RNA signals can be kept to certain degrees at the level worthy of analysis in FFPE specimens.
Lastly, while it seems that most of the manuscript focuses on the application of standard diagnostic techniques on FFPE section, this is never clearly stated and there are a few instances where cytology preparations, frozen sections, and fixatives different from formalin are considered. This creates confusion in the reader and undermines the fruition and translatability of the concepts herein contained.
In daily practice of diagnostic pathology, the main target of IHC must be FFPE material. I also have a long experience of immunocytochemistry for cytology, but it can be a separate manuscript, I believe. This review article thus focused on the chromogenic IHC using FFPE sections. When necessary, chromogenic immunostaining using cytology preparations, frozen sections and fixatives different from formalin were included, as much as possible. The main readers should be the experienced persons with immunohistochemical expertise, as described in the Introduction (lines 30-32).
In lines 66-68, I added a sentence “The present review article is focused on chromogenic immunostaining for diagnostic pathology routinely employing FFPE sections”.
The Author should do an effort to simplify the outline of the manuscript providing a more systematic and concise guide into the art of IHC. In this context, it would be useful for the author to include a clear statement on what he/she intends to cover in this review. This would allow narrowing the focus on specific aspects avoiding confusing digressions on aspects that are not directly connected with the scope of the work.
Thanks for your precious advice. In the Introduction, I stated as follows.
The present review article overviews technical aspects, knowhows, pitfalls and trou-ble-shooting guides in routinely performing chromogenic immunostaining, especially for experienced persons with immunohistochemical expertise. It should be emphasized that all the data presented below were obtained by the author and author’s colleagues. (lines 25-34)
SPECIFIC POINTS OF CRITICISM AND COMMENTS
- Abstract:
The term “morphofunctional correlations” is unclear. IHC allows the study of specific biological and clinical features such as the origin/histogenesis, behavioral characteristics, therapeutic targets, and prognostic/predictive biomarkers of a neoplastic lesion. Please, consider replacing “morphofunctional” with “clinicopathological”.
The word “morphofunctional” was replaced by “clinicopathological”,
I inserted a sentence in the abstract: In neoplastic lesions, immunohistochemistry allows the study of specific clinical and biological features such as the histogenesis, behavioral characteristics, therapeutic targets and prognostic biomarkers. (lines 12-14)
- The needs for immunostaining
Line 30_IHC is not only critical to formulating a diagnosis, but also to inform the biology of a lesion, patient prognosis, and predict response to certain therapeutic approaches. Please, consider these additional aspects in your examples.
I inserted a paragraph at the beginning of section 2.
IHC is critical both to informing the biology of lesions and to formulating the histopathological diagnosis. The patient’s prognosis and the response to therapeutic approaches may be predicted. (lines 36-38)
The caption of the section was changed to “The needs for chromogenic immunostaining in diagnostic pathology”,
Line 33_See above, the term “morphofunctional diagnosis” is unclear.
In general, consider clarifying and using the following definitions in a consistent way throughout the manuscript:
IHC markers (as other biomarkers) can be classified into four types: diagnostic, prognostic, predictive, and therapeutic. A diagnostic marker defines the nature (histogenesis/origin) of a lesion. A predictive marker predicts the response of the patient to targeted therapy. A prognostic marker is a clinical/biological characteristic that provides information on the likely course of the disease informing the probable outcome of the patient. A therapeutic marker is generally a structure that could be used as the target for a therapy (the definition partially overlaps with the one of the predictive markers).
Thank you very much. I inserted a new paragraph in lines 47-53.
IHC markers can be classified into four types, as other biomarkers: diagnostic, prognostic, predictive and therapeutic. A diagnostic marker defines the nature (histogenesis/origin) of the lesion. A prognostic marker suggests clinical/biological characteristics that provide information on the likely course of the disease and inform a probable outcome of the patient. A predictive marker predicts the response of the lesion to targeted therapy. A therapeutic marker represents a structure that could be used as the target of therapy.
- Selection of methodology for immunostaining
Line 50_Please, consider replacing “autostaining apparati” with “automated instruments” or “automated platforms”.
The caption of the section was changed to: Selection of methodology for chromogenic immunostaining.
I replaced the term “autostaining apparati” with “automated instruments”.
Lines 51-53_Please, consider including the source/vendor of the reagents listed (here and elsewhere in the manuscript).
I added the source/vendor of the reagents: Envision Flex (Agilent Technologies/Dako, Santa Clara, CA, USA), Simple Stain Max (Nichirei, Tokyo, Japan) and Novolink (Leica Biosystems/Novocastra, Nussloch, Germany). (lines 73-75)
I also tried to add the source/vendor of the reagents, as much as possible.
Lines 59_“b) staining steps are a little bit complicated”. Please, consider using scientific jargon and explain the added complexity of the standard biotin-based detection methods.
I reworded the phrase as follows: b) three steps are required for immunostaining,
Line 67_”because of the agitation of background staining”. This sentence is unclear. Please, rephrase it. Also, specify what those “ultra-high sensitivity techniques” are.
I changed the sentence:
“Antigenic detection with ultra-high sensitivity techniques such as CSA-II is not necessarily needed, because of the augmentation of background staining and difficulty in maintaining the highly diluted antibodies in refrigerators or freezers.” (lines 90-92)
Line 68_consider replacing “stably keeping” with “maintaining”.
I reworded “stably keeping” as “maintaining”, as described above.
Lines 70-79_There are several options to dilute antibodies and preserve their IHC performances that are not considered in this paragraph. One of the most convenient method is the addition of glycerol in the diluent as cryopreservative thus eliminating the risks associated with repeated freezing and thawing cycles. This method is widely used by leaders in manufacturing and distributing antibodies for IHC such as Abcam and Cell Signaling Technology.
Thanks for your advice.
I added a sentence in lines 103-105 with a new reference 14.
For eliminating the risk associated with repeated freezing and thawing cycles, the addi-tion of 25–50% glycerol in the diluent as a cryopreservative is a convenient and practical method of choice [14].
Line 84_Mayer’s hematoxylin is technically BLUE and not PURPLE.
I reworded purple with deep blue. (line 110)
- Application of double staining
It is not clear why the focus is limited to double staining using chromogenic IHC. A comprehensive dissertation on that topic would include a description of other (arguably much more common) approaches for multiplexing using fluorescence. In that context, the author should provide a side-by-side comparison of the pros and cons of multiplexing using chromogenic and fluorescence techniques.
I inserted an introductory paragraph describing the comparison with immunofluorescent IHC, as follows. (lines 113-116)
“Double staining is the method simultaneously localizing two different antigenic substances in a single section. Immunofluorescent IHC can be applied to FFPE sections to give beautiful double labeling [15, 16]. Chromogenic IHC (immunostaining with enzyme-labeled probes) is also applicable, as described below.”
Additional sentences were also added to lines 132-136.
When the intracellular localization of antigens is overlapped (i.e. both antigens are localized on the plasma membrane), immunofluorescent IHC should be employed. Double immunostaining with enzyme-labeled probes is superior to the immunofluorescent IHC in that stained sections can be preserved permanently.
Line 96_Please, consider replacing “immunostaining” with “chromogenic immunohistochemistry”. The author uses the term “immunostaining” in comparison with “immunofluorescence”. However, “immunostaining” is a very broad designation that include both chromogenic IHC and immunofluorescence.
I accept the term “chromogenic immunostaining” for distinguishing it from the immunofluorescent IHC. I inserted the word “chromogenic”, as far as I recognized the need throughout the text. The title of the present review article was also changed to “Pitfalls and caveats in applying chromogenic immunostaining to histopathological diagnosis”.
Line 109_The author should explain what he/she means for “serial sections” and how this approach compares to multiplexing.
I reworded the term “serial sections” with “consecutive sections”.
I extended the explanation for chromogenic IHC for visualizing double location of peptides in single nerve fibers, as follows.
“Two different neuropeptides in single nerve fibers can be visualized with the following meticulous technique of chromogenic immunostaining [21]. This approach provides an alternative method for double immunofluorescent IHC. Consecutive sectioning is not suitable in this situation, because the same nerve fibers cannot be included in two separate sections at 4 mm thickness.” (lines 145-149)
- Effect of formalin fixation
5.1. Antigenic alterations by formalin fixation
The author should include the specific molecular alterations caused by formalin on the different components of the tissue specimens (DNA, RNA, protein, and other molecules) in a more structured and systematic manner including the potential preanalytical/analytical disadvantages and possible solutions to restore the native molecular properties (e.g. epitote retrieval)
I added words and phrases to the text, as follows.
“Formalin fixation provokes a cross-linkage of protein molecules and fragmentation of DNA and RNA. Sugar moieties suffer a relatively minor damage by formalin fixation. The damage to the antigenicity or the destruction of genomic information on DNA or RNA can be mitigated when buffered or neutralized formalin is used. Noteworthy is that genomic DNA or total RNA extracted from acidic formalin-fixed paraffin-embedded sections can be amplified by polymerase chain reaction (PCR) for DNA analysis and reverse transcription-PCR (RT-PCR) for RNA analysis, when the primer pairs are designed to yield a target genomic sequence as short as 100 base-pair length [24, 25].” (lines 182-189)
Comments on the choice of epitope retrieval are described in lines 205-207 in 5.2 “Selection of fixatives”.
“In brief, the epitope retrieval procedure can break and loosen the protein cross-linkage to expose the masked antigenicity, as described below (section 7.8).”
Lines 137-144_The advantages of using neutral buffered formalin instead of acidic formalin have been widely documented. In that context, the author should refrain from the use of misleading and controversial statements that are based on personal preferences and not supported by scientific evidence.
The following phrases
“The damage to the antigenicity or the destruction of genomic information on DNA or RNA can be mitigated when we use buffered or neutralized formalin, but the morphological preservation for H&E preparations is not good enough for diagnostic purpose when compared with sections fixed in the acidic formalin solution, the author believes so.”
were simply reworded as follows (and personal preferences were deleted).
“Formalin fixation provokes a cross-linkage of protein molecules and fragmentation of DNA and RNA. Sugar moieties suffer a relatively minor damage by formalin fixation. The damage to the antigenicity or the destruction of genomic information on DNA or RNA can be mitigated when buffered or neutralized formalin is used.” (lines 182-185)
Figure 5. Examining the pictures, it appears that FFPE tissue sections yield a much better preservation of the antigens compared to frozen sections. This seems to be in contrast with what is reported in the text. Please, clarify.
Sorry. The explanations in the legend for the left and right panels were reversed.
The legend of Figure 5 should be as follows.
Figure 5. Antigenic deterioration in FFPE sections. Top panels (a&b): CEA in normal gastric mucosa, Bottom panels (c&d): Cytokeratin in normal infantile thymus. Left (a&c): paraformaldehyde-fixed frozen sections, Right (b&d): FFPE sections. Antigenicities of CEA and cytokeratin detected by polyclonal antibodies are evidently weakened after the FFPE process. Arrows indicate Hassall’s corpuscles in the thymic medulla. The nuclei are counterstained with methylgreen.
5.2. Choice of fixatives
A precise definition of epitope retrieval is critical at this point for the reader to understand the concept explained in the following paragraphs.
I added a sentence. (lines 205-207)
In brief, the epitope retrieval procedure can break and loosen the protein cross-linkage to expose the masked antigenicity, as described below (section 7.8).
Line 173_The author should include references to confirm that platelet factor 4 and insulin are stored in lipid-rich granules and verify that their claims about the loss of antigenicity and the osmiophilic properties are adequately supported by the literature.
I cited two references [27, 28]. (lines 220-221)
Line 176_Reference needed.
I cited a new reference [29]. (line 223)
5.3. Use of archival pathology material
Line 187_See comment above regarding the use of acidic formalin.
I added a paragraph at the beginning of the section 5.3. Actually, the author wants to emphasize that the fixation in acidic formalin has some beneficial aspects. (lines 233-234)
“The author would like to emphasize a beneficial aspect of formalin fixation by demonstrating IHC examples of archival pathology specimens.”
- Artifacts by formalin fixation
6.1. Penetration of plasma proteins into the cytoplasm of certain cells: A diffusion artifact
Assuming that the diffusion artifact is not anecdotal evidence, the author should include relevant references demonstrating its development.
The author believes that the diffusion artifact is an observatory fact.
I added a short sentence in lines 291-292.
The findings represent a nonspecific diffusion artifact, which the author proposed in 1984 [32].
6.2. False-positive and false-negative results due to uneven formalin fixation
This subchapter would benefit from the discussion on how homogenous fixation resulting from a timely exposure to formalin is critical to obtain consistent and reproducible IHC results when targeting phosphorylated epitopes.
The author should be also suggesting the implementation of relevant procedures for sample preparation to avoid these artifacts.
Please understand that it is a representative and inevitable fixation artifact when one uses immersion fixation, which is an internationally common way in diagnostic pathology practice. Such an artifact is also encountered in case of paraformaldehyde-fixed frozen sections, as illustrated in Figure 15.
Therefore, I added some phrases, as follows.
When the specimen is immersed in formalin solution, uneven fixation may lead to both false positivity and false negativity [1]. (lines 327-328)
Figure 15 illustrates glutathione peroxidase immunoreactivity seen at the outermost part of the paraformaldehyde-fixed frozen rat liver tissue, probably because of the difficulty in fixing highly soluble cytoplasmic proteins such as glutathione peroxidase. (lines 336-337)
I added a new paragraph in the last part of section 6.2.
“To avoid such fixation artifacts, perfusion fixation can be tried in surgical and autopsy samples; i.e., transarterial perfusion fixation for the brain [36] and intrabronchial infusion fixation for the lung [37]. However, in routine practice of diagnostic pathology, immersion fixation in formalin must be used. The size of the specimen is the key factor: for better and homogeneous fixation, the specimen should be cut as small as possible.” (lines 343-347)
- Tips and pitfalls in immunostaining
7.1. To get low background staining
Consider rephrasing as “how to get low background staining”
I rephrased the section title as “How to get low background staining”.
Lines 309-310
- a) The concentration of the primary or secondary antibodies is too high.
- b) The antibody titer is too low.
Point b seems to contradict point a. Please, clarify.
While antibody concentration and inadequate rinsing represent important factors when troubleshooting background staining, other (arguably more important) variables such as protein blocking step, duration of the antigen retrieval, the addition of detergents to the washing buffers are not taken into consideration by the author. A thorough elaboration of these additional features is critical to provide the reader with a comprehensive overview of the matter.
The items a and b are not contradictory.
The item b was rephrased as follows: b) The titer of the antibody is not sufficient enough.
I added words and an independent paragraph for a comprehensive overview of the matter.
“When immunostaining with a high background noise is encountered, the following three possibilities should be checked first.” (line 370)
“In order to obtain satisfactory and beautiful immunostaining, the additional variables described in the following sections should also be considered: i.e., i) the protein blocking procedure using normal animal serum, ii) the choice of epitope retrieval procedure which has an effect for lowering background staining [38], and iii) the choice of methodology for chromogenic visualization.” (lines 396-400)
In addition, it should be clarified that PBS is not the only washing buffer used in IHC.
Thank you for your advice. I added sentences in the text.
In case an ALP-labeled probe is employed for immunostaining, Tris-buffered saline (TBS) should be selected for the rinsing solution, instead of PBS, because the inorganic phosphate iron inhibits ALP activity [18]. (lines 401-403)
It should be noted that ALP activity is hampered by the presence of inorganic phosphate iron [18], Tris-buffered saline (TBS) should be used for rinsing, instead of PBS. (lines 122-123)
7.2.3. Endogenous protein A and protein G activity
It would be worth adding that, in a similar fashion, an unspecific signal may result from the binding of the Fc fraction of the IgG by the FcR displayed by a variety of inflammatory/immune cells (PMID: 16006601). In addition, nonspecific staining of mast cell granules following immunoperoxidase-based IHC is a well-documented finding that has been attributed to ionic interaction between the F(ab′)2 segments of antibodies and the heparin constituent of the mast cell secretory granules (PMID: 8226102).
I added words and an independent paragraph, as follows.
“These proteins strongly bind the Fc portion of IgG molecules, functioning as IgG-Fc receptors.” (lines 459-460)
“IgG-Fc receptors on mononuclear blood cells can bind to IgG molecules, and nonspecific binding of IgG-type antibodies to inflammatory/immune cells in tissues may occur in frozen sections of lymphoid tissue and cytological preparations. However, such interaction does not cause a problem in FFPE specimens because of inactivation of Fc receptors during preparation of FFPE sections [45]. Fc-binding activity of the immune cells is not retrieved by HIER. Nonspecific staining of mast cell secretory granules in chromogenic immunostaining is attributed to ionic interaction between the F(ab’)2 fragments of IgG and the heparin constituent of the mast cell granules [46].” (lines 475-482)
7.2.4. Endogenous pigments
Please, note that the formation of formalin pigment is greatly reduced by the use of buffered formalin instead of acidic formalin.
Formalin pigments are particularly troublesome in autopsy cases. The formation of formalin pigments is greatly reduced by the use of buffered formalin instead of acidic (unbuffered) formalin. (lines 491-493)
7.3. Staining artifacts: effects of dewatering, contamination of cytokeratin-positive squams (please, consider using scales or debris instead of squams) and insufficient deparaffinization
Consider using the term “dehydration” instead of “dewatering”.
I reworded the term squams as scales and dewatering as drying or dehydration.
The caption of the section 7.3: Staining artifacts: effects of drying, contamination of cytokeratin-positive scales and insufficient deparaffinization
“This is because the antibody molecules are dry-fixed onto the glass slide in the area of dehydration. Sections are occasionally contaminated with cytokeratin-positive scales of epidermal origin, which may cause false-positive judgment.” (lines 519-522)
7.4. Antigenic deterioration due to prolonged conservation of unstained sections
Line 453-453_“Particularly susceptible are nuclear antigens, such as Ki-67 (MIB-1), p53, ER and PgR (Figure 24)” Reference needed.
I added references [48 and 49].
Importantly, the detection of certain antigens is not suitable for the storage at room tem-perature for a long period of time [48]. Particularly susceptible are nuclear antigens, such as Ki-67 (MIB-1), p53, ER and PgR (Figure 24) [49]. (lines 536-538)
7.5. Effects of section-stretching temperature on a hot plate and drying period after cutting sections
The author should specify which type of section (i.e., frozen or paraffin) he is referring to as it is not clear from the text.
All the materials of target are FFPE sections, when not otherwise specified, throughout the text.
I added words, as follows.
Some antigens in FFPE sections are quite susceptible to high section-stretching temperature on a hot plate (Figure 25). (lines 552-553)
Lines 472-473_Assuming that the author is considering paraffin sections, it is hard to believe that 40C for 20 seconds or at 50C for 10 seconds would be enough for a satisfactory stretching as standard paraffin for histology has a melting point of 56 to 58 C.
For effectively stretching paraffin sections, paraffin should be kept unmelted.
I added a sentence: “The temperature at 70C is above the melting temperature of paraffin around 56–58C.” (lines 557-558)
7.8. Pitfalls and caveats in antigen retrieval sequences
Line 562_There are different types of glass slides with enhanced adhesion properties for the sections, an overview of the most common ones would be interesting.
I added a paragraph describing an overview of the coated glass slides.
“To prevent detachment of sections during the pretreatments, the use of coated glass slides is inevitable. There are different types of glass slides with enhanced adhesion properties [60]. Classically, glass slides were coated with ovalbumin or gelatin, but they are not suitable when enzymatic digestion is employed. Because of contamination of avidin molecules, ovalbumin coating should be avoided when biotin-based method is employed. The use of synthetic resins for hydrophobic coating is practical, including 0.1% Neoprene (polychloroprene) in toluene (Okenshoji, Tokyo, Japan), 0.1% poly-L-lysine aqueous solution (molecular weight >150 kDa, Sigma-Aldrich, St. Louis, MI, USA), and 2% (3-aminopropyl)trimethoxysilane in acetone (Sigma-Aldrich). At present, (3-aminopropyl)trimethoxysilane-coated glass slides are most widely utilized.” (lines 655-664)
7.8.1. Proteinase pretreatment
Lines 567-569_”The proteinase pretreatment is effective for retrieving the antigenicity of type 4 collagen (Figure 30) and laminin. Cytokeratins and lymphocyte surface markers may be retrieved with this pretreatment” this statement is misleading, it all depends on the epitope specifically recognized by the antibody in question and the effect of the proteinase on that molecular target. Since there are thousands of antibodies for those targets, it must be clear to the readers that some will work well with proteinase pretreatment while others will not.
That’s right. I added a phrase.
Cytokeratins and lymphocyte surface markers may be retrieved with this pretreatment: some will work well with proteinase digestion, but others will not. (lines 669-671)
7.8.2. Heat-induced epitope retrieval (HIER)
Lines 371-373_It must be clear to the readers that these effects are not target-related but personal observations for the specific antibodies that the author has been using.
A theoretical background of HIER is described as the first paragraph of section 7.8.2.
“Hydrated heating is quite effective for retrieving hidden antigenicities in FFPE sections [3, 6]. The process of HIER may represent a simple removal of formalin fixation-induced protein cross-linkage sterically interfering with the binding of antibodies to linear epitopes in tissue sections [65]. Boenisch has uniquely shown that HIER recovers electrostatic charges on the hydrophilic surfaces of antigens [66].“ (lines 709-713)
I also inserted a sentence in lines 731-733:
The effects of HIER are dependent upon antibodies used, so that one must predetermine if HIER effectively works or not, and what kind of conditions are most suitable for the HIER.
Fig 42, lines 694-698_Even if the large polymeric tail of the original Dako Envision technology is suboptimal for comprehensive antigen detection, the reviewer has used this reagent successfully in the past for the detection of non-mitotic proliferative cells with diverse Ki-67 clones including SolA15, MIB1 and SP6. In addition, it is hard to imagine a situation where only mitotic cells are stained because of the dissolution of the nuclear membrane. Please note that the author is considering FFPE sections where nuclei have been “opened up” by the sectioning itself and the nuclear membrane doesn’t pose any barrier to the penetration of the reagents.
Actually, we vigorously tried to reproduce this phenomenon in order to identify mitotic cells in paraffin sections, but regrettably in vain. It may happen or may not happen even when the same staining condition is used. The sentence was a little bit changed to the following: “When a high molecular weight polymer (EnVision, Agilent/Dako Co) is employed as the secondary reagent, the cytoplasm of the mitotic cells may be stained for Ki-67 while the nuclei of proliferative cells remain unstained (Figure 42).” (lines 797-799)
7.8.3. Other methods for epitope retrieval
Lines 700-702_”Immunoreactivity of b-amyloid protein in the brain is retrieved by soaking in 100% formic acid solution for 5 minutes (Figure 43), while protease treatment and HIER are ineffective”. Again this is not universal. It must be stressed that this is true only for specific antibodies, while others might work with more standard antigen retrieval methods (see for example Cell Signaling Technology β-Amyloid (D3D2N) Mouse mAb #15126).
Thank you for your information.
I changed the paragraph in the following:
Immunoreactivity of b-amyloid protein in the brain and other amyloidogenic substances, including prion protein and prealbumin (transthyretin), in various tissues are retrieved by soaking in 100% formic acid solution for 5 minutes (Figure 43) [70], while protease treatment and HIER are ineffective. In contrast, immunoreactivity of b-amyloid protein can be visualized with a mouse monoclonal antibody D3D2N after conventional hydrated heating. (lines 814-819)
In the legend of Figure 43, a sentence “A polyclonal antibody available from Agilent Co. was used.” was inserted. (lines 828-829)
- How to judge the immunostained results
8.1. False positivity or equivocal negativity
Here the author provides a single example, but it would be much more useful if he can also draw a flow chart on how to recognized false positivity/equivocal negativity for diagnostic purposes.
The caption of section 8 was changed to “The way results are judged”.
I’m very sorry. It’s ideal but impossible for the author to draw a flow chart, because the judgment strategy is different from case to case.
I added a sentence at the last part of the paragraph: The key point for the appropriate judgment is to carefully recognize the intracellular localization of the target antigen, as described in the next part. (lines 841-843)
8.3. Specificity of antibodies
Lines 778-779_The author should be more specific about the identity of the antibody. There are several polyclonal antibodies against myoglobin, it is difficult to believe that all of them show the same non-specific signature.
I agree. Myoglobin staining was shown as an example that I experienced.
I added words, as follows.
“For example, a certain lot of anti-myoglobin rabbit antiserum stains the epidermal keratinocytes, vascular endothelial cells and pancreatic islet cells (Figure 48).” (lines 905-907)
In the legend of Figure 48, a few words were inserted: Rabbit antiserum available from Agilent/Dako may contain natural antibodies against intermediate filament proteins. (line 911)
Lines 791_ Reference/s needed.
I’m so sorry. There is no single scientific paper describing the cross-reactivity of the Ascaris proteins to the smooth muscle, cartilage and some epithelial cells in human tissues, as well as to neuroendocrine tumors. The author described the fact in the textbook written in Japanese, as a reference [1]. (see lines 921-922)
8.4. Specificity of the specific markers
The meaning of this title is unclear. The author is describing examples of uncommon expression patterns of well-established antibodies used in diagnostic pathology. Please, rephrase the title so that it can better represent what is reported in this subsection.
The caption of section 8.4 was changed to “Uncommon expression of antigenicities widely used in diagnostic pathology”. (line 942)
Line 816_Pitfalls of cytokeratin immunostaining are not described in Fig. 51.
No pitfalls are presented here.
I’m sorry. Figures 51 and 52 were interchanged.
The legend caption for Figure 51 was changed to “Diagnostic utility and pitfall of cytokeratin immunostaining”. (line 951)
Line 845 (Fig. 53 caption)_Please, replace “indifferent” with “different”.
The legend caption for Figure 53 was changed to “Expression of specific markers in unrelated cells”. (line 978)
8.5. Nonspecific adsorption of antibodies by certain cells
As mentioned above, it would be worth adding that, in a similar fashion, an unspecific signal may result from the binding of the Fc fraction of the IgG by the FcR displayed by a variety of inflammatory/immune cells (PMID: 16006601). In addition, nonspecific staining of mast cells granules following immunoperoxidase-based IHC is a well-documented finding that has been attributed to ionic interaction between the F(ab’)2 segments of antibodies and the heparin constituent of the mast cell secretory granules (PMID: 8226102).
The first paragraph in section 8.5 was considerably reworded as follows.
“In FFPE sections, tissue mast cells [87], neuroendocrine cells (particularly gastrin cells) [88], parietal cells of the gastric oxyntic mucosa, and HBs antigen-positive ground-glass hepatocytes in HBV carriers [89] may adsorb antibodies nonspecifically. Nonspecific staining of mast cell granules is attributed to ionic interaction between F(ab)’2 segments of IgG and heparin constituent of the mast cell secretory granules [45]. In frozen sections, non-specific signals may be seen on the plasma membrane of inflammatory or immune cells, resulting from the binding of the Fc fraction of IgG through their Fc receptors [90]. However, such Fc receptor-mediated nonspecific staining is scarcely encountered in FFPE sections. Representative examples of nonspecific adsorption of antibodies are demonstrated in Figure 54.” (lines 986-995)
8.6. Positive and negative controls
This subsection would benefit from the definition of biological and technical controls. Biological controls are accomplished by including tissues that are known to express (or not to express) the target protein (e.g., for insulin IHC one should include pancreas as biologically positive control and tonsil as biologically negative control). Technical controls are serial sections of the biopsy under diagnosis that, depending on the primary antibody used, are incubated with normal animal serum (for polyclonal antibodies) or isotype-matched immunoglobulins (for monoclonal antibodies).
According to your advice, I inserted a new paragraph at the beginning of section 8.6.
Biological and technical controls can be used for immunostaining. Biological controls are accomplished by including tissues that are known to express or not to express the target protein. For example, for gastrin immunostaining one should include gastric antral mucosa as a biologically positive control and colonic mucosa as a biologically negative control. Technical controls employ consecutive sections of the specimen under diagnosis that are incubated with normal animal serum for polyclonal antibodies or isotype-matched immunoglobulins for monoclonal antibodies. (lines 1004-1010)
11.1. Use of antisera against pathogens showing wide cross-reactivity
In addition to citing his/her own article, the author should add additional references demonstrating the suitability of this technique not only for bacterial organisms but also for fungi and protozoa. Please, see here a couple of examples: PMID: 11301538, PMID: 9448206, PMID: 9448206, PMID: 9448206.
Thanks a lot. I cited the two references, and reworded the paragraph in the following:
“Bacterial or fungal antigens have been visualized with antisera against BCG showing wide cross-reactivity [102, 103]. Based on the author’s own experience, commercially available rabbit antisera against BCG, Bacillus cereus, Treponema pallidum and Escherichia coli can be utilized for this purpose: The immunostaining is valuable for screening of bacterial infection in FFPE sections [101].” (lines 1143-1148)
May 21, 2021
Yutaka Tsutsumi, M.D.
Diagnostic Pathology Clinic, Pathos Tsutsumi
clinic director
1551-1 Sankichi-ato, Yawase, Inazawa, Aichi 492-8342, Japan
email: pathos223@kind.ocn.ne.jp
phone: +81-587-96-7088, fascimile: +81-587-96-7098
mobile: +81-80-6641-9802
Reviewer 2 Report
Author of the present article has a great experience in the use of immunohistochemical methods, which he generously shares with the readers. Although it is mainly addressed to those who specialize in medical diagnostics, the article will be very useful for a wider range of readers involved in immunolocalization. Still I have some advices, which should be addressed to make the text clearer. In general, figures should be marked making it easier for less experienced readers to understand what is meant. Arrows inserted in the present figures are too broad making it frequently unclear, where they point. Some terms and abbreviations should be deciphered. As far as I could understand, most of presented data were obtained by the authors. I think that this should be emphasized
- I am not sure that references are arranged in the proper way. I thought that ciphers should be placed inside the brackets according to the rules of the journal.
- Figure 1. Its legend says that “Cytokeratin immunoreactivity with a monoclonal antibody CAM5.2 clearly illustrates their distribution (arrows).” – but arrow are present only on the left figure, while immunostaining is on the right.
- Lines 57-59. “The reasons are as follows: a) endogenous biotin activity in mitochondria is retrieved by the pretreatment with heat-induced epitope retrieval (HIER)” – the need of HIER is clearly justified below in this article. But here it looks unclear. I think it would be nice to promise here that clearer explanation is to follow.
- Figure legends should be self-sufficient and it would be better to cipher such abbreviations as CSA (Figure 2) in the figure legend although it is deciphered in the text.
- Line 189. I am not sure that term “genome” is proper in this case. Genome means the complete set of genes or genetic material present in a cell or organism.
- Figure 14. “The poorly fixed central part”. I advise to somehow mark this central part.
- Figure 17. Again I advise to insert certain marks indicating certain structures to make clearer the statement that “With 1:400 diluted antiserum neoplastic hepatocytes, bile ducts and arterial wall reveal nonspecific staining. At an appropriate dilution (1:3,200), positive signals of AFP are seen only in the hepatoma cells.”
- Figure 20. I am not sure that it is easy to see what authors wanted to show. Some more detailed comments with references to exact figure are necessary.
- Figure 21. I think it would be better to indicate DAB staining and pigments with arrows.
- Figure 22. I think that indication of “true positive” and “false-negative” rounded areas on the figure would be very useful.
- Figure 23. “round-shaped negative zones” promised in the text (line440) should be indicated on the sections
- Figure 29. “Paradoxical weakening of the antigenicity… when the primary antibody concentration is high”- but for me the figure with high concentration of antibodies looks more intensively stained.
- Line 623 “chelation of calcium iron by” – is not calcium AND iron meant”
- Figure 34. Nuclei should be indicated for clarity.
- Line 682. “When one employs a high molecular weight polymer reagent (EnVision, Agilent/Dako Co) as the secondary probe.” - This “polymer reagents” and “probe” appears a bit abruptly. I think some explanations are necessary to clarify when they are used.
Author Response
Point-by-point response to Reviewer 2’s comments
Cells-1219134
Author of the present article has a great experience in the use of immunohistochemical methods, which he generously shares with the readers. Although it is mainly addressed to those who specialize in medical diagnostics, the article will be very useful for a wider range of readers involved in immunolocalization. Still I have some advices, which should be addressed to make the text clearer.
Thank you very much for your helpful advice and suggestions.
In general, figures should be marked making it easier for less experienced readers to understand what is meant. Arrows inserted in the present figures are too broad making it frequently unclear, where they point. Some terms and abbreviations should be deciphered. As far as I could understand, most of presented data were obtained by the authors. I think that this should be emphasized.
I inserted arrows or arrowheads, as much as I can, in order to making for less experienced readers to understand what is meant. The abbreviations used in the Figure Legend are commonly used in the text, I believe. Some were deciphered. All the presented data were obtained by the author and his colleagues, so that short comments are inserted in the last sentence of the Introduction (lines 32-34).
“It should be emphasized that all the data presented below were obtained by the author and author’s colleagues.”
- I am not sure that references are arranged in the proper way. I thought that ciphers should be placed inside the brackets according to the rules of the journal.
References cited were placed inside the brackets. The guideline says the references may be in any style, provided that the author uses the consistent formatting throughout. I arranged the references in a traditional way, I believe.
- Figure 1. Its legend says that “Cytokeratin immunoreactivity with a monoclonal antibody CAM5.2 clearly illustrates their distribution (arrows).” – but arrows are present only on the left figure, while immunostaining is on the right.
Arrows were inserted in both (left and right) panels.
- Lines 57-59. “The reasons are as follows: a) endogenous biotin activity in mitochondria is retrieved by the pretreatment with heat-induced epitope retrieval (HIER)” – the need of HIER is clearly justified below in this article. But here it looks unclear. I think it would be nice to promise here that clearer explanation is to follow.
I inserted a short explanation as (for clear explanation, see below, section 7.8.2), in lines 81-82: “-----endogenous biotin activity in mitochondria is retrieved by the pretreatment with heat-induced epitope retrieval (HIER) [13] (for clear explanation, see below, section 7.8.2) , -----"
- Figure legends should be self-sufficient and it would be better to cipher such abbreviations as CSA (Figure 2) in the figure legend although it is deciphered in the text.
I ciphered CSA as catalyzed signal amplification in Figure 2 legend.
- Line 189. I am not sure that term “genome” is proper in this case. Genome means the complete set of genes or genetic material present in a cell or organism.
The term “genome” was changed to “a target genomic sequence”. (line 189)
- Figure 14. “The poorly fixed central part”. I advise to somehow mark this central part.
I inserted the words, as follows: The poorly fixed central part (representing a lower half of the panels) is false-negative for both vimentin and CD20. (line 350)
- Figure 17. Again I advise to insert certain marks indicating certain structures to make clearer the statement that “With 1:400 diluted antiserum non-neoplastic hepatocytes, bile ducts and arterial wall reveal nonspecific staining. At an appropriate dilution (1:3,200), positive signals of AFP are seen only in the hepatoma cells.”
I added arrows and arrowheads in Figure 17, to indicate nonspecific staining in the non-neoplastic hepatocytes, bile ducts and arterial wall by arrowheads and specific staining in the hepatoma cells by arrows.
- Figure 20. I am not sure that it is easy to see what authors wanted to show. Some more detailed comments with references to exact figure are necessary.
I added some words to the Figure 20 legend. MRSA binds rabbit and mouse IgG through heat-retrieved Fc-binding activity by protein A in panels c and d. F(ab)’2 fragments, lacking the Fc portion of IgG, show no binding activity as shown in panel b. The reference #43 was cited in the text.
- Figure 21. I think it would be better to indicate DAB staining and pigments with arrows.
In Figure 21, specific DAB reactions, colored in brown, are indicated by arrows (a) and arrowheads (b).
- Figure 22. I think that indication of “true positive” and “false-negative” rounded areas on the figure would be very useful.
No true positivity of an entamoebic antigen detected by a monoclonal antibody EHK153 is seen here. I added a sentence “The entamoebic antigen must be judged as negative in this case” in the Figure 22 legend. (lines 513-514)
- Figure 23. “round-shaped negative zones” promised in the text (line440) should be indicated on the sections.
Round-shaped negative zones are evident in the background of diffuse positivity of alpha-SMA in the smooth muscle neoplasm in panel c of Figure 23. Dried area (a), contaminated scales (b) are also easily recognized. I believe the arrows are unnecessary in these situations.
- Figure 29. “Paradoxical weakening of the antigenicity… when the primary antibody concentration is high”- but for me the figure with high concentration of antibodies looks more intensively stained.
In panel b immunostained with a high concentration of antibody, plasma membrane reactivity of CD4 is blurred in association with high background staining. I added words “Equivocal CD4 staining on the plasma membrane with a high background happens”. (line 637)
The next sentence was also added: The background staining was caused by the artificial diffusion of the membrane antigen into the cytoplasm. (lines 638-639)
- Line 623 “chelation of calcium iron by” – is not calcium AND iron meant”
Calcium irons (Ca2+) should be chelated with citrate or EDTA. I don’t mean calcium AND iron.
- Figure 34. Nuclei should be indicated for clarity.
Instead of indicating the nuclei, the p53-negative adenoma component is indicated with asterisks.
- Line 682. “When one employs a high molecular weight polymer reagent (EnVision, Agilent/Dako Co) as the secondary probe.” - This “polymer reagents” and “probe” appears a bit abruptly. I think some explanations are necessary to clarify when they are used.
I reworded as follows: “When a high molecular weight polymer (EnVision, Agilent/Dako Co) is employed as the secondary reagent,” (lines. 797-798)
I believe Agilent/Dako Co. has developed a lower molecular weight polymer reagent (EnVision Plus or EnVision Flex), in order to avoid such artifactual negative staining as described herein.
May 21, 2021
Yutaka Tsutsumi, M.D.
Diagnostic Pathology Clinic, Pathos Tsutsumi
clinic director
1551-1 Sankichi-ato, Yawase, Inazawa, Aichi 492-8342, Japan
email: pathos223@kind.ocn.ne.jp
phone: +81-587-96-7088, fascimile: +81-587-96-7098
mobile: +81-80-6641-9802

Reviewer 3 Report
The work entitled "Pitfalls and caveats in applying immunostaining to histopathological diagnosis" presents a general review of the available techniques for immunohistochemical staining and gives some interesting tips for troubleshoting during sample staining. Although the information is not new or innovative, the summary offered by the author, is thankfull, consistent and interesting.
The article includes huge amount a variety of microscopical images and in my opinion in suitable for publication once some relevant shortcomings are corrected.
In my opinion one of the most relevant problems is the use of English. In general, although the article can be undestood, there is a excessive use of personal reference.
For example LINE 73 "Suppose you have a 100 µL aliquot of an antibody working at a 1:1,000 dilution", can be written as "In case there is a 100 µL aliquot of an antibody working at a 1:1,000 dilution". Or LINE 141 "can be mitigated when we use buffered or neutralized formalin" can be written as "can be mitigated when buffered or neutralized formalin is used". The use of passive sentences is recommended in many cases troughout the text.
On the other hand, a more consistent explanation and information in the figure captions copuld also helpful. I would recommend the authors to the use of letter in all figures (as in Figure 8 or 20) instead of "left, center, right" so all the figures are consistent. Moreover, in the figure captions I wouls start indicating the stained molecule/pathology and then give the description/information of the image. In many cases the use of asteriscs or arrows (as in Figures 1, 2 or 5) is also welcome. Finally, it would be good to have the magnification or scale bars indicated somewhere in the figures.
Finally, in some cases, the addition of a reference is also welcome.
MINOR POINTS:
LINE 12: "deepened knowledge of immunohistochemical markers". Please give more information regarding this point.
LINE 19 KEYWORDS: I would put "Immunostaining" as first KEYWORD.
LINE 24: Is "measures" or "tools"?
LINE 32: Replace "chosen" for "selected"
LINE 37: Put "fopr example" before "breast cancer"
LINE 37: A reference is welcome after "Figure 1"
LINE 39: A reference is welcome.
LINE 51: A reference is welcome.
LINE 57: Replace "as follows" for "the following"
LINE 61: Why? A reference is welcome.
LINES 68-69: Rewrite the sentence
LINE 73: Rewrite the sentence
LINE 109: Why "serial sectioning is not suitable in this situation"?
LINE 110: Why "raised in rabbit are used here"?
LINE 120: A reference is welcome. After "activity"
LINE 123 (and others): "It is of note" is correct? I would use "it is noteworthy" or somethin similar.
LINE 144: A reference is welcome, not only "the author believes so".
LINES 167-169: Rewrite this sentence. Not easy to undertand.
LINE 176: A reference is welcome.
LINES 200-204: Rewrite this sentence. Not easy to undertand.
LINE 211: Replace "before" for "ago"
LINE 244: Put the fulkl name at leats the first time new acronyms are used Reed-Stemberg first time, then RS.
LINE 248: A reference is welcome. At leats for kenatinocytes.
LINES 253-254: Put sentence in passive.
LINE 280: What is the meaning of this? Plase give a more detailed explanation.
LINE 305 and 312: What want the author to say with an "artistic" immunohistoichemical staining?
LINES 334-338: I would move this part form here and write in line 344.
LINE 363: Remove "any longer"
LINE 410: Is the reference for Figure 22 right? Or should be refered to Figure 23?
LINES 428-429: Rewrite this sentence. Not easy to undertand.
LINE 442: Is the reference for Figure 23 right? Please check.
LINE 472: A reference is welcome. After "weakened".
LINE 500: Please specify which "routine decalcification" is suitable or not.
LINE 520: A reference is welcome. After "staining".
LINE 521: A reference is welcome.
LINE 541: Can an image of the example of recovery added to Figure 29?
LINE 552: A reference is welcome. After "strongly".
LINE 598: A reference is welcome.
FIGURES 34, 35, 36, 37, 38, 40: Please add time and/or temperature of the treatment.
LINE 653: A reference is welcome.
LINE 672: A reference is welcome. After Figure 40.
LINE 707: Any reason or explanation for this statement?
LINE 715: Replace "how to judge the results" for "the way results are judged".
LINE 825: Replace "cytoleratin" for "cytokeratin".
LINES 882-883: What is the meaning of this sentence? Please specify.
SECTION 11.2: I find this poitn a little bit out of the scope of the work. Maybe it can be removed.
LINE 1103: Rewrite this sentence. Not easy to undertand.
Author Response
Point-by-point Responses to Reviewer 3’s comments
Cells-1219134
The work entitled "Pitfalls and caveats in applying immunostaining to histopathological diagnosis" presents a general review of the available techniques for immunohistochemical staining and gives some interesting tips for troubleshooting during sample staining. Although the information is not new or innovative, the summary offered by the author, is thankful, consistent and interesting.
The article includes huge amount a variety of microscopical images and in my opinion in suitable for publication once some relevant shortcomings are corrected.
In my opinion one of the most relevant problems is the use of English. In general, although the article can be understood, there is an excessive use of personal reference.
Thank you very much for your helpful and precious advice and suggestions.
I’m proud that I’m a Japanese pioneer of diagnostic pathologists who effectively utilize chromogenic immunostaining in FFPE sections. My colleagues and I have published a number of original and review papers on the clinicopathological applications and technical pitfalls and caveats in diagnostic IHC. I believe some are very unique, although some are written in Japanese. This is my comprehensive review article on the experience of diagnostic IHC for more than 40 years. Therefore, I sincerely hope you to accept the frequent citation of our own references.
For example, LINE 73 "Suppose you have a 100 µL aliquot of an antibody working at a 1:1,000 dilution", can be written as "In case there is a 100 µL aliquot of an antibody working at a 1:1,000 dilution". Or LINE 141 "can be mitigated when we use buffered or neutralized formalin" can be written as "can be mitigated when buffered or neutralized formalin is used". The use of passive sentences is recommended in many cases throughout the text.
I changed sentences in a passive form, as much as I could.
In case there is a 100 mL aliquot of an antibody working at a 1:1,000 dilution, a recommended management of the antibody is as follows. (lines 97-98)
The damage to the antigenicity or the destruction of genomic information on DNA or RNA can be mitigated when buffered or neutralized formalin is used. (lines 184-185)
Others are highlighted in red in the text. Examples include:
Measures and tools against technical artifacts and appropriate trouble-shooting tips are needed. (lines 29-30)
----. when the CSA-II system is employed [59]. (line 623)
When a high molecular weight polymer (EnVision, Agilent/Dako Co) is employed, --- (line 797)
Cases of cytokeratin-positive malignant lymphoma, glioma, sarcoma or melanoma may be encountered. (lines 1032-1033)
Or the specimen is left in tap water (line 1071)
---- can be utilized for this purpose. (lines 1146-1147)
On the other hand, a more consistent explanation and information in the figure captions could also helpful. I would recommend the authors to the use of letter in all figures (as in Figure 8 or 20) instead of "left, center, right" so all the figures are consistent. Moreover, in the figure captions I would start indicating the stained molecule/pathology and then give the description/information of the image. In many cases the use of asterisks or arrows (as in Figures 1, 2 or 5) is also welcome. Finally, it would be good to have the magnification or scale bars indicated somewhere in the figures.
I changed to show a, b, c, d, etc., instead of indicating as left, center and right, in all the figures. I changed the sequence of explanation, as much as I could. I frequently used arrows, arrowheads and asterisks for indicating the site of positivity or key microscopic structures. Regrettably enough, it’s very hard for me to have magnifications or scale bars in the respective figures, because many of the figures are years-old and some have no exact information on the magnification.
Finally, in some cases, the addition of a reference is also welcome.
I placed additional references as much as I could. The total number of references was increased to 109 (originally. the number was 72).
MINOR POINTS:
LINE 12: "deepened knowledge of immunohistochemical markers". Please give more information regarding this point.
The phrase “deepened knowledge of immunohistochemical markers” was changed to “advanced applicability of immunohistochemical markers”. (line 15)
LINE 19 KEYWORDS: I would put "Immunostaining" as first KEYWORD.
In the keywords, I included “Chromogenic immunostaining” and put it as the first keyword. The keywords, however, were arranged in an alphabetical manner, but by their importance.
LINE 24: Is "measures" or "tools"?
Thank you very much. I accept both.
Measures and tools against technical artifacts and appropriate trouble-shooting tips are needed. (lines 29-30)
LINE 32: Replace "chosen" for "selected",
I replaced “chosen” for “selected”. (line 43)
LINE 37: Put "for example" before "breast cancer"
I put for example after breast cancer. (line 56)
Instead, I deleted “For example” from the preceding sentence “Cytokeratins are useful to show ---“. (line 55)
LINE 37: A reference is welcome after "Figure 1"
Reference 9 was cited after the words Figure 1. (line 56)
LINE 39: A reference is welcome.
Reference 10 was cited. (line 58)
LINE 51: A reference is welcome.
Reference 11 was cited. (line 76)
Reference 12 was also cited. (line 78)
LINE 57: Replace "as follows" for "the following"
I replaced “as follows” for “the following”. (line 80)
LINE 61: Why? A reference is welcome.
Reference 12 was cited. (line 84)
LINES 68-69: Rewrite the sentence
I rewrote the sentence.
“Antigenic detection with ultra-high sensitivity techniques such as CSA-II is not necessarily needed, because of the augmentation of background staining and difficulty in maintaining the highly diluted antibodies in refrigerators or freezers.” (lines 90-92)
LINE 73: Rewrite the sentence
I rewrote the sentence.
In case there is a 100 mL aliquot of an antibody working at a 1:1,000 dilution, a recommended management of the antibody is as follows. (lines 97-98)
LINE 109: Why "serial sectioning is not suitable in this situation"?
I added the sentence.
Consecutive sectioning is not suitable in this situation, because the same nerve fibers cannot be included in two separate sections at 4 mm thickness. (lines 147-149)
LINE 110: Why "raised in rabbit are used here"?
It’s an example using antibodies raised in the same animal like a rabbit.
I added “For example” to the sentence: “For example, antibodies raised in rabbit are used here.” (line149)
LINE 120: A reference is welcome. After "activity"
Reference 22 was cited. (line 159)
LINE 123 (and others): "It is of note" is correct? I would use "it is noteworthy" or something similar.
I rephrased “It is of note” with “It is noteworthy”. (line 162)
In other parts, I abandon the use “It is of note that ---, or Of note is that ---“:
“It is noteworthy that ---" or “Noteworthy is that ---" (lines 167, 186, 215, 692, 734, and 904)
LINE 144: A reference is welcome, not only "the author believes so".
According to the comments by another reviewer, the sentence was rephrased in the following (the personal experience or opinion was omitted):
The damage to the antigenicity or the destruction of genomic information on DNA or RNA can be mitigated when buffered or neutralized formalin is used. (lines 184-185)
LINES 167-169: Rewrite this sentence. Not easy to understand.
I rewrote the sentence in the following:
Disadvantages of this method are as follows. A significant tissue shrinkage may cause the difficulty in preparing sections, and the shrunken nuclei tend to stain with hematoxylin to be seen hyperchromatic. (lines 212-214)
LINE 176: A reference is welcome.
A total of three references [27-29] were added to two sentences.
“In fact, the a-granules of the platelets and b-granules of insulin cells are rich in lipid com-ponents [27, 28]. In contrast, somatostatin can be visualized in ethanol-fixed paraffin sections. It is known that somatostatin granules show low electron density in ultra-structural appearance [29].” (lines 219-223)
LINES 200-204: Rewrite this sentence. Not easy to understand.
I rewrote the sentence in the following:
The author visited the Gordon Museum of Pathology at Guy’s Hospital in London, UK, and he had a chance to have unstained glass slides of autopsied Hodgkin’s lymphoma, which was sampled by Dr. Thomas Hodgkin himself 170 years ago. (lines 250-252)
LINE 211: Replace "before" for "ago"
I replaced “before” for “ago”, as described above. (lines 252 and 262)
LINE 244: Put the functional name at least the first time new acronyms are used Reed-Sternberg first time, then RS.
Actually, RS cells were already cited in line 257. However, RS cells were spelled out in line 296, here again.
LINE 248: A reference is welcome. At least for keratinocytes.
Reference 34 was cited. (line 301)
LINES 253-254: Put sentence in passive.
I changed the sentence in a passive style.
This is the reason why albumin should be selected for an indifferent control, when IgG and alpha-1 antitrypsin are immunostained in FFPE sections. (lines 306-307)
LINE 280: What is the meaning of this? Please give a more detailed explanation.
Figure 15 illustrates glutathione peroxidase immunoreactivity seen at the outermost part of the paraformaldehyde-fixed frozen rat liver tissue, probably because of the difficulty in fixing highly soluble cytoplasmic proteins such as glutathione peroxidase. (lines 334-337)
LINE 305 and 312: What want the author to say with an "artistic" immunohistochemical staining?
I deleted the sentences regarding “artistic IHC”.
the staining must be artistic, the author believes. (line 368)
The words “artistic immunostaining” were reworded as “beautiful immunostaining”. (line 374)
LINES 334-338: I would move this part form here and write in line 344.
The sentences “Eosinophils and neutrophils have a strong peroxidase activity, and hemoglobin in red cells reveals pseudoperoxidase activity. Peroxidase activity in macrophages, platelets and epithelial cells (salivary gland, mammary gland, thyroid, and renal tubules) is completely inactivated during FFPE preparations.” were moved to the last part of the paragraph. (lines 413-417)
LINE 363: Remove "any longer"
I deleted the words “any longer” from the sentence. (line 438)
LINE 410: Is the reference for Figure 22 right? Or should be referred to Figure 23?
I’m sorry. Figures 22 and 23 were erroneously reversed.
LINES 428-429: Rewrite this sentence. Not easy to understand.
I rewrote the sentences in the following.
“As another trouble-shooting tip, ALP-labeled secondary reagent should be selected, instead of HRP-labeled probe [20]. The ALP reaction products with azo dyes can be visualized in red or blue.” (lines 497-499)
LINE 442: Is the reference for Figure 23 right? Please check.
I’m sorry. Figures 22 and 23 were erroneously reversed.
LINE 472: A reference is welcome. After "weakened".
Reference 53 was cited. (line 560)
LINE 500: Please specify which "routine decalcification" is suitable or not.
I described the method of decalcification that Dr. Mukai’s group described.
“In contrast, Mukai, et al. described that tissues routinely decalcified with EDTA, formic acid, nitric acid or Plank-Rychlo solution could be used for immunostaining without significant loss of immunoreactivity [55].” (lines 588-590)
LINE 520: A reference is welcome. After "staining".
Reference 57 was cited. (line 608)
LINE 521: A reference is welcome.
No appropriate reference was found. Instead, a comment was inserted: (see Figure 29). (lines 609-610)
I added new sentences in lines 629-630 and in the legend of Figure 29 (lines 638-639)
The background staining was increased, principally due to artificial diffusion of the membrane antigen into the cytoplasm. (lines 629-930)
The background staining was caused by the artificial diffusion of the membrane antigen into the cytoplasm. (lines 638-639)
LINE 541: Can an image of the example of recovery added to Figure 29?
The left-sided panel (a) serves as a kind of recovery sample with membrane immunoreactivity of CD4.
LINE 552: A reference is welcome. After "strongly".
The sentence was rephrased as follows.
---, and normal mammary ducts and normal gastric foveolar cells also occasionally tend to be stained moderately. (lines 643-644)
No appropriate reference was found.
LINE 598: A reference is welcome.
The markers were listed up in our textbook [1], but written in Japanese. (see line 701)
FIGURES 34, 35, 36, 37, 38, 40: Please add time and/or temperature of the treatment.
I described the method and temperature. The treating time was 10 minutes.
autoclaving at 121C (Figures 34 and 35)
pressure pan heating at 121C (Figures 36, 37, 38 and 40)
LINE 653: A reference is welcome.
No appropriate reference was found.
I inserted the word “theoretically” to the sentence below.
This paradoxical phenomenon can be explained theoretically in the following. (lines 762-763)
LINE 672: A reference is welcome. After Figure 40.
The markers were described in our textbook [1]. I added the phrase “according to the author’s experience”.
“These include BM-1 (a marker of the myeloid precursors), neutrophil elastase (NP57), von Willebrand factor, NSE and GST (a, m and p) [1], according to the author’s experience.” (lines 785-787)
LINE 707: Any reason or explanation for this statement?
No clear reason was described. It was a kind of experience probably resulting from trials and errors. (line 824)
LINE 715: Replace "how to judge the results" for "the way results are judged".
The caption of section 8 was changed from "How to judge the results" to "The way results are judged". (line 830)
LINE 825: Replace "cytoleratin" for "cytokeratin".
The word “cytokeratin” was replaced. (line 960)
LINES 882-883: What is the meaning of this sentence? Please specify.
I added some words in the following.
“It should be significant clinicopathologically when immunohistochemical results are different from H&E-based expectation.” (lines 1029-1030)
SECTION 11.2: I find this point a little bit out of the scope of the work. Maybe it can be removed.
I strongly protest against this opinion. It is very important for pathologists and scientists in developing countries where infectious diseases are common and the commercial antibodies are not easily available. This is my original idea and practically useful to detect pathogens in FFPE sections, I believe. I dare to keep this section, particularly for practicians in the developing countries.
LINE 1103: Rewrite this sentence. Not easy to understand.
I rewrote the sentence.
Nowadays, it is clearly expected that chromogenic immunostaining is beautiful and specific. (line 1259-1260)
May 21, 2021
Yutaka Tsutsumi, M.D.
Diagnostic Pathology Clinic, Pathos Tsutsumi
clinic director
1551-1 Sankichi-ato, Yawase, Inazawa, Aichi 492-8342, Japan
email: pathos223@kind.ocn.ne.jp
phone: +81-587-96-7088, fascimile: +81-587-96-7098
mobile: +81-80-6641-9802

Round 2
Reviewer 1 Report
The author has performed a thorough revision of the original manuscript addressing most of the comments raised during the review process.
The main points of criticism raised during the first round of review have been addressed satisfactorily. However, there are a few aspects that still need to be reviewed before considering the manuscript suitable for publication.
General comments:
As previously suggested, the author should really do an effort to narrow the scope of the manuscript on the application of chromogenic IHC on FFPE sections for diagnostic purposes omitting any aspects (cytology, frozen sections, etc,) that are not pertinent to that. It must be understood that this manuscript currently includes 55 pages which an exceptional “size”, even for a review paper. The excessive length of the manuscript might undermine the full comprehension of its content. Therefore, a more focused and linear narrative approach would facilitate the fruition of the main concepts herein contained.
It is understood that the author draws from his/her vast personal experience to compile most of the sections of this manuscript. However, any generalizations based on purely anecdotal evidence should be avoided or clearly reported as “personal experience” not to confound the reader.
Specific comments
Line 14_”The needs are prompted by technical development” It is not clear what “needs” are considered here. The introduction of this sentence is confusing and should be rephrased.
Lines 44-45_As these are not universal markers of malignancy, please, specify pertinent examples where either p53 or Ki-67 are indicative of poor prognosis.
Lines 76-77_HRP-based tiramyde amplification systems are also commonly used to boost the immunoreactivity.
Line 109_Please, replace “coloring” with “chromogenic”.
Lines 127-129_The statement is unclear unless the author explains the reasons.
Line 326_6.2. False positive and false negative results due to uneven formalin fixation. As already suggested, the author should also include the example of phosphorylated epitopes.
Line 715_Please, include TRIS-EDTA pH 9 which is arguably the most commonly used buffer.
Author Response
Point-by-point responses
Cells-1219134-R2
Reviewer 1
The author has performed a thorough revision of the original manuscript addressing most of the comments raised during the review process.
The main points of criticism raised during the first round of review have been addressed satisfactorily. However, there are a few aspects that still need to be reviewed before considering the manuscript suitable for publication.
General comments:
As previously suggested, the author should really do an effort to narrow the scope of the manuscript on the application of chromogenic IHC on FFPE sections for diagnostic purposes omitting any aspects (cytology, frozen sections, etc,) that are not pertinent to that. It must be understood that this manuscript currently includes 55 pages which an exceptional “size”, even for a review paper. The excessive length of the manuscript might undermine the full comprehension of its content. Therefore, a more focused and linear narrative approach would facilitate the fruition of the main concepts herein contained.
It is understood that the author draws from his/her vast personal experience to compile most of the sections of this manuscript. However, any generalizations based on purely anecdotal evidence should be avoided or clearly reported as “personal experience” not to confound the reader.
As a practical and veteran diagnostic pathologist, I am quite sure that experience itself provides the most important facts and findings for better diagnostic services. When we experience a strange or unusual phenomenon in chromogenic immunostaining, we pursue the cause of the phenomenon, as I frequently cite in this review article. I believe this is why reviewer 1 has an impression that the review is too dependent upon the author’s personal experience. Some of the phenomena were reported in papers written in Japanese, and I sincerely want to inform my international colleagues of what is practical and useful in performing chromogenic immunostaining for patients exhibiting these phenomena. This is the true reason for writing the present review article.
I have tried to include chromogenic immunostaining in cytology preparations and frozen sections as references for immunostaining using FFPE sections. Please refer to Figures 3, 5, 6, 15, 18, 37, 57, and 58. The addition of further wording on immunostaining for cytology preparations and frozen sections in the text should be avoided, I believe.
I tried to extinguish the hint of “personal experience” as much as possible, but I am not sure that I have satisfied the reviewer’s criticism, and I am very sorry for this limitation. In fact, I deleted the text suggesting personal experience as much as I could as below, with these line numbers referring to the revised manuscript:
According to the author’s experience, etc. (lines 56, 797, 1155)
The author strongly recommends pressure pan cooking,…
--- Pressure pan cooking is strongly recommended,… (lines 725-726)
We happen to…,
--- Occasionally, false negativity of nuclear antigens such as --- can be experienced. (lines 767-769)
Finally, the manuscript was checked by a native English speaker and simultaneous interpreter, Prof. Tina Tajima at the St. Marianna University School of Medicine, Kawasaki, Japan.
Newly revised portions are highlighted in blue font in the text.
Specific comments
- Line 14_”The needs are prompted by technical development” It is not clear what “needs” are considered here. The introduction of this sentence is confusing and should be rephrased.
I added the following words:
The needs for appropriate and reproducible methods of immunostaining are prompted by --- (lines 14-15)
- Lines 44-45_As these are not universal markers of malignancy, please, specify pertinent examples where either p53 or Ki-67 are indicative of poor prognosis.
There are so many papers published on this. I specified the references [9–11] describing p53 and Ki-67 as prognostic factors in three kinds of malignancies occurring in the oral cavity, colon, and breast.
“The degree of malignancy can be analyzed by immunostaining for p53 and Ki-67 (MIB-1), a cell proliferation marker, in a variety of malignancies including carcinomas of the oral cavity, colon and breast [9–11].” (lines 43-46)
I added three new references.
- Motta Rda R, Zettler CG, Cambruzzi EZ, et al. Ki-67 and p53 correlation prognostic value in squamous cell carcinomas of the oral cavity and tongue. Braz J Otorhinolaryngol. 2009; 75(4): 544–549. doi: 10.1590/S1808-86942009000400013.
- Lumachi F, Orlando R, Marino F, et al. Expression of p53 and Ki-67 as prognostic factors for survival of men with colorectal cancer. Anticancer Res. 2012; 32: 3965–3967.
- Ding L, Zhang Z, Xu Y, Zhang Y. Comparative study of Her-2, p53, Ki-67 expression and clinicopathological characteristics of breast cancer in a cohort of northern China female patients. Bioengineered. 2017; 8(4): 383–392.
- Lines 76-77_HRP-based tyramyde amplification systems are also commonly used to boost the immunoreactivity.
Actually, FITC tyramide is used to amplify chromogenic signal amplification but not for immunofluorescence. Here, FITC is used as a haptenic antigen, as you know.
I added the following text.
“When necessary, catalyzed signal amplification (CSA)-II using fluorescein isothiocyanate (FITC)-labeled tyramide (Agilent/Dako) can be applied in which the anti-FITC antibody mediates amplification of the chromogenic signal (Figure 2) [15].” (lines 77-79)
I added some words in the legend of Figure 2 as follows:
“The immunoreactivity is significantly enhanced by the CSA-II method using FITC tyramide as an amplifier.” (lines 90-91)
- Line 109_Please, replace “coloring” with “chromogenic”.
I changed the word “coloring” to “chromogenic”. (line 113)
I also changed the wording similarly in lines 129-130 and line 494.
- Lines 127-129_The statement is unclear unless the author explains the reasons.
I added the reason in the following sentence. (lines 132-134)
Under a fluorescent microscope, it may be difficult for pathologists to identify what types of cells are positive, particularly when focal positivity is obtained.
- Line 326_6.2. False positive and false negative results due to uneven formalin fixation. As already suggested, the author should also include the example of phosphorylated epitopes.
I was not aware of the fixation artifact of phosphorylated epitopes. I apologize and thank you.
I added a new paragraph on lines 348-353 and cited a new reference [39].
“If phosphorylated epitopes are targeted in immunostaining, the timely exposure to formalin is critical to obtaining consistent and reproducible IHC results. For example, phospho-heat shock protein-27 and phospho-S6 ribosomal protein, which are involved in post-translational modification and stress response pathways, increased in expression or phosphorylation levels. The phosphorylated epitopes are quite labile, and loss of antigenicity is reported within 1–2 hours of the delay in fixation [39].”
- Vassilakopoulou M, Parisi F, Siddiqui S, et al. Preanalytical variables and phosphoepitope expression in FFPE tissue: quantitative epitope assessment after variable cold ischemic time. Lab Invest. 2015; 95: 334–341.
- Line 715_Please, include TRIS-EDTA pH 9 which is arguably the most commonly used buffer.
Yes, I added “…or in Tris-EDTA (10 mM Tris base, 1 mM EDTA, and 0.05% Tween 20), pH 9.0” to the list of hydrated solutions in lines 722-723.
June 10, 2021
Reviewer 3 Report
Dear editor.
Overall, the author has included the comments/ammendments proposed by this reviewer. In my opinion the work has been improved with the changes added. The fact that still being in many cases a personal experience based work (which is very interesting) but at the same time the increasse of the bibliography of about 25% is very welcomed.
On the other hand, the improve in english and the more consistent way of presenting the large amount of figures (also thankfull) is also welcome.
In my opinion, this work is ready for publication in the present state.
Author Response
Point-by-point responses
Cells-1219134-R2
Reviewer 3
Overall, the author has included the comments/ammendments proposed by this reviewer. In my opinion the work has been improved with the changes added. The fact that still being in many cases a personal experience-based work (which is very interesting) but at the same time the increase of the bibliography of about 25% is very welcomed.
On the other hand, the improve in english and the more consistent way of presenting the large amount of figures (also thankfull) is also welcome.
In my opinion, this work is ready for publication in the present state.
Newly revised portions are highlighted in blue font in the text.
In this version of the manuscript, I tried to extinguish the hint of “personal experience” as much as possible, but I am not sure that I have satisfied the reviewer’s criticism, and I am very sorry for this limitation.
I deleted the text suggesting personal experience, as much as I could, with these line numbers referring to the revised manuscript:
According to the author’s experience, etc. (lines 56, 797, 1155)
The phrase “The author strongly recommends pressure pan cooking,…” was changed to;
Pressure pan cooking is strongly recommended,… (lines 725-727)
We happen to...;
Occasionally, false negativity of nuclear antigens … can be experienced (lines 767-769)
The manuscript was checked by a native English speaker and simultaneous interpreter, Prof. Tina Tajima at St. Marianna University School of Medicine, Kawasaki, Japan.
I sincerely appreciate and thank you very much for your help.
June 10, 2021